# Live-seq enables temporal transcriptomic recording of single cells

Wanze Chen[1,2,3,6], Orane Guillaume-Gentil[4,6], Pernille Yde Rainer[1,2], Christoph G. Gäbelein[4], Wouter Saelens[1,2], Vincent Gardeux[1,2], Amanda Klaeger[1,2], Riccardo Dainese[1,2], Magda Zachara[1,2], Tomaso Zambelli[5], Julia A. Vorholt[4,6 ✉] & Bart Deplancke[1,2,6 ✉]

Single-cell transcriptomics (scRNA-seq) has greatly advanced our ability to characterize cellular heterogeneity[1]. However, scRNA-seq requires lysing cells, which impedes further molecular or functional analyses on the same cells. Here, we established Live-seq, a single-cell transcriptome profiling approach that preserves cell viability during RNA extraction using fluidic force microscopy[2,3], thus allowing to couple a cell's ground-state transcriptome to its downstream molecular or phenotypic behaviour. To benchmark Live-seq, we used cell growth, functional responses and whole-cell transcriptome read-outs to demonstrate that Live-seq can accurately stratify diverse cell types and states without inducing major cellular perturbations. As a proof of concept, we show that Live-seq can be used to directly map a cell's trajectory by sequentially profiling the transcriptomes of individual macrophages before and after lipopolysaccharide (LPS) stimulation, and of adipose stromal cells pre- and post-differentiation. In addition, we demonstrate that Live-seq can function as a transcriptomic recorder by preregistering the transcriptomes of individual macrophages that were subsequently monitored by time-lapse imaging after LPS exposure. This enabled the unsupervised, genome-wide ranking of genes on the basis of their ability to affect macrophage LPS response heterogeneity, revealing basal *Nfkbia* expression level and cell cycle state as important phenotypic determinants, which we experimentally validated. Thus, Live-seq can address a broad range of biological questions by transforming scRNA-seq from an end-point to a temporal analysis approach.

A cell's response to an internal or external stimulus is known to be heterogeneous, with some cells responding more strongly than other seemingly similar cells[4–8]. However, it is challenging to understand how a cell's molecular state affects the speed and the extent to which it responds to a perturbation such as an inflammatory signal or a differentiation stimulus[7,9]. The main obstacle is that most genome-wide profiling methods destroy the cell, which makes follow-up molecular or phenotypic experiments on this same cell impossible.

Several cell profiling approaches have been developed in recent years to address this limitation of destructive sampling, which can be broadly grouped into two categories. First, purely computational approaches will infer a cell's past on the basis of a snapshot measurement, either through connecting similar cellular states into a continuous trajectory or through modelling messenger RNA splicing dynamics[10,11]. Although insightful, the models generated by these tools should still be interpreted as statistical expectations rather than the actual transition path of the cells[12,13]. Moreover, the power of these techniques to correctly infer a cell's past at longer time scales and at the individual gene level

remains controversial[14,15]. A second category of approaches will tag a cell or some of its molecules. For example, metabolic labelling of mRNA can distinguish newly synthesized mRNA molecules from old ones[14–16]. Alternatively, cells can be tagged genetically, and their daughter cells can then be tracked independently for molecular or phenotypic profiling[17–22]. Although these molecular approaches are powerful, they are typically only applicable on short time scales or on a few features per cell, and depend on several biological assumptions to infer the initial state of a cell. These assumptions include homogenous degradation rates across cells and genes or maintenance of a molecular state over generations (for example, when using the cell carbon-copying approach[8]), which can be violated at the level of individual genes[23].

Here, we introduce Live-seq, a technology that keeps the cell alive after transcriptome profiling by using a cytoplasmic biopsy. The approach is based on fluidic force microscopy (FluidFM)[2,3,24], which couples force control with volume control[2,24] and was previously shown to allow single-cell extractions[3]. By optimizing both the FluidFM procedure and a low-input RNA-sequencing (RNA-seq) approach[25] and

[1]Laboratory of Systems Biology and Genetics, Institute of Bio-engineering and Global Health Institute, School of Life Sciences, Swiss Federal Institute of Technology (EPFL), Lausanne, Switzerland. [2]Swiss Institute of Bioinformatics, Lausanne, Switzerland. [3]CAS Key Laboratory of Quantitative Engineering Biology, Shenzhen Institute of Synthetic Biology, Shenzhen Institute of Advanced Technology, Chinese Academy of Sciences, Shenzhen, China. [4]Department of Biology, Institute of Microbiology, ETH Zurich, Zurich, Switzerland. [5]Laboratory of Biosensors and Bioelectronics, Institute for Biomedical Engineering, ETH Zurich, Zurich, Switzerland. [6]These authors contributed equally: Wanze Chen, Orane Guillaume-Gentil, Julia A. Vorholt, Bart Deplancke. ✉e-mail: jvorholt@ethz.ch; bart.deplancke@epfl.ch

combining them, we show that high-quality, cytoplasmic mRNA can be withdrawn from live, single cells in an amount that is compatible with transcriptome profiling. As this procedure can be performed while keeping the cells alive, it allows to directly couple the current state of a cell to its downstream molecular and phenotypic properties. We demonstrate how Live-seq can be used to perform sequential molecular profiling of the same cells. In addition, we show that Live-seq can function as a transcriptomic recorder by preregistering the transcriptomes of individual macrophages to subsequently identify factors underlying macrophage lipopolysaccharide (LPS) response heterogeneity, uncovering basal *Nfkbia* expression level and cell cycle state as important phenotypic determinants.

## Results

### The basics of Live-seq

We have previously shown that FluidFM can be used to sample a fraction of the cytoplasm of a cell, while preserving its viability[3]. The quantitative PCR (qPCR) assays for selected house-keeping genes thereby demonstrated that the collected biopsies contain mRNA, raising the exciting possibility that FluidFM could be used to profile the transcriptome of a live cell. Before exploring this opportunity, we first optimized the FluidFM extraction procedure to maximize RNA recovery. This is because we anticipated that cytoplasmic biopsies of about 1 pl would only yield a few picograms of RNA, down even to subpicograms, given that the total RNA content in mammalian cells ranges from as little as 1 up to 50 pg, dependent on cell type[26]. We thus set out to minimize the degradation of sampled RNA and the loss of (already picolitre-scale) sample during the cell-to-tube transfer by (1) reducing the extraction time, (2) lowering the temperature, (3) implementing a preloading of the FluidFM probe with sampling buffer with the goal of immediately mixing the extracted cytoplasmic fluid with RNase inhibitors, (4) releasing the extract into a microlitre droplet containing buffer that is compatible with downstream RNA-seq and (5) implementing image-based cell tracking for sequential extraction (relevant for sequential Live-seq as described below). The process is detailed in Fig. 1a and Methods.

In parallel, we aimed to maximize the generation of complementary DNA from the extracted mRNA. As Smart-seq2 (ref. [25]) was widely appreciated as one of the most sensitive RNA-seq methods to detect low amounts of RNA[27–29] at the time of method development, we examined whether it could amplify cDNA at the expected picogram scale. Although successful for RNA inputs above 5 pg, little to no cDNA was recovered from inputs of 2 pg or less (Extended Data Fig. 1a). To improve recovery, we systematically assessed and subsequently optimized the efficiency of each step in the workflow (detailed in Supplementary Note). These efforts yielded a protocol that enables the reliable detection of 1 pg of total RNA (Extended Data Fig. 1b, detailed in Supplementary Fig. 1a–f and Methods). Read-outs of several parameters attested to the high sensitivity of the method (Extended Data Fig. 1c, d)[25], including (1) the high rate of uniquely mapped reads, (2) the number of detected genes and (3) the gradual increase of the cumulative proportion of each library assigned to the top-expressed genes from 1 pg input RNA, while absent in the negative control (0 pg input RNA). Consistent with this, we observed that the reads from the negative control are overrepresented by poly A and TSO sequence stretches and that they map to only few genes (Extended Data Fig. 1e, f).

Next, we combined FluidFM-based cytoplasmic sampling with our enhanced, low-input RNA profiling workflow to explore the possibility of profiling transcriptomes from live cells. As an initial proof of concept, we sampled mouse immortalized brown preadipocyte (IBA) cells[30] and detected on average about 2,100 genes (nGene) at a sequencing depth of around 1 million reads per sample (Fig. 1b). As the same FluidFM probe can be used for sequential sampling of several cells, a wash process was implemented to prevent cross-contamination (>99% accuracy based on read mapping) (Extended Data Fig. 1g and Methods).

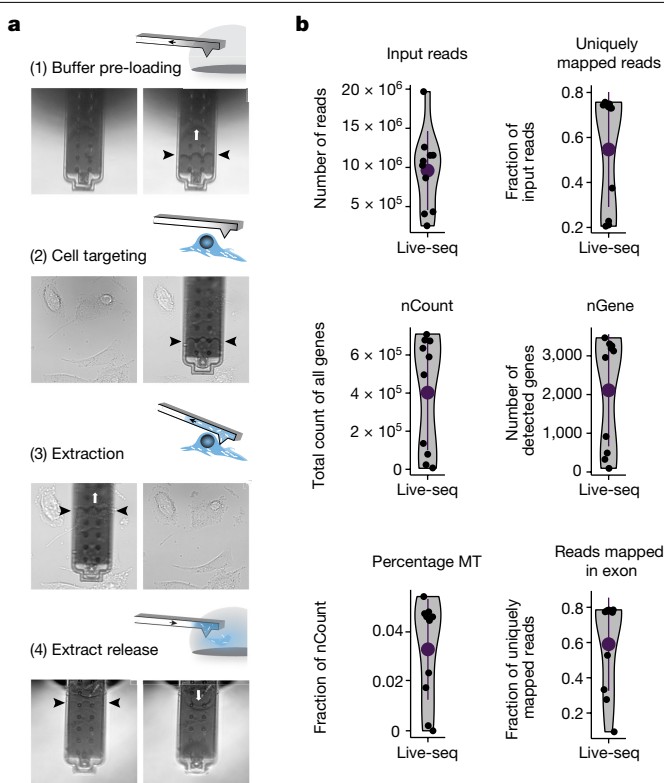

**Fig. 1 | Live-seq combines optimized FluidFM-based live-cell biopsy with enhanced Smart-seq2 RNA-seq. a**, Illustration and representative images of the Live-seq sampling procedure using FluidFM (here, applied on brown preadipocyte IBA cells). The white arrows indicate the application of under- or overpressure. The black arrows indicate the amount of buffer and extract in the probe. Scale bar, 20 µm. **b**, Quality control of Live-seq applied on IBA cells based on the parameters that are listed above each panel. *n* = 10 cells. nGene, number of detected genes; nCount, total count of all genes; percentage MT, percentage of counts from mitochondrial genes. Error bars represent mean ± s.d.

Together, these technological advances constitute the methodological foundation of Live-seq for transcriptome profiling using cytoplasmic biopsies.

### Stratification of cell types and states

To assess the cell identity- or even cell state-resolving power of Live-seq-derived cell transcriptomes, we applied our approach to distinct cell types (Fig. 2a). These included IBA cells, primary mouse adipose stem and progenitor cells (ASPCs) and two monocyte or macrophage-like RAW264.7 cell lines: one wild-type and one RAW264.7 subline (RAW-G9) containing an mCherry reporter driven by the *Tnf* promoter[31] to facilitate downstream functional analyses (shown below). The latter cell line and its parental RAW264.7 line are molecularly and phenotypically similar[31] and will therefore be further referred to as 'RAW' cells. In total, we acquired 641 samples across five replicates and generated 588 libraries for sequencing (see Methods for a description of the different datasets). From these, 294 samples passed our filtering criteria, including (1) more than 1,000 genes per sample, (2) fewer than 30% of reads derived from mitochondrial genes and (3) more than 30% uniquely mapped reads (Supplementary Fig. 2a and Methods). Similar to Smart-seq2 (ref. [25]), reads spanned the full length of transcripts (Supplementary Fig. 2b). An average of 4,112 genes were detected (Extended Data Fig. 2a) with further quality controls shown in Supplementary Fig. 2c, d.

We observed that the Live-seq data separated into five clusters largely determined by cell type and state (treatment) rather than library

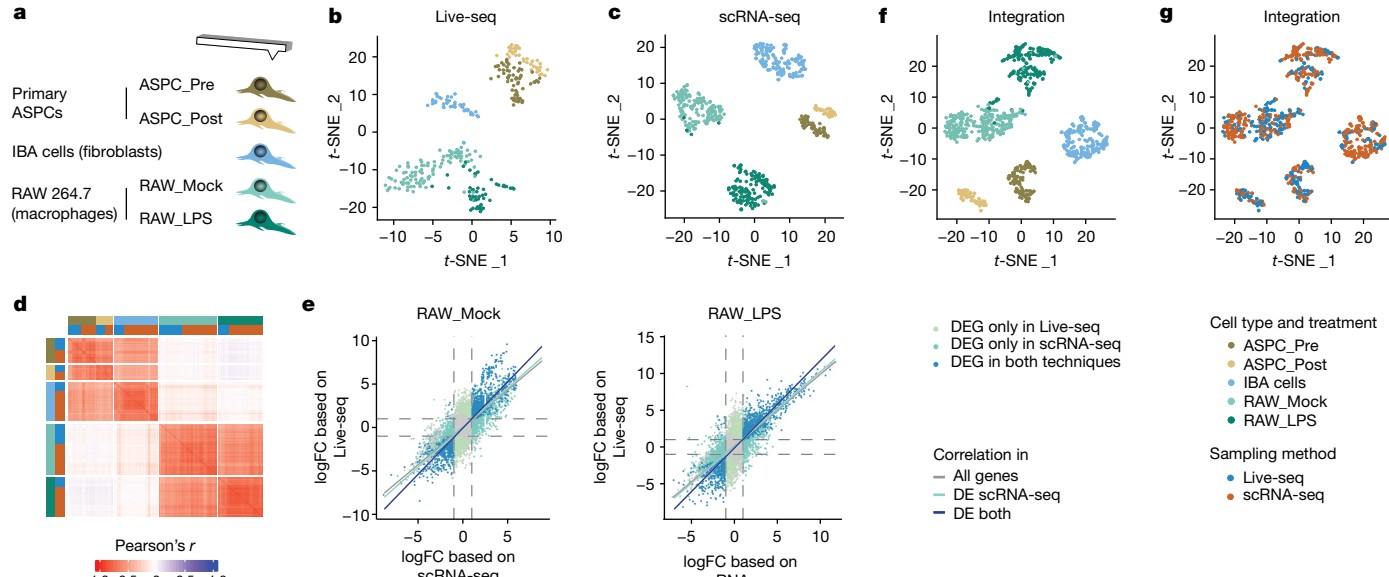

**Fig. 2 | Live-seq enables the stratification of cell type and state (treatment).**
**a**, Experimental setup. 100 nM LPS or PBS was used for RAW_LPS and RAW_Mock, respectively. A chemical cocktail was used to differentiate ASPCs (Methods). ASPC_Pre and ASPC_Post denotes ASPCs pre- and post- (2 days)-adipogenic differentiation induction. **b**, *t*-SNE projection of Live-seq data coloured by cell type/states. *n* = 61 for ASPC_Pre, *n* = 37 for ASPC_Post, *n* = 44 for IBA, *n* = 102 for RAW_Mock, *n* = 50 for RAW_LPS. **c**, *t*-SNE projection of scRNA-seq data (Smart-seq2) coloured by cell type or state. *n* = 60 for ASPC_Pre, *n* = 35 for ASPC_Post, *n* = 153 for IBA, *n* = 157 for RAW_Mock, *n* = 149 for RAW_LPS. **d**, Gene expression correlation (Pearson's *r*) between all the cells from both scRNA-seq and Live-seq. **e**, A direct comparison of the log fold gene expression change derived from Live-seq and scRNA-seq data when comparing the cluster of cells corresponding to a focal cell state (here, left, RAW_Mock and left, RAW_LPS) to the rest of all the cells (Methods). For the correlation, *P* = 2.2 × 10⁻¹⁶ for all conditions (All genes, DE scRNA-seq and DE both), two-sided *F*-test. **f**, **g**, Visualization of Live-seq and scRNA-seq data after anchor-based data integration (Methods) reveals no obvious molecular differences. *t*-SNE projection of the integrated Live-seq and scRNA-seq data according to cell type and state (treatment) (**f**) and approach (**g**).

complexity (number of genes detected), replicate identity or RAW cell subline (Fig. 2b, Extended Data Fig. 2b–d and Methods). This supports Live-seq's ability to effectively profile and distinguish both primary and cultured cells, and distinct cell states. The analysis per cell type comparing ASPC and RAW cells before and after stimulation thereby showed consistent results (Extended Data Fig. 2e, f).

To validate the biological relevance of the five clusters, we performed differential gene expression analysis (Supplementary Table 1) followed by Gene Ontology enrichment, yielding results that were largely consistent with the cellular characteristics of each cluster (Extended Data Fig. 3a, b, Supplementary Table 1 and Methods). In addition, matching of the top 100 marker genes to Mouse Gene Atlas annotations[32] confirmed the expected molecular similarity of clusters 1, 2 and 3 cells to mouse embryonic fibroblasts and osteoblasts (Extended Data Fig. 3c), reflecting their mesenchymal or fibroblast nature. Cluster 4 cells were correctly annotated as RAW264.7/macrophages without treatment and cluster 5 cells as macrophages with LPS treatment (Extended Data Fig. 3c).

To evaluate whether cytoplasmic mRNA biopsies can act as suitable representations of full (lysed) cell transcriptomes, we compared Live-seq gene expression profiles to those obtained using the whole-cell Smart-seq2 assay (Methods). Using the same cell types and treatment groups, we sequenced 573 cells with scRNA-seq, with 554 cells passing the same filtering criteria applied to Live-seq samples (Supplementary Fig. 3). We detected, as expected, a greater average of genes per cell compared to Live-seq: 8,328 genes at a sequencing depth of around 70,000 reads per cell with the quality control parameters attesting to high data quality (for metrics shown side-by-side with those from Live-seq, Supplementary Fig. 3). Graph-based clustering accurately captured the underlying cell type and/or state (treatment) groups (Fig. 2c, Extended Data Fig. 4a). Overall, the Adjusted Rand Indexes comparing the clustering with the ground truth were high for both Live-seq and scRNA-seq (0.81 and 0.95, respectively), revealing high clustering accuracy (Extended Data Figs. 2b and 4a). Although cell types were easily identified (Extended Data Figs. 2c and 4b), a few cells were incorrectly assigned to cell treatment (Extended Data Figs. 2d and 4c). This observation was consistent across the two techniques, which probably reflects the heterogeneity of the magnitude of cell responses to the treatments. However, slightly more cells were misassigned in Live-seq data. Specifically, for both ASPCs and RAW cells combined, these represented 7.6 versus 2% of the total number of cells, potentially due to the lower sensitivity of Live-seq compared to scRNA-seq. Results obtained by differential gene expression analysis (Extended Data Fig. 4d and Supplementary Table 2) followed by Gene Ontology enrichment (Extended Data Fig. 4e) were in line with those obtained by Live-seq (Extended Data Fig. 3b,c) in that the five clusters correctly annotated mesenchymal or fibroblast-like cells and mock-treated or LPS-treated macrophages, respectively (Extended Data Fig. 4f).

We then integrated the Live-seq and scRNA-seq data to explore putative molecular differences between the two conceptually distinct approaches. We found that cells of the same type and state highly correlated regardless of the sampling method (that is, Live-seq and scRNA-seq) (Fig. 2d). Even without batch correction, we observed a dominance of cell type/state signals over those stemming from the sampling approach (Extended Data Fig. 5a). Moreover, the differentially expressed (DE) genes derived from the Live-seq and scRNA-seq datasets overlapped to a large extent (Extended Data Fig. 5b), supported by a high correlation in fold changes when comparing each cell state to the rest of all the cells (here, shown for RAW cells, Fig. 2e) and within each cell type (Supplementary Fig. 4a). Compared to Live-seq, scRNA-seq yielded a larger number of DE genes that were not detected by Live-seq. Down-sampling the scRNA-seq data to a library complexity that was similar to that of Live-seq reduced the proportion of DE genes that were detected by scRNA-seq only (82% for ASPCs and 78% for RAW cells,

Supplementary Fig. 4b, c), suggesting that, as expected, scRNA-seq provides greater power than Live-seq. Gene Ontology enrichment on the DE genes that were specifically identified by either Live-seq or scRNA-seq did not reveal a striking pattern of enriched terms across the different cell types (Supplementary Fig. 4d), indicating that Live-seq is not biased towards a specific set of genes. Correlation analysis further revealed that a meta read-out of cells (that is, collapsing all cells into a virtual bulk sample) produced the expected pattern with the Live-seq gene expression profiles of IBA and pre- and postdifferentiation ASPC correlating highest with the scRNA-seq-based ones and the same for RAW cells with and without LPS treatment, respectively (Extended Data Fig. 5c). We finally applied a Seurat[33]-embedded, single-cell-tailored canonical correlation analysis (CCA) and mutual nearest neighbours (MNNs)-based approach to remove any potential technical batch effects. This further consolidated the clustering of cells according to cell type and state (Fig. 2f), as illustrated by specific marker gene expression (Extended Data Fig. 5d) regardless of sampling approach (Fig. 2g) and sequencing depth (Extended Data Fig. 5e, Supplementary Fig. 5 and Methods).

Together, these results demonstrate that Live-seq enables the stratification of cell types and states similar to conventional scRNA-seq, but without the need to lyse cells.

## Live-seq's impact on cellular functions

The analyses described above allowed us to benchmark Live-seq to a widely used conventional, whole-cell transcriptome profiling approach. However, for Live-seq to be effective as a temporal rather than end-point analysis tool, it must preserve cell viability and not inflict any undesired perturbations on the sampled cells. To test these capacities, we first evaluated cell viability after sampling (Extended Data Fig. 6a and Methods). Cell viability was similar for the three cell types, ranging between 85 and 89%. This was only slightly lower than that after conventional trypsin-based cell dissociation (90–95%, data not shown). Cell viability did not scale with the extracted volume, which itself ranged from 0.2 to 3.5 pl with an average of 1.1 pl (Extended Data Fig. 6a and Methods). We note here that, even though the extracted volume can be readily determined, the one of the extracted adherent cells cannot (for cell volume measurements of dissociated cells, see Extended Data Fig. 6b). Although we observed a very weak correlation between extracted volume and the number of detected genes and read count, especially when all cell types were considered together (Extended Data Fig. 6c), 'extracted volume' as a feature did not affect cell clustering (Extended Data Fig. 6d). Together with the relatively high variation in number of detected genes on a per assay basis also for conventional scRNA-seq approaches[34] and the fact that scRNA-seq and Live-seq data integrated well, these results indicate that Live-seq and scRNA-seq may be subject to comparable technical constraints linked to limited RNA input.

The observed high cell viability suggests that cells are able to quickly recover after a portion of their cytoplasm has been removed. To study the recovery of Live-seq-probed cells in greater detail, we next examined the growth dynamics of RAW cells after Live-seq using time-lapse microscopy. We focused on RAW cells for two reasons: (1) their semi-adherent nature and consequently, almost spherical shape allowed for a straightforward approximation of their cell volume over time; and (2) their size is the smallest among the three sampled cell types, and thus their behaviour may be most affected by a cytoplasmic extraction. We first measured longitudinal cell volume profiles for control RAW cells that were not subjected to Live-seq sampling (Extended Data Fig. 7a). The obtained volume profiles showed a continuous increase under normal growth conditions, whereas they showed a plateau after exposure to LPS, consistent with the known ability of LPS to arrest the macrophage cell cycle[35]. Upon mitosis, the two daughter cells featured equal volumes, representing 50% of the mother cell, followed by overlapping volume growth. We next examined cells that underwent Live-seq sampling, and were subsequently exposed to LPS.

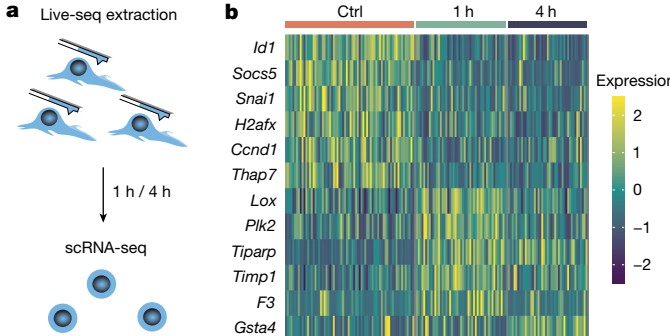

**Fig. 3 | Live-seq induces few expression changes. a**, Schematic of the experimental design to evaluate putative transcriptomic changes after Live-seq extraction. IBA cells were first extracted after which they were collected and subjected to scRNA-seq 1 h and 4 h postextraction. Cells that were not extracted were included as controls. **b**, All of the genes (12) that were found DE between the control and Live-seq-sampled IBA cells from **a**.

Upon extraction, a drop in cell volume was observed. This decrease matched the measured volume of cytoplasmic fluid collected in the FluidFM probe, and represented 40% to 70% of the total cell volume (Extended Data Fig. 7a–c). The extraction was followed by a steep cell volume increase, whereby the extracted RAW cells recovered their pre-extraction volume within 100 to 320 min and then continued to grow with a general growth pattern that was similar to the control cells (Extended Data Fig. 7a, b). Moreover, even though overall cell behaviour was variable, we observed that some cells divided already a few hours post-Live-seq sampling, in similar fashion to non-extracted cells (Extended Data Fig. 7a, d). Although we cannot rule out that Live-seq introduces a small cell cycle delay, as cell size and cell division tend to be intrinsically coupled[36], our results indicate that cells quickly recover their volume and still progress through their cell cycle even after having been subjected to a cytoplasmic biopsy.

Assessing cell viability and growth dynamics, however, does not exclude putative, short-term molecular effects that Live-seq may impose on targeted cells that may not be visible at the phenotypic level. To address this, we used standard scRNA-seq (Methods) to profile the individual transcriptomes of IBA cells 1 h and 4 h postcytoplasmic sampling (Fig. 3a), while using non-probed cells as negative controls. Quality controls on the cDNA and sequencing libraries (such as mapping rate, number of detected genes) were indistinguishable among the three cell groups (control, 1 and 4 h post-Live-seq sampling; Extended Data Fig. 7e, f). Data analysis revealed no distinct condition-related clusters (Extended Data Fig. 7g), further supported by the observation that only 12 genes were found to be significantly DE among the three conditions (Fig. 3b). To investigate subtle changes in biological processes that may not be visible at the DE level (because of hard cut-offs), we performed a Gene Set Enrichment Analysis on Gene Ontology biological process terms using the expression levels of all genes, but this did not reveal any specifically enriched processes. Thus, these experiments suggest that Live-seq does not induce major gene expression alterations.

Altogether, we conclude that Live-seq enables the profiling of cell transcriptomes without imposing major perturbations on a cell's basic properties such as viability, growth or transcriptome. This in turn opens a new avenue to link a cell's state directly to its present and future molecular and phenotypic properties, paving the way to directly map a cell's trajectory or to record molecular events that are predictive of a cell's downstream phenotype, as illustrated below.

## Sequential Live-seq cell sampling

A large number of computational approaches have been developed to infer cellular dynamics from one-time sampling data[10]. Although

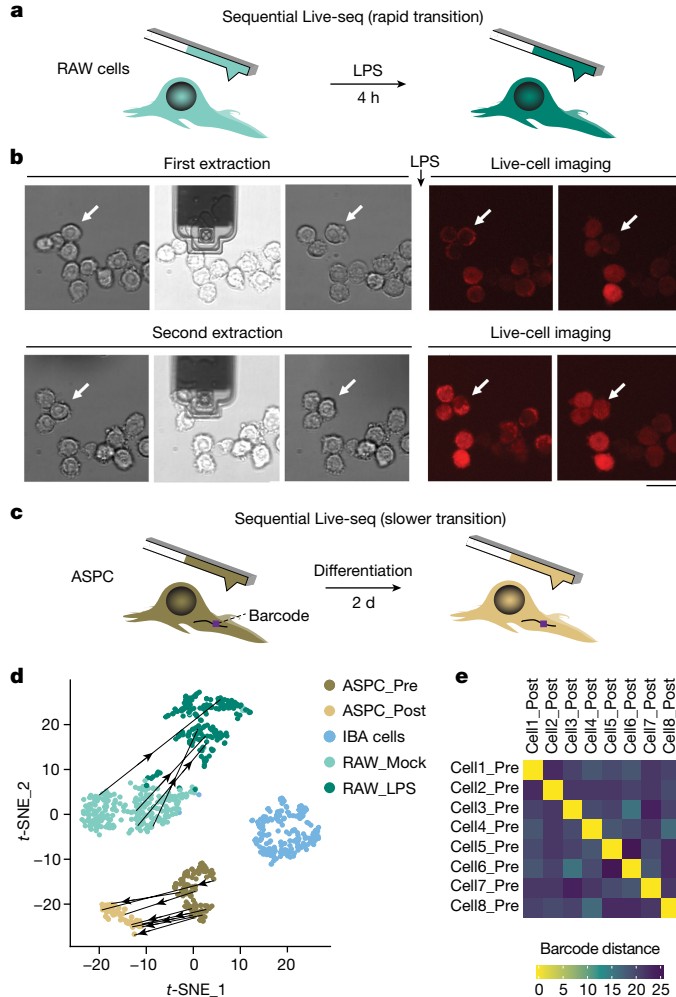

**Fig. 4 | Live-seq enables sequential single-cell transcriptomic sampling.** **a**, Schematic illustrating the sequential sampling procedure in a rapid cell state transition model: LPS stimulation of RAW cells. **b**, Time-lapse brightfield and mCherry fluorescence images of sequential Live-seq sampling in RAW cells (that is, sequential extractions). The white arrow points to the sequentially sampled cell. Representative images of one of the 24 sequentially sampled cells are shown. Scale bar, 20 μm. **c**, Schematic illustrating the sequential sampling procedure in a slower cell state transition model: adipogenic differentiation of ASPCs. **d**, t-SNE-based visualization of integrated Live-seq and scRNA-seq data, highlighting the direct trajectory of sequentially sampled cells (12 cells in total) from one state to another. **e**, Heatmap showing the Hamming distance of the tracking barcodes among the sequentially sampled ASPCs (Methods).

clearly helpful, these approaches suffer from the fact that definition of the original molecular cell state is highly challenging in the absence of direct measurements[37]. To alleviate this issue, we sought to establish a proof-of-principle, sequential Live-seq sampling approach, which would allow us to record a cell's molecular signature before and after cell state transitioning. To do so, we considered a rapid and a slower cell state transition model: response to extracellular stimuli (LPS) (Fig. 4a, b) and differentiation (adipogenesis) (Fig. 4c), respectively.

For the former model, we sampled 24 RAW cells for a first time, then stimulated them with LPS, after which we sampled the same cells a second time. In addition, the respective cells were also monitored by time-lapse microscopy after both sampling events (Fig. 4a, b and Methods). These efforts yielded four sequentially probed cells, reflecting the relatively modest proportion (roughly 40% per extraction round) of

samples that passed the transcriptome quality control cut-off criteria (Methods). To verify whether the respective gene expression profiles correctly showed the transition from a basal to an LPS-stimulated state, we projected the Live-seq sequential sampling information on top of the integrated Live-seq and scRNA-seq data shown in Fig. 2f,g. We found that the sequentially sampled cells, each constituting two distinct points in the same t-distributed stochastic neighbour embedding (t-SNE) map, mapped correctly to their respective cell state clusters, providing a direct, transcriptome-wide read-out of a cell's trajectory (Fig. 4d and Extended Data Fig. 8a).

Similarly, we set out to directly measure the adipogenic transition from a pre- to a post-differentiation state (Fig. 4c). To assure that the same ASPCs were sampled twice within this longer time window and thus can be paired, we implemented additional tracking strategies. These involved the use of a unique barcode in the 3′ untranslated region of a green fluorescent protein (GFP) reporter that was lentivirally transduced in a subset of ASPCs that were subsequently mixed with wild-type ASPCs on cell seeding (Methods). Using this procedure, we were able to sequentially sample 44 cells and obtain eight paired, quality control-passing gene expression profiles from ASPCs before differentiation and 2 days after adipogenic cocktail induction, as confirmed by the recovery of the correct, respective barcodes (Fig. 4e). Further monitoring of the cells for up to 7 days after the second extraction revealed no drop in cell viability of extracted cells (two out of 44 cells died, 95%) compared to control, non-extracted cells (three out of 41 cells died, 93%). In addition, we observed lipid droplets in these sequentially probed cells indicative of their retained adipogenic differentiation capacity (for representative images, see Extended Data Fig. 8b). Projection of the retrieved 16 ASPC transcriptomes (eight pairs) onto the integrated Live-seq and scRNA-seq data revealed the expected transition from pre- to differentiating ASPCs (Fig. 4d and Extended Data Fig. 8a).

These results demonstrate that for both transition models, Live-seq data could be exploited to establish the true trajectory of cells that were processed using conventional scRNA-seq. This contrasts with an inference-based analysis that we conducted further on the basis of both co-expression and RNA velocity, which revealed several spurious connections between unrelated cell states on scRNA-seq data, when all cell types were combined as well as when each cell type was analysed separately (Extended Data Fig. 8c, d and Supplementary Fig. 6a, b). Even a limited number of Live-seq sampled cells, as provided here, might thus provide a reference coordinate system to map cell trajectories of existing and future scRNA-seq datasets. Taken together, sequential sampling with Live-seq allows the acquisition of transcriptomic dynamics from the same cell, thus providing a direct read-out of both rapid and slower cell state transitions.

## Recording ground-state transcriptomics
It is well recognized that macrophages, including the RAW cells, respond to LPS in a heterogeneous fashion[31]. Although this has been the subject of intense interrogation[38], no systematic, genome-wide analysis of the molecular factors driving this heterogeneity has been performed. We used Live-seq to profile the transcriptomes of individual RAW cells to link the molecular state of each macrophage to its downstream, LPS-induced phenotype. Specifically, we first examined single RAW cells with Live-seq to record their respective transcriptomes in the ground state. We then subjected the same cells to LPS treatment and monitored LPS-induced *Tnf* promoter-driven mCherry expression (hereafter referred to as *Tnf*-mCherry) by live-cell imaging (Fig. 5a). We observed that *Tnf*-mCherry intensity increased on LPS treatment, although this intensity varied greatly between RAW cells (Extended Data Fig. 9a), consistent with previous findings[31]. No significant difference in the heterogeneity of *Tnf*-mCherry intensity profiles was observed between cells subjected to Live-seq sampling and those that were not (Extended Data Fig. 9a, two-sided Wilcoxon rank-sum test, $P = 0.88$).

These results provide additional support to the notion that Live-seq does not markedly affect a cell's phenotype.

Given the canonical heterogeneous response of RAW cells to LPS (Extended Data Fig. 9a), we then set out to uncover the principal ground-state molecular factors (as derived by Live-seq) that drive this heterogeneity. We used a linear regression model correlating the expression of Live-seq-detected genes within individual cells with the corresponding *Tnf*-mCherry response profiles. Specifically, given that we observed that the log form of each response profile is a linear function during a 3 to 7.5 h post-LPS treatment time window (Extended Data Fig. 9b, detailed in Supplementary Fig. 7), we aimed to use ground-state gene expression data to predict the two principal *Tnf*-mCherry profile parameters: the basal *Tnf*-mCherry expression (intercept) and the rate of fluorescence intensity increase (slope) (Fig. 5b and Extended Data Fig. 9c). Applying the linear model revealed *Tnf* as the best predictor ($R^2 = 0.50$, false discovery rate (FDR) of 0.57) of basal *Tnf*-mCherry intensity, validating the accuracy of Live-seq (Extended Data Fig. 9c and Supplementary Table 3). Next, we set out to identify factors that predict both the dynamics and amplitude to which a cell responds to LPS as defined by the rate of *Tnf*-mCherry intensity increase. Using the Live-seq data, we generated an unsupervised rank of factors according to which extent they affected a macrophage's ability to respond to LPS. We thereby identified gelsolin (*Gsn*) as one of the top negatively correlating factors ($R^2 = 0.33$, FDR = 0.51) (Extended Data Fig. 9d), consistent with its role in suppressing LPS-induced *Tnf* expression[39], probably by direct LPS inactivation[40]. The strongest transcriptional predictor of the rate of *Tnf*-mCherry intensity increase was *Nfkbia* (also termed NF-kappa-B inhibitor alpha, *IkBα*; $R^2 = 0.60$, FDR = 0.10) (Fig. 5b, c and Supplementary Table 4). This anticorrelation only emerged because of Live-seq's ability to measure ground-state gene expression and downstream phenotypic response, whereas conventional, 'end-point' scRNA-seq data in fact revealed a positive correlation between *Nfkbia* and *Tnf* (Fig. 5d and Extended Data Fig. 9e). The *Nfkbia*-*Tnf* anticorrelation is consistent with the well-known function of NFKBIA as a key negative regulator of the LPS-NF-κB signalling pathway by suppressing NF-κB, which itself regulates the co-expression of both *Nfkbia* and *Tnf*[41]. These findings highlight the complementarity of the Live-seq-based dynamics approach and the scRNA-seq-based snapshot assay. Moreover, although many genes are known to regulate the LPS-NF-κB signalling pathway, including IkappaB Kinases as positive regulators and A20 as a negative one[41], the transcriptome-wide read-out enabled by Live-seq points to *Nfkbia* expression as the strongest predictor of LPS-induced *Tnf* expression. Thus, although Live-seq sampling is currently throughput-limited with 4–5 extractions per hour due to downstream processing and following the fate of individual cells by live imaging, it still provided sufficient data to generate testable hypotheses.

To validate the hypothesis that *Nfkbia* expression can act as a transcriptional predictor of LPS-induced *Tnf* expression, we first analysed transcriptomic data from several primary macrophage populations. We uncovered *Nfkbia* expression as the most variable among the annotated NF-κB pathway components (Extended Data Fig. 9f), supporting *Nfkbia* expression as an important driver of macrophage LPS response heterogeneity. Next, we set out to experimentally validate our Live-seq-based findings in RAW cells by generating a RAW-G9 line containing a blue fluorescent protein (BFP) reporter under the control of the *Nfkbia* promotor. We observed that LPS treatment effectively induced an increase in *Nfkbia*-BFP fluorescence intensity, which was synchronous with the increase in fluorescence intensity of the *Tnf*-mCherry reporter, similar to the endogenous gene[38] (Extended Data Fig. 9g). These observations suggested that the generated reporter can be used as a proxy for *Nfkbia* expression. We thus demonstrate that, as proposed, basal *Nfkbia*-BFP intensity is a transcriptional predictor ($R^2 = 0.11$, $P = 0.0008$, $F$-test, Pearson's $r = -0.34$) of the rate of *Tnf*-mCherry intensity increase (the slope) (Fig. 5e and Extended Data Fig. 9h for an independent experimental replicate).

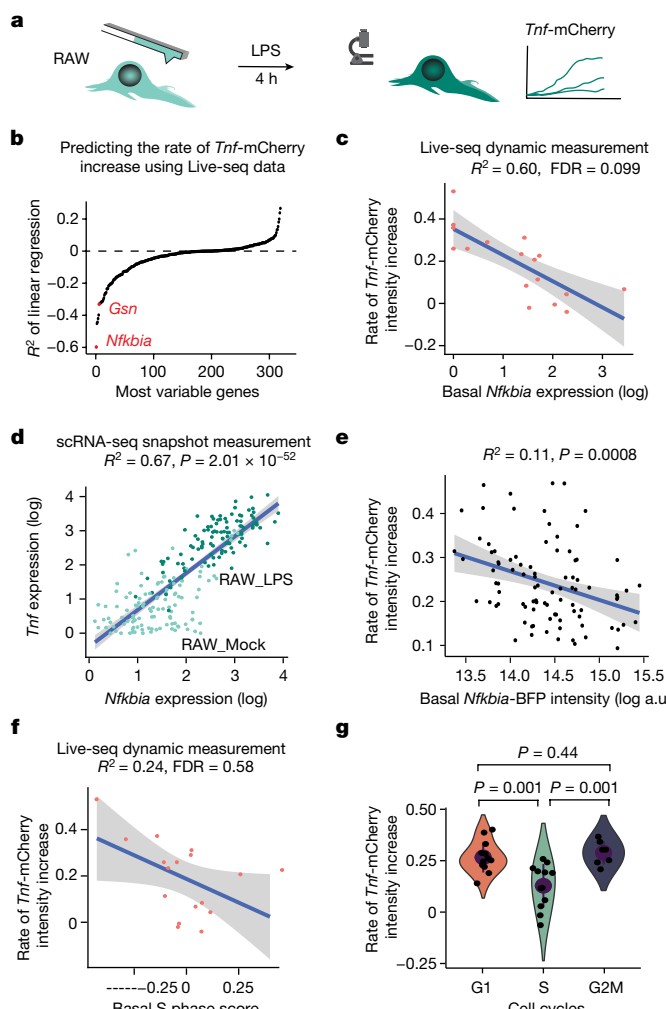

**Fig. 5 | Live-seq links transcriptomic state with subsequent functional analysis on the same cells. a**, Schematic illustrating the coupling of Live-seq with live-cell imaging. RAW cells were first subjected to Live-seq and subsequently exposed to LPS while tracking *Tnf*-mCherry fluorescence by time-lapse imaging. **b**, A linear regression model was used to predict the slope (the rate of *Tnf*-mCherry fluorescence intensity increase) calculated from data shown in Extended Data Fig. 9b and Supplementary Fig. 7 on the basis of ground-state gene expression data recorded by Live-seq. The most variable genes from both Live-seq and scRNA-seq data were used, whereas genes with dispersion <0.1 in Live-seq samples were removed (Methods). **c**, Basal *Nfkbia* expression, as determined by Live-seq, anticorrelates with the rate of *Tnf*-mCherry fluorescence intensity increase. $R^2$ and FDR are listed. **d**, The expression of *Nfkbia* and *Tnf* is highly correlated in conventional scRNA-seq data. $P = 2 \times 10^{-52}$, two-sided $F$-test. **e**, Validation of *Nfkbia* expression as a predictor of the extent by which a macrophage will respond to LPS using a *Nfkbia*-BFP reporter (Methods). $n = 91$, $R^2 = 0.11$, $P = 0.0008$, two-sided $F$-test; Pearson's $r = -0.34$, $P = 0.0008$. Another biological replicate is shown in Extended Data Fig. 9h. a.u., arbitrary units. **f**, The (basal) cell cycle S phase score of Live-seq samples (inferred from respective transcriptomes) anticorrelates with the rate of *Tnf*-mCherry fluorescence intensity increase, suggesting that cells in S phase respond weaker to LPS treatment. $R^2$ and FDR values are listed. The error band (**c**–**f**) represents the s.d. **g**, Validation of cells in S phase responding weaker to LPS exposure using a Fucci cell cycle indicator (Extended Data Fig. 10b,c and Methods). $n = 32$ cells over two independent experiments. Error bars represent the mean ± s.d. and $P$ values were determined by a two-sided Wilcoxon rank-sum test.

Finally, we correlated the ground-state cell cycle phase of each cell, as inferred from prerecorded Live-seq data, to its corresponding downstream LPS response profile. This analysis was motivated by

LPS's reported capacity to inhibit macrophage cell cycle progression[35], which seems consistent with the observed anticorrelation between the conventional scRNA-seq-derived S phase score of RAW cells and *Tnf* expression (Extended Data Fig. 10a). Yet, this anticorrelation may also reflect that a macrophage's ability to respond to LPS is influenced by its cell cycle phase, which is why we exploited our Live-seq data to distinguish between these two plausible scenarios. As shown in Fig. 5f, our analysis revealed that cells tend to respond weaker to LPS when they are in S phase ($R^2 = 0.24$, FDR = 0.58). To experimentally validate this observation, we expressed a fluorescent ubiquitination-based cell cycle indicator (Fucci) miRFP709-hCdt1 (ref. [42]) in RAW cells (Methods). Before LPS stimulation, this indicator was tracked for 24 h to determine the cell cycle phase of each cell (Extended Data Fig. 10b, detailed in Supplementary Fig. 8), after which the LPS-induced *Tnf*-mCherry fluorescence intensity was measured. As the Fucci system does not allow distinguishing the S from G2M phase, the boundary between the S and G2M phase was assigned by timing. As shown in Extended Data Fig. 10b and quantified in Fig. 5g and Extended Data Fig. 10c, cells in S phase responded significantly weaker (two-sided Wilcoxon rank-sum test, $P = 0.001$ for both) to LPS stimulation (rate of *Tnf*-mCherry intensity increase) compared to their G1 and G2M phase counterparts, validating the Live-seq data. Together, these findings underscore the power of Live-seq to act as a recording tool for the transcriptomes of individual cells and predict their phenotypic behaviour.

## Discussion

It is well known that seemingly similar cells can respond differently to an external signal. However, resolving the molecular mechanisms underlying differential cellular responses has been challenging given the difficulty in coupling an initial molecular state to a downstream response. Indeed, most current methods are either limited in the number of features that can be assessed per cell[17–22] or require the destruction of cells to access their state/transcriptome. To alleviate this, we established Live-seq by coupling an enhanced FluidFM-based live-cell biopsy technology to a highly sensitive RNA-seq approach. We showed that Live-seq is capable of distinguishing cell types and states, while keeping cells alive and functional. This conclusion is based on the fact that cells proceeded in their cell cycle, remained LPS-responsive similar to control cells, preserved adipogenic differentiation capacity and showed only minor transcriptomic changes after cytoplasmic sampling.

There is great interest in new technologies that can improve our ability to determine cellular dynamics[43]. This is exemplified by the recent development of (1) scRNA-seq-coupled lineage tracing, which superimposes clonal information on single-cell transcriptomes to better resolve lineage relationships[44], and (2) methods that metabolically label newly synthesized RNA with the aim of recording transcriptional dynamics by differentiating old versus new RNA[14–16]. In particular, mRNA synthesis rates can be read out using metabolic labelling, making it a valuable tool to assess how a perturbation affects gene expression dynamics such as changes in mRNA production and degradation rates. Answering the inverse question, that is, resolving how the ground-state transcriptome of a cell affects its response to a perturbation, remains challenging because of the various assumptions that need to be made to infer a cell's initial state when using end-point methodologies (that is, involving cell lysis). Live-seq now bypasses these assumptions by directly measuring ground-state gene expression levels. As a result, Live-seq is well suited to interrogate the gene expression state of a cell before exposing this same cell to one or more perturbations, either on short or long time scales. This is the case, for example, when aiming to identify which genes, pathways or cellular processes drive a heterogeneous cell response without having to select a set of focal genes, as demonstrated with our study on variation of LPS response in macrophages. Indeed, by leveraging the ability of Live-seq to act as a transcriptomic recorder as well as mRNA expression's dominant role in determining protein levels

at steady state[45], we generated a genome-wide, statistical ranking of a gene's impact on a macrophage's response to LPS. This revealed that the initial molecular state and, in particular, the expression levels of *Nfkbia* and *Gsn*, contribute significantly to overall phenotypic heterogeneity (Fig. 5b–e and Supplementary Table 4). Moreover, we uncovered that the cell cycle phase also influences the LPS response, which we confirmed experimentally (Fig. 5f, g). Transcription and DNA replication in a cell's S phase are in inherent conflict, which can be a source of DNA damage and genome instability[46]. It would in this regard be of interest to further explore whether cells in S phase respond weaker to transcriptional stimulations as a general strategy to temporarily decouple DNA replication and transcription[47].

Because Live-seq excels over longer time scales while minimizing the need for inference, it complements already available techniques to study cellular dynamics. However, the scalability of Live-seq, compared to the other mentioned methods, is still limited. Live-seq will therefore benefit from further automation procedures that may also expand the scope of Live-seq beyond cell culture. At present with around 40% of the samples passing our data quality control criteria, an increase in mRNA detection sensitivity may further increase Live-seq's efficiency. The implementation of low-input scRNA-seq methods such as Smart-seq3 (ref. [48]), which already shares several features with the enhanced Smart-seq2 method developed here, will in this regard be highly valuable. Not only will it improve the overall sensitivity and power of Live-seq, but also its ability to work with even smaller cytoplasmic extraction volumes and thus to further reduce any possible cellular impact, which is especially important when probing smaller cell types such as T cells and certain stem cells[49].

In summary, Live-seq enables single-cell transcriptome profiling as well as downstream molecular and functional analyses on the same cell at distinct time points (Figs. 4 and 5), providing opportunities to address some of the long-standing biological questions pertaining to cell dynamics or cellular phenotypic variation. We thereby expect the next generation of the Live-seq approach to allow for sampling of many more cells, aiming to alleviate relevant statistical power and cellular resolution concerns. As such, we anticipate that Live-seq will transform single-cell transcriptomics, and possibly other omics technologies such as single-cell proteomics and metabolomics, from the current end-point type assay into a temporal analysis workflow.

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

# Methods

## Biological materials

RAW264.7, 293T and HeLa cells were obtained from ATCC. RAW264.7 cells with *Tnf*-mCherry reporter and *relA*-GFP fusion protein (RAW-G9 clone) were kindly provided by I.D.C. Fraser (National Institutes of Health). The IBA cell line derived from the stromal vascular fraction of interscapular brown adipose tissue of young male mice (C57BL6/J) was kindly provided by C. Wolfrum's laboratory (ETH Zurich). Primary ASPCs were isolated from subcutaneous fat tissue of female C57BL/6J mice at age between 7 and 10 weeks as previously described[50]. The cell lines have not been authenticated or tested for mycoplasma contamination. All these cells were cultured in high-glucose DMEM medium (Life Technologies) supplemented with 10% foetal bovine serum (FBS) (Gibco) and 1× penicillin/streptomycin solution (Life Technologies) in a 5% $CO_2$ humidified atmosphere at 37 °C and maintained at less than 80% confluence before passaging.

## Oligos

Biotinylated-olig0-dT30VN (IDT):/5Biosg/AAGCAGTGGTAT-CAACGCAGAGTACTTTTTTTTTTTTTTTTTTTTTTTTTTTTTVN.

TSO (Exiqon): AAGCAGTGGTATCAACGCAGAGTACATrGrG+G.

Iso-TSO (Exiqon): iCiGiCAAGCAGTGGTATCAACGCAGAGTrGrG+G.

Biotinylated-TSO (Exiqon): /5Biosg/AAGCAGTGGTATCAACGCAGAGTACATrGrG+G.

Hairpin-TSO (Exiqon):GGCGGCGCGCGCGCGCCGCCAAGCAGTGGTATCAACGCAGAGTACATrGrG+G.

Biotinylated-ISPCR oligo (IDT): /5Biosg/AAGCAGTGGTATCAACGCAGAGT.

Hairpin-TSO was annealed in IDT Duplex Buffer before use.

## FluidFM setup

A FluidFM system composed of a FlexAFM-near-infrared scan head and a C3000 controller driven by the EasyScan2 software (Nanosurf), and a digital pressure controller unit (ranging from −800 to +1,000 mbar) operated by a digital controller software (Cytosurge) was used. A syringe pressure kit with a three-way valve (Cytosurge) was used in addition to the digital pressure controller to apply underpressure lower than −800 mbar and overpressure higher than +1,000 mbar. FluidFM Rapid Prototyping probes with a microchannel height of 800 nm were obtained from Cytosurge. A 400-nm wide triangular aperture was custom-milled by a focused ion beam near the apex of the pyramidal tips and imaged by scanning electron microscopy as previously described[3]. The FluidFM probes were plasma treated for 1 min (Plasma Cleaner PDG-32G, Harrick Plasma) and coated overnight with SL2 Sigmacote (Sigma-Aldrich) in a vacuum desiccator[3].

## Optical microscopy setup

The FluidFM scan head was mounted on an inverted AxioObserver microscope equipped with a temperature-controlled incubation chamber (Zeiss), and coupled to a spinning disc confocal microscope (Visitron) with a Yokogawa CSU-W1 confocal unit and an EMCCD camera system (Andor). Phase-contrast and fluorescence images were acquired using ×10 and ×40 (0.6 numerical aperture, NA) objectives and a ×2 lens switcher using VisiView software (Visitron). Microscopy images were analysed using the AxioVision and ImageJ softwares.

## Enhanced Smart-seq2

The overall workflow follows the same steps as Smart-seq2 (ref. [25]), but several modifications were introduced. RNA, Live-seq samples or single cells were transferred into a PCR tube with 4.2 μl of lysis buffer, which contains 1 μl of dNTP (10 mM each), 1 μl of biotinylated-oligo-dT30VN oligo (10 μM), 2 μl of 0.2% Triton-X-100, 0.1 μl of Recombinant ribonuclease inhibitor (40 U μl$^{-1}$, no. 2313, Takara) and 0.1 μl ERCC RNA Spike-In Mix ($10^7$ dilution, no. 4456740, ThermoFisher). After briefly

spinning, the PCR tubes/plates were heated at 72 °C for 3 min and cooled down on ice for 1 min. Then, 5.2 μl of reverse transcription mix (1.29 μl of $H_2O$, 0.06 μl of $MgCl_2$ (1 M), 2 μl of Betaine (5 M), 0.08 μl of biotinylated-TSO oligo (100 μM) (or other TSOs as listed in Supplementary Fig. 1c), 2 μl of Maxima H Mimus RT buffer (5×), 0.25 μl of recombinant ribonuclease inhibitor and 0.1 μl of Maxima H Minus Reverse Transcriptase (200 U μl$^{-1}$, EP0751, ThermoFisher)) were added. The reverse transcription step was performed in a thermal cycler using the following program: (1) 42 °C for 90 min; (2) 50 °C for 2 min; (3) 42 °C for 2 min, go to step (2) for four cycles; (4) 85 °C for 5 min and (5) end. The reverse transcription products were directly used for PCR by adding 15 μl of PCR mixture, which contained 12.5 μl of KAPA HiFi HotStart ReadyMix (2×, 07958935001, KAPA), 0.25 μl of Biotinylated-ISPCR oligo (10 μM) and 2.25 μl of $H_2O$. The PCR program involves the following steps: (1) 98 °C for 3 min; (2) 98 °C for 20 s; (3) 67 °C for 15 s; (4) 72 °C for 6 min, go to step (2) for *x* cycles (for Live-seq, *x* = 24; for scRNA-seq, *x* = 18); (5) 72 °C for 5 min and (6) end. The PCR products were purified twice with AMPure XP beads (A63882, Beckman Coulter) according to the manufacturer's instructions with 0.6 volume of beads per 1 volume of PCR product. The concentration and profile of the purified DNA were measured by Qubit double-stranded DNA HS Assay Kit (no. Q32854, ThermoFisher) and fragment analyser (Advanced Analytical), respectively. Then 1 ng of DNA was used for tagmentation in the following reaction: 1 μl TAPS buffer (50 mM TAPS-NaOH, pH 8.3, 25 mM $MgCl_2$), 0.5 μl of Tn5 (made in-house, 12.5 μM), $H_2O$ to 5 μl in total. The tagmentation reaction was assembled on ice, initiated by placing the mix on a thermal cycler at 55 °C for 8 min after which it was kept on ice. Then 1.5 μl of SDS (0.2%) was added to inactivate the Tn5 enzyme. PCR reagent mixture was added directly to the tagmented DNA, which contained 1.5 μl of dNTP (10 mM each), Nextera XT index primers (or other compatible indexing primers, with a final concentration of 300 nM for each primer), 10 μl of KAPA HiFi Fidelity buffer (5×), 1 μl of KAPA HiFi DNA Polymerase (1 U μl$^{-1}$, KK2101, KAPA) and $H_2O$ to 50 μl in total. Then, the DNA was amplified using the following program: (1) 72 °C for 3 min; (2) 98 °C for 30 s; (3) 98 °C for 10 s; (4) 63 °C for 30 s; (5) 72 °C for 1 min, go to step (3) for ten cycles; (6) 72 °C for 5 min and (7) end. The PCR products were purified twice with AMPure XP beads (A63882, Beckman Coulter) according to the manufacturer's instructions with 0.6 volume of beads per 1 volume of PCR product. The concentration and profile of the purified DNA were measured using a Qubit dsDNA HS Assay Kit and fragment analyser, respectively, after which the libraries could be sent for sequencing using Illumina sequencers. We thereby noted that the 'cDNA' yield from the negative control (0 pg input RNA) (Extended Data Fig. 1b) was mainly derived from oligo sequences, because sequencing of the respective library revealed mainly reads with a very high A/T content, which poorly aligned to the mouse/human genome (Extended Data Fig. 1e, f).

## Cytoplasmic biopsies

**Cell samples and reagents.** For extraction experiments, the cells were seeded and cultured for at least 18 h onto 50-mm tissue-culture-treated low μ-dishes (Ibidi). Shortly before the experiment, the culture medium was replaced with 5 ml of $CO_2$-independent growth medium supplemented with 10% FBS, 1× penicillin/streptomycin solution and 2 mM glutamine. For the extraction of LPS-stimulated RAW cells, the culture medium was replaced with 5 ml of $CO_2$-independent growth medium supplemented with 10% FBS, 1× penicillin/streptomycin solution, 2 mM glutamine and 100 ng ml$^{-1}$ LPS, and the cells were extracted between 4 and 5 h after the addition of LPS. For the extraction of differentiating ASPCs, induction medium (ASPC adipogenic differentiation section) was replaced shortly before the experiment with 5 ml of $CO_2$-independent growth medium supplemented with 10% FBS, 1× penicillin/streptomycin solution, 2 mM glutamine, 1 μM dexamethasone, 0.5 mM 3-isobutyl-1-methylxanthine, 1 μM rosiglitazone and 167 nM insulin. Sampling buffer for preloading was prepared by supplementing a 0.2%

solution of Triton-X 100 in nuclease-free water with 2 U µl$^{-1}$ recombinant RNase inhibitors (Clonetech). Lysis buffer for extract transfer was prepared as that used in the enhanced Smart-seq2 protocol. Whereas cells were maintained at 37 °C for time-lapse microscopy before and after extractions, the extraction procedures were all performed at room temperature.

**Extraction process.** The probe reservoir was loaded with 10 µl of mineral oil (Sigma-Aldrich) and a pressure of Δ + 1,000 mbar was applied to flow the oil into the microchannel. Once the probe microchannel was filled, the probe was shortly immersed in nuclease-free water, and then kept in air with the residual water carefully blotted off the probe holder with a kimwipe tissue. A 1.0-µl drop of sampling buffer was deposited onto an AG480F AmpliGrid (LTF Labortechnik). The cantilever was introduced into the drop using the micrometre screws to displace the atomic force microscope. Once the cantilever was located inside the drop, underpressure (−800 mbar) was applied for the suction of roughly 0.5 pl of sampling buffer into the probe. The cantilever was then withdrawn from the drop using the micrometre screws.

Next, the preloaded probe was immersed in the cell sample experimental medium, the cell to be extracted was visualized by light microscopy and the tip of the FluidFM probe was placed above the cytoplasm of the selected cell. The tip of the probe was then inserted into the cytoplasm through a forward force spectroscopy routine driven by the $Z$-piezo. The probe was then maintained inside the cell at constant force. Although lower forces may be sufficient, a force setpoint of 500 nN was used to ensure the full insertion of the probe aperture into all the cell types. Underpressure larger than −800 mbar was applied to aspirate the cellular content into the probe, whereby the harvested cytoplasmic fluid immediately mixed with the preloaded sampling buffer. The pressure-assisted flow of the intracellular content into the FluidFM probe was interrupted by switching the pressure back to zero. We collected cytoplasmic extracts of 1.1 pl on average, ranging from 0.1 up to 4.4 pl. The probe was then retracted out of the cell, shortly immersed in nuclease-free water and then kept in air with the residual water carefully blotted off the probe holder with a kimwipe tissue.

A 1.0-µl drop of lysis buffer was deposited onto an AG480F AmpliGrid (LTF Labortechnik). The cantilever was introduced into the drop, and overpressure (more than 1,000 mbar) was applied to release the extract. The microchannel was then rinsed three times by suction and release of lysis buffer into the probe. The 1.0-µl drop was then pipetted into a PCR tube containing an additional 3.2 µl of lysis buffer, and the solution was briefly centrifuged and stored at −80 °C until further processing.

All steps were monitored in real time by optical microscopy in brightfield. The entire procedure took roughly 15 min per sample, with around 5 min for loading the sampling buffer and approaching a selected cell, 5 min to extract the cytoplasmic biopsy and 5 min to transfer the extract into the lysis buffer and then to the PCR tube. We performed 43 experiments, collecting 10 to 20 biopsies per experiment.

**Alternated human/mouse sampling.** For the assessment of cross-contamination between samples that were extracted sequentially with the same probe, alternated human/mouse cell sampling was conducted, with HeLa and IBA cells seeded on separate dishes. The cells were extracted alternatively from one or the other cell type as described above, using the same FluidFM probe.

**Sequential sampling of individual RAW cells.** For the sequential sampling of the same RAW cell, up to 24 cells were extracted as described above, with all the cells monitored within one vision field. The first extractions were performed within a 1 h time window, 0.5 to 1.5 h before the addition of LPS. The cells were then incubated at 37 °C under time-lapse monitoring with intervals of 5 min. After 30 min, LPS was added to the dish for stimulation and the cells were further monitored

at 5 min intervals for another 30 min. The time-lapse intervals were then increased to 30 min, and the cells were further monitored for 3.5 h. The temperature was then switched back to room temperature, and the same cells were extracted a second time, in a time interval of 4 to 5 h after the addition of LPS.

**Sequential sampling of individual ASPC cells.** For the sequential sampling of ASPCs, barcoded ASPCs (ASPC genetic barcoding section) were mixed with ASPCs at a 1:1,000 ratio and the cells were then seeded to a final density of 200,000 cells per cm$^2$ into a well with a growth area of 0.22 cm² (two-well insert, Ibidi). The cells were grown in the insert for 2 days. The insert was then removed and the growth medium exchanged for $CO_2$-independent medium. The entire culture area was first imaged at ×10 magnification, in brightfield and at 488 nm, and the images were assembled to create a map of the GFP-expressing, barcoded ASPCs in the entire culture area. Selected GFP-expressing ASPCs were then extracted for a first time, during a time window of 3 to 4 h. Following extraction, the medium was exchanged for induction medium (ASPC adipogenic differentiation), and the cells were incubated in a 5% $CO_2$ humidified atmosphere at 37 °C. After 2 days, the medium was exchanged for $CO_2$-independent medium supplemented with 1 µM dexamethasone, 0.5 mM 3-isobutyl-1-methylxanthine, 1 µM rosiglitazone and 167 nM insulin. The entire culture area was imaged again at ×10 magnification, in brightfield and at 488 nm, and the images assembled to create the map. The cells that had been extracted were relocalized from their respective position in the created map, and were extracted a second time. After the extractions, the medium was exchanged for the maintenance medium (ASPC adipogenic differentiation section), and the cells were incubated for 2 days in a 5% $CO_2$ humidified atmosphere at 37 °C. The medium was then exchanged for complete culture medium, and the cells were incubated for another 3 to 5 days with the medium exchanged every 2 days. Lipid staining was then performed using 5 µg ml$^{-1}$ BODIPY 558/568 (Invitrogen) for 20 min. Nucleus staining with 5 µg ml$^{-1}$ 4,6-diamidino-2-phenylindole (DAPI) was performed at the same time. The entire culture area was then imaged to create the map, before imaging individual barcoded cells in brightfield, at 405, 488 and 640 nm.

**Molecular recording of RAW cells.** For the molecular recording to predict a cell's downstream phenotype, RAW cells within one vision field were extracted as described above, within a 1 h time window. The cells were then monitored at 37 °C under time-lapse imaging with intervals of 5 min. After 30 min, LPS was added to the dish for stimulation (0.5 to 1.5 h after extraction), and the cells were further monitored with 5 min intervals for another 30 min. The time-lapse intervals were then increased to 30 min, and the cells were further monitored for at least 8 h.

### LPS stimulation of RAW cells

LPS (no. L4391-1MG) was prepared in PBS at 100 µg ml$^{-1}$ as a stock solution. For stimulation of RAW cells, LPS solution was added to the 5 ml of $CO_2$-independent medium to reach a final concentration of 100 ng ml$^{-1}$.

### ASPC genetic barcoding

The genetic barcoding of ASPCs was performed using the pLARRY system[51]. LARRY Barcode Library v.1 was a gift from F. Camargo (Addgene no. 140024). The DNA was prepared directly by collecting cells directly from a Luria–Bertani agar plate rather than from a liquid culture to maximally preserve the overall library complexity, which was further confirmed by next-generation sequencing.

Barcode-bearing lentivirus was produced in 293T cells using the third-generation lentivirus packing system. The virus-containing medium was collected 48 h posttransfection. ASPCs were transduced by the virus-containing medium and fresh medium with Polybrene

(final concentration at 10 μg ml$^{-1}$), followed by centrifugation at 1,500$g$ for 30 min after which cells were returned to the cell incubator. The medium was exchanged 12 h later and cells were maintained at a density lower than 80% confluency.

### ASPC adipogenic differentiation

To differentiate ASPCs into adipocytes, confluent cells were exposed to adipogenic induction medium (complete culture medium supplemented with 1 μM dexamethasone, 0.5 mM 3-isobutyl-1-methylxanthine, 167 nM insulin and 1 μM rosiglitazone (DMIR), all from Sigma). After 2 days, the induction medium was removed and the maintenance medium (the complete culture medium supplemented with 167 nM insulin) added. At day 4, the maintenance medium was removed and complete culture medium was added. Lipid droplets were stained using BODIPY 558/568 (Invitrogen) at 5 μg ml$^{-1}$ for 20 min after which cells were subjected to imaging.

### Recovery of genetic barcodes from Live-seq-sampled ASPCs

The barcodes were retrieved from the cDNA of Live-seq samples. The barcode region was enriched from 1 ng cDNA by PCR using the primers BC-FOR1 (5′-ctgagcaaagaccccaacgagaa-3′) and BC-REV1 primers (5′-gctggcaactagaaggcacag-3′) and using the following program: (1) 98 °C for 30 s; (2) 98 °C for 10 s; (3) 63 °C for 30 s; (4) 72 °C for 1 min, go to step (2) for 15 cycles; (5) 72 °C for 5 min and (6) end. Then 1 μl of PCR product was subjected to a second round of PCR with For-MEDS-A (5′-tcgtcggcagcgtcagatgtgtataagagacagcgttgctaggagagaccatatg-3′) and Rev-MEDS-B primers (5′-gtctcgtgggctcggagatgtgtataagagacaggtcgacaccagtctcattcagc-3′), and using the following program: (1) 98 °C for 30 s; (2) 98 °C for 10 s; (3) 63 °C for 30 s; (4) 72 °C for 1 min, go to step (2) for 15 cycles; (5) 72 °C for 5 min and (6) end. The result PCR product was purified with AMPure XP beads (Beckman) using a 2.5 volume of beads per 1 volume of PCR product ratio, and eluted with 20 μl of nuclease-free water. Then 10 μl of the purified product was indexed using the Nextera index kit (Illumina), using the following program: (1) 98 °C for 30 s; (2) 98 °C for 10 s; (3) 63 °C for 30 s; (4) 72 °C for 1 min, go to step (2) for four cycles; (5) 72 °C for 5 min and (6) end. The resulting product was purified with AMPure XP beads (Beckman) using a 2.5 volume of beads per 1 volume of PCR product ratio, and eluted with 20 μl of nuclease-free water. The concentration was then measured using a Qubit dsDNA HS Assay Kit and subjected to sequencing in the Gene Expression Core facility at the EPFL.

For the data analysis, we created a .fasta file representing the vector sequence, replacing the barcodes by N strings. All samples were then aligned on this vector using STAR v.2.7.9a (ref. [52]) with default parameters and these two extra options: '--outFilterMultimapNmax 1' (for removing several mapped reads) and '--outFilterMismatchNmax 2 --outFilterScoreMinOverLread 0 --outFilterMatchNminOverLread 0 --outFilterMatchNmin 0' (for authorizing alignment on Ns, and a maximum of two mismatches). We then extracted the sequences from the aligned .bam file at the position of the two barcodes and counted their occurrence to find the most prevalent ones. A Hamming distance of the barcodes between cells was then calculated using the function StrDist of the DescTools package[53] in R using the parameter (method = 'hamming', mismatch = 1, gap = 1).

### Determination of the extracted volumes

For each extraction, the volume of preloaded sampling buffer and the volume of cytoplasmic extract mixed with the sampling buffer were measured on brightfield images using the AxioVision software. The area occupied by the aqueous solutions confined in the cantilever was multiplied by the channel height of 0.8 μm, and the volume of the hollow pyramidal tip (90 fl) was added. To determine the volume of extracted cytoplasmic fluid, the volume of preloaded sampling buffer (FluidFM probe before extraction) was subtracted to the volume of mixed sampling buffer and extract (FluidFM probe after extraction).

### Determination of the cell volumes

To quantify the whole-cell volumes, IBA and ASPC cells were dissociated by trypsinization and RAW264.7 cells with a cell scraper. The dissociated cells were then imaged in brightfield with the ×40 objective and the ×2 lens switcher, and the diameter of the rounded cells was measured from the micrographs using the AxioVision software. Three diameters were measured and averaged for each cell ($n$ = 277 for ASPC and $n$ = 500 for IBA and RAW). The volumes were calculated using the formula for a sphere. For the longitudinal measurements of RAW cell volumes, the areas of selected semi-adherent cells were measured on brightfield images acquired with the ×40 objective and the ×2 lens switcher at several time points before and after extraction, and cell volumes were calculated assuming a spherical cell shape.

### Time-lapse monitoring of mCherry expression

For live imaging of RAW-G9 cells expressing mCherry, the Ibidi dish was covered and the temperature was maintained at 37 °C. Time-lapse images were acquired at 5 min intervals for 1 h, then at 30 min intervals overnight. For each time point, two sequential frames were acquired, in brightfield and for mCherry (561 nm laser and 609/54 nm emission filter), using the ×40 objective and the ×2 lens switcher. At the end of the time-lapse recording, 2 μl of tricolour calibration beads (Invitrogen MultiSpeck Multispectral Fluorescence Microscopy Standards Kit, Life Technologies Europe B.V.) was added to the sample and at least 30 individual beads were imaged with the 561 laser. For image quantification, all the cells that moved out of view or focus, died or overlapped with other cells were excluded from analysis in each time-lapse frame. Cell boundaries were all manually defined in Fiji, and background intensities were measured for each time point and subtracted from all intensity values. We analysed 77 extracted cells, 122 LPS-stimulated cells that were not extracted and 23 cells that were not extracted and not stimulated with LPS. The fluorescence intensity of the calibration beads with subtracted background intensity was measured using Fiji, and the average of the 30 bead intensities was used to normalize the fluorescence measurements of the cells acquired in different experiments. The area under a curve was calculated from 3 to 7.5 h post-LPS treatment. A two-sided Wilcoxon rank-sum test was used to examine the differences between conditions.

### Cell viability after extraction

To evaluate the postextraction viability of ASPC ($n$ = 33) and IBA ($n$ = 37) cells, the cells were stained between 2 and 4 h after extraction using a LIVE/DEAD Cell Imaging Kit 488/570 (Invitrogen), and following the manufacturer's protocol. To evaluate the postextraction viability of RAW264.7 cells, the extracted cells ($n$ = 72) were monitored by time-lapse microscopy during around 10 h at 30 min intervals. Cells were evaluated as dead or alive on the basis of their morphology, movements and expression of mCherry in response to LPS stimulation. The postextraction viability was assessed for 42 cells extracted once before stimulation, 30 cells extracted once after LPS stimulation and ten cells extracted twice, before and after stimulation.

The volumes that were extracted ranged between 0.7 and 3.3 pl (mean 1.4 pl) for ASPCs, between 0.4 and 2.9 pl (mean 1.0 pl) for IBA cells and between 0.2 and 3.5 pl (mean 1.1 pl) for RAW cells. The viability of all the cell types was calculated as an absolute value without normalization.

### scRNA-seq

Cells were trypsinized and dissociated into a single-cell suspension and then kept on ice for all the downstream processing. After passing through a 40-μm cell constrainer and DAPI staining, live singlet cells were sorted into 96-well PCR plates with 4.2 μl of lysis buffer (mentioned above). At least three wells without cells were preserved as negative controls. The plate was quickly spun and further processed using the Smart-seq2 method[25].

To perform scRNA-seq on cells post-Live-seq extraction, cells were first cultured in a dish containing a silicone micro-insert (Ibidi, no. 80409) at a density of around 20 cells per well. The insert was then removed just before Live-seq sampling and all the cells were subjected to Live-seq extraction as described above, during a 1 h time window. To extract all the cells in this time, the extracted cytoplasm was not preserved. Then 1 and 4 h after the middle of the extraction time window (that is, $1 \pm 0.5$ and $4 \pm 0.5$ h postextraction), the cells, along with the control cells not extracted, were collected on ice and single cells were picked using a serial dilution approach. The downstream processing followed a similar workflow as the Smart-seq2 method.

## Cell cycle analyses

The cell cycle reporter vector pCSII-EF-miRFP709-hCdt(1/100) was a kind gift from V. Verkhusha (Addgene plasmid no. 80007; http://n2t.net/addgene:80007; RRID Addgene_80007). Lentivirus carrying this vector was transduced into RAW-G9 cells after which miRFP709-positive cells were sorted to enrich for transduced cells. The cells were seeded on an Ibidi dish and monitored for miRFP709 (640 nm laser and 700/75 nm emission filter) and in brightfield for 24 h at 60 min intervals with the ×40 objective and the 2× lens switcher. The experimental growth medium was then exchanged, LPS was added at a final concentration of 100 ng ml$^{-1}$, and the cells were monitored for another 24 h at 60 min intervals in brightfield, for miRFP709 and for mCherry (561 nm laser and 609/54 nm emission filter). Fluorescence intensities of the cells were measured as described above (Time-lapse monitoring of mCherry expression) for both fluorescent reporters. The mCherry intensities were further normalized using the beads calibration.

## *Nfkbia* reporter analyses

The mouse *Nfkbia* promotor (+1,606 to −121) fragment was obtained by PCR (forward primer ttcaaaattttatcgatcagtgaaatccagaccagccgggcctac, reverse primer ggctgtgcggggctgagcgg) from mouse genomic DNA. The TagBFP CDS fragment was obtained by PCR (forward primer tgcagcctgcacccgctcagccccgcacagccACCatgagcgagctgattaaggagaac, reverse primer tgtaatccagaggttgattgtcgacgcggccgcttaattaagcttgtgccccagtttgc) from a TagBFP-bearing plasmid. A linearized lentivirus vector devoid of the EF1a promoter was obtained by PCR (forward primer gcggccgcgtcgacaatcaac, reverse primer cccggctggtctggatttcactgatcgataaaattttgaattttgtaatttgttttttgtaattc) using the pLV-vector as template (kindly provided by J. Han). All three fragments were then assembled using a Gibson Assembly Master Mix (NEB) according to the manufacturer's instructions.

The lentivirus production from 293T and transduction into RAW-G9 cells were performed in the same way as implemented for ASPC barcoding, as described above. The resulting cells were then seeded into 96-well plates with each well containing no more than one cell to generate single clones. The single clones were then screened for their capacity to respond to LPS as well as the brightness and the broad distribution of BFP fluorescence at basal level. Cells from a selected clone were seeded in a 50-mm low-wall dish (Ibidi) and grown for 1 day at normal cell culture conditions. One culture area was then imaged with a ×40 objective and the ×2 lens switcher, at 37 °C, in brightfield and at 405 nm (BFP) and 561 nm (mCherry), at 30 min intervals for 10 h. The cells were first monitored for 1.3 h to record their basal level of *Nfkbia*-BFP and *Tnf*-mCherry, after which LPS was added to the sample and the cell were imaged for another 8.7 h to monitor their response to the stimulation. Fluorescence intensities of the cells were measured as described above (Time-lapse monitoring of mCherry expression section) for both fluorescent reporters. The mCherry and BFP intensities were further normalized using the beads calibration. The basal BFP intensity was averaged from the three first time-frames, acquired before LPS stimulation.

## Live-seq and scRNA-seq data analysis

**Dataset description.** There were ten IBA cells and HeLa cells processed through Live-seq sampling in an alternating fashion. In addition, 588

Live-seq samples were prepared across five experimental replicates, with each of them containing both single and sequential sampling events. All the data from these 588 cells were used for the analyses presented in Figs. 2–5. Then 554 cells were processed using conventional scRNA-seq as part of three experimental replicates and the data linked to these cells are presented in Figs. 2–5. Finally, to evaluate the potential molecular perturbation of cytoplasmic sampling, scRNA-seq data of IBA cells 1 h (49 cells) and 4 h (43 cells) postextraction were generated using cells (70 cells) that were not subjected to such sampling as control.

**Alignment and feature counting.** Libraries were sequenced in either 75 single-read or 2 × 75 paired-end format using Nextera indexes on Illumina Nextseq500 or Hiseq4000 sequencers at the Gene Expression Core facility (at EPFL). Basecalls are performed using bcl2fastq (v.2.19.1). To keep consistency, only read 1 with 75 bp was used for further analysis. The reads were aligned to the human (hg19/GRCh38) or mouse (mm10/GRCm38) genomes using STAR (v.2.6.1c)[52] with default settings and filtered for uniquely mapped reads. Then, the number of reads per feature (gene) was counted using HTseq (v.0.10.0)[54] with parameter 'htseq-count -s no -m union -f bam' and the gene annotation of Ensembl release 87 supplemented with ERCC, EGFP and mCherry features was used. The counts of all samples were merged into a single gene expression matrix, with genes in rows and samples in columns (Gene Expression Omnibus (GEO) accession number GSE141064, processed data).

To evaluate the cross-sample contamination, we sampled human and mouse cells alternatively. Reads were aligned to the mixed human:mouse reference genome (hg38 and mm10) using STAR and the number of reads per feature was counted by HTseq using the same settings as mentioned above. Digital gene expression matrices were generated for each species. We analysed the downstream data using R (v.3.5.0), plots generated using the R package ggplot2 (v.3.2.1).

**Cell and gene filtering.** ERCCs were used for technical evaluation by testing the correlation with the expected number of spike-in RNA molecules, and not included for further analysis. The top 20 genes found in negative controls (samples were merged) are probably due to misalignment/sequencing errors of the oligos (Supplementary Table 5), and were thus also removed from downstream analyses. Ribosomal protein-coding genes were removed as they confound the downstream differential gene expression analysis, consistent with previous findings[55]. The downstream analysis followed the procedures of the Seurat R package (v.3.0). Samples showing low quality were filtered out, with quality cut-offs being: (1) the number of genes fewer than 1,000, (2) the mitochondrial read ratio more than >30% or (3) the uniquely mapped rate fewer than <30%. Then, the data were normalized to the total expression, multiplied by a scale factor of 10,000, after which a pseudo-count was added and the data were log transformed.

**Feature selection, dimensionality, reduction clustering and others.** The top 500 highly variable genes were chosen on the basis of the variance stabilizing transformation (vst) result (function, FindVariableFeatures(object, selection.method = 'vst')). Scaling was applied to all the genes using the function 'ScaleData'. The scaled data of the 500 highly variable genes were used for principal component analysis (PCA) analysis. The first ten principal components were chosen for further clustering and *t*-SNE analysis on the basis of the ranking of the percentage of variance explained by each principal component (ElbowPlot function). Seurat-embedded graph-based clustering was applied. In both Live-seq and scRNA-seq data, postdifferentiated ASPCs were in early differentiation stage rendering their classification by treatment difficult to capture when in a dataset driven by more distinct cell types and states. Furthermore, the clustering of these cells was prone to be biased by batch as only one of the two batches contained both pre- and postdifferentiation ASPCs. Therefore, to capture ASPCs' heterogeneity, ASPCs were clustered separately for both scRNA-seq and Live-seq

data (default Seurat pipeline and data scaled for nCount and nGene), and the clustering found on the whole datasets, at resolution 0.2, was adapted. For data visualization, individual cells were projected on the basis of the ten principal component scores onto a two-dimensional map using *t*-SNE[56].

To evaluate the effect of varying sequencing depth on the clustering of Live-seq data, we down-sampled the raw counts to the desired number. The down-sampled matrices were analysed in the same way as described above. The clusters were consistent with the original analysis, indicating that the clusters were not driven by sequencing depth.

The DE genes were analysed as described below (Comparison of DE genes between Live-seq and scRNA-seq section). The pseudo-genes were filtered out[57]. For the Gene Ontology analysis, the top first 100 DE genes, ordered by logFC, were loaded into the EnrichR package (v.3.0)[32] to determine gene enrichment among the biological processes of Gene Ontology and predict the cell types using the Mouse Gene Atlas database. We carried out Gene Set Enrichment Analysis in R using the ClusterProfiler package[58] v.3.14.3 and msigdbr v.7.1.1 using default settings with all the gene expression changes. Both wikipathways-20200810-gmt-Mus_musculus.gmt and the Gene Ontology terms obtained from the R database org.Mm.eg.db (v.3.10.0) were used as reference.

The scores of cell cycle phases were calculated using the Seurat function 'CellCycleScoring' on the basis of canonical markers[59].

To integrate Live-seq and scRNA-seq data, the CCA and mutual nearest neighbours (MNNs)-based approaches embedded in Seurat v3 were used. Specifically, the top 500 highly variable genes of both datasets were chosen for PCA analysis independently. The first ten principal components of each were used to identify the anchors and for data integration. As another level of control, data stemming from cells belonging to each cluster in both Live-seq and scRNA-seq analyses were collapsed into a 'bulk' RNA-seq dataset after which the Pearson's correlation between the bulk Live-seq and the bulk scRNA-seq datasets was determined. To evaluate the effect of varying sequencing depth on data integration, we down-sampled the raw counts to the desired number. The down-sampled matrices were then analysed in the same way as described above.

The cells sampled with Live-seq or scRNA-seq from the two cell types, RAW cells and ASPCs, were analysed on a cell type-by-cell-type basis following Seurat's pipeline. For the ASPCs of Live-seq and scRNA-seq, the only batch containing both DMIR-treated and non-treated cells Live-seq the scRNA-seq were selected. The cells were filtered using the same filtering as previously described, then the data normalized and scaled for nCounts, nFeatures and batch (if more than one batch) were used to compute the PCA. Finally, the first ten principal components were used to compute the *t*-SNE with perplexity set to 10.

**Comparison of DE genes between Live-seq and scRNA-seq.** Differential gene expression analysis was conducted using edgeR[60] v.3.34.0. Only genes expressed in at least 5% of the data with a minimal count of 2 (filterByExpr() min.count = 2, min.prop = 0.05) and in at least 15% of the cells of one of the categories with a minimal count of 2 in the scRNA-seq data were considered. The dispersions and negative binomial glm were fitted (glmQLFit and glmQLFTest functions) for the model roughly 0 + categories + batch, the categories being the different groups to be tested. The quasi-likelihood *F*-tests were calculated for each group, with the null hypothesis being that there is no difference between the mean in the group of interest and the average over all the remaining categories (clusters). The *P* values were corrected using the Benjamini–Hochberg procedure. The genes with an FDR < 0.05, absolute logFC > 1, and expressed in at least 15% of one of the two groups were considered DE.

To compare scRNA-seq and Live-seq data, the differential expression analyses were performed separately for the two techniques. For each result, the log fold-expression changes of each gene were plotted against each other. The DE genes were highlighted (Bonferroni adjusted *P* value < 0.05 and an absolute logFC > 1). A linear model was fitted using lm() over (1) all the genes, (2) only the genes detected as DE in the scRNA-seq datasets or (3) the genes defined as DE in both the Live-seq and scRNA-seq datasets. Similar analyses were also applied per cell type basis, that is in RAW cells before and after LPS treatment and ASPC before and after differentiation.

Differential expression analysis per cell type was performed using edgeR package by contrasting between non-treated versus treated cells in the same fashion as described above. The genes with an FDR < 0.05, absolute logFC > 1 and expressed in at least 15% of one of the two groups of cells with at least two counts were considered DE.

For the two cell types mentioned above, cells of the scRNA-seq datasets were randomly selected to match the number of cells of the Live-seq data. The count matrices were then down-sampled per cell to have the same density distribution of the number of features as the corresponding Live-seq data. More precisely, the cells were ordered by the number of features in the scRNA-seq and Live-seq datasets and paired on the basis of this metric. The number of sampled reads (with replacement) of the scRNA-seq cells were defined so that the down-sampled data reached a similar number of features (absolute difference below 5) compared to its paired Live-seq cell. The down-sampled data were then (up-)sampled with replacement to match the library size of their paired Live-seq cell. Differential expression analysis was then performed as described above.

**Live-seq and live-cell imaging integration.** Among the 40 cells that were both subjected to Live-seq and tracked for LPS-induced *Tnf*-mCherry fluorescence, 17 of them passed the quality control as mentioned in the Live-seq section. For each cell in the time-course, we calculated the intercept (that is, basal expression) and slope (that is, extent of response) using a linear model between the time after LPS treatment and the natural log of the *Tnf*-mCherry fluorescence intensity. As we were mainly interested in the initial response and as the curve was linear from the first three to 7.5 h (Extended Data Fig. 9b), we only used values from this time window. To rank genes, we then constructed a linear model that predicts the intercept or slope of the mCherry response using the expression of a gene measured by Live-seq before treatment with LPS. Given the limited number of cells and thus to increase the statistical power of the models, only the 500 most variable genes from both the RAW-G9 Live-seq and scRNA-seq data were used, but removing genes with a dispersion lower than 0.1. The genes were then ranked on the basis of their respective $R^2$ values. We used two tests to assess the significance of each gene: an *F*-test on the overall model as implemented using R's lm function, and a bootstrapping approach in which we randomly sampled cells with replacement to calculate an empirical *P* value. *P* values were corrected for multiple testing using the Benjamini–Hochberg procedure as implemented using the R's p.adjust function.

**Trajectory inference**

To infer the trajectory from conventional scRNA-seq data, the dynverse R package (v.0.1.2)[10] with multiple wrapped trajectory inference methods was used[61–65]. The parameters shown below were used for the 'answer_questions' function- 'multiple_disconnected = NULL, expect_topology = NULL, expected_topology = NULL, n_cells = 554, n_features = 1,000, memory = '100GB', docker = TRUE'. The most suggested methods were chosen on the basis of the guidelines provided by dynverse. In addition, we also applied the Monocle DDRTree method[66] given its widespread use. All these methods were run in the docker with default parameters. For the RNA velocity analysis, annotated spliced, unspliced and spanning reads in the measured cells were generated in a single loom file using the command line 'velocyto run_smartseq2 -d1' function. This also generates an HDF5 file containing detailed molecular mapping information that was used for the analysis model on the basis of gene structure. Genes were filtered to have a min.max.cluster.average of at least 5, 1 and 0.5 for the exonic, intronic and spanning read

expression matrices, respectively. Three different algorithms were used following the velocyto (v.0.6) pipeline (http://pklab.med.harvard.edu/velocyto/notebooks/R/chromaffin2.nb.html): (1) cell kNN pooling with the gamma fit based on extreme quantiles, with function 'gene.relative. velocity.estimates(deltaT = 1, kCells = 5, fit.quantile = 0.05)', (2) relative gamma fit without cell kNN smoothing, with function 'gene.relative. velocity.estimates(deltaT = 1, deltaT2 = 1, kCells = 1, fit.quantile = fit. quantile)' and (3) velocity estimate based on gene structure. Here, the unfiltered intronic and spanning expression matrix was used to include more genes for the genome-wide model fit.

## Analysis of the most variable genes from primary macrophage scRNA-seq data

We retrieved single-cell expression data of five macrophage subsets from the GEO database (GSE117081)[67]. We calculated for each gene a standardized variance by modelling the relationship between the observed mean expression and variance using local polynomial regression, as implemented in the Seurat (v.3.1.4) FindVariableFeatures function. Genes of the KEGG NF-κB signalling pathway downstream TLR4 receptor were highlighted.

## Bioethics

All mouse experiments were conducted in strict accordance with the Swiss law, and all experiments were approved by the ethics commission of the state veterinary office (licence number VD 3406, valid from 14 October 2018 to 14 January 2022). Mice were housed under specific pathogen-free conditions at 20–24 °C with 45–65% humidity and a 12 light/12 dark cycle.

## Reporting summary

Further information on research design is available in the Nature Research Reporting Summary linked to this paper.

## Data availability

All Live-seq and scRNA-seq data are available in the GEO with accession number GSE141064.

## Code availability

The codes used to analyse the Live-seq and scRNA-seq data are incorporated into the Methods sections listed above, shared at https://github. com/DeplanckeLab/Live-seq and archived at https://doi.org/10.5281/zenodo.6611232.

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

**Acknowledgements** We thank J. Russeil and P.C. Schwalie for support and J. Pezoldt, G. La Manno and J. Kribelbauer for reviewing the manuscript and data analysis. We thank I.D.C. Fraser (National Institutes of Health) for kindly providing the RAW-G9 cells. We thank the Gene Expression (GECF, EPFL) and Flow Cytometry (FCCF, EPFL) core facilities for technical support. This work was supported by a Swiss National Science Foundation grant (no. 310030_182655), a Precision Health & Related Technologies grant (no. PHRT-502) and institutional funding (EPFL) to B.D., a National Key R&D Program of China (grant no. 2021YFA0911100), Marie Skłodowska-Curie fellowship, EPFL Fellows (grant no. 665667) to W.C. and by a grant from the Volkswagen foundation (Initiative 'Life'), a European Research Council Advanced grant (no. 883077) and institutional funding (ETH Zurich) to J.A.V.

**Author contributions** W.C., O.G.-G., J.A.V. and B.D. designed the study and wrote the manuscript. W.C. and O.G.-G. performed experiments and data analysis with the support of C.G.G., R.D., P.Y.R., W.S., M.Z. V.G., A.K. and T.Z. All authors read and approved the final manuscript.

**Competing interests** The authors declare no competing interests.

**Additional information**
**Correspondence and requests for materials** should be addressed to Julia A. Vorholt or Bart Deplancke.

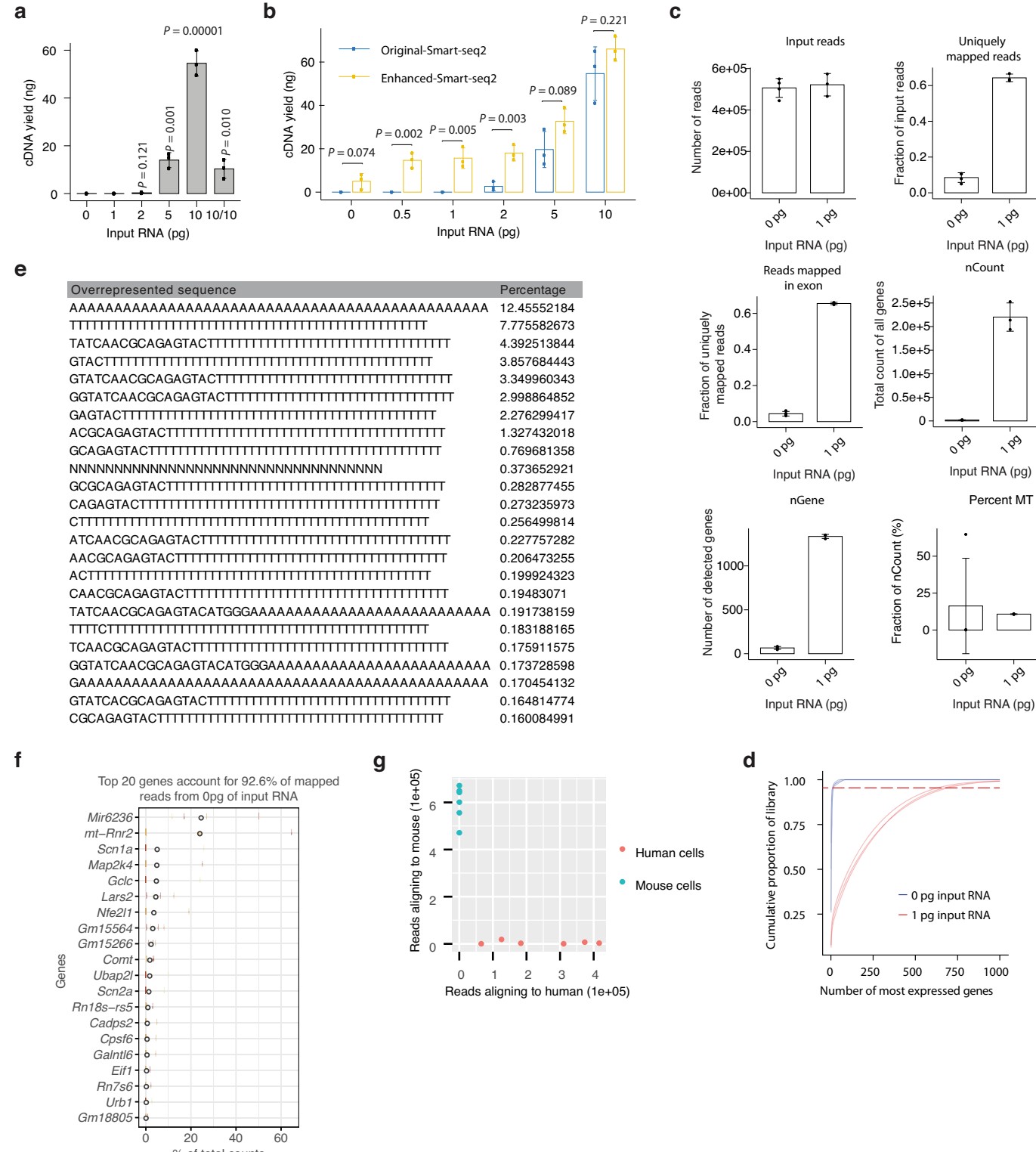

**Extended Data Fig. 1 |** See next page for caption.

**Extended Data Fig. 1. | Supporting data of Fig. 1.** (**a**) cDNA yields of different amounts of total RNA input using Smart-seq2. 10/10:10% of the material reverse transcribed from 10 pg total RNA was used for PCR amplification. N = 3 replicates. $P$ values determined by two-sided $t$-tests comparing each condition to 0 pg input RNA. (**b**) Enhanced Smart-seq2 is more sensitive than the original Smart-seq2 in the low input range (0.5–2 pg). N = 3 replicates for all conditions. $P$ values determined by two-sided $t$-tests. Two and three distinct experiments were performed in a) and b), respectively, yielding consistent results. (**c**) Quality control of enhanced Smart-seq2 based on the parameters listed above each panel, comparing negative control (0 pg, N = 4 replicates) to IBA cell RNA (1 pg, N = 3 replicates). nGene: number of detected genes. nCount: total count of all genes. Percent MT: percentage of counts from mitochondrial genes. (**d**) Cumulative proportion of each library (y axis) assigned to the top-expressed genes (x axis). The top 20 genes absorb around 95% of all the reads in the negative control (N = 4 replicates), while the ~700 top genes take that same portion of reads in samples with 1 pg input RNA (N = 3 replicates). The dashed line indicates the 95% proportion. (**e**) Overview of the sequences overrepresented in the negative control, mostly from the oligo-dT and TSO. (**f**) Proportion of reads mapped to each gene in negative control samples. The top 20 genes account for more than 90% of all reads. (**g**) Human (HeLa) and mouse (IBA) cells were sampled alternatively with the same probe. The number of reads mapped to the human and mouse genomes were determined for each sample to assess potential cross-sample contamination. Error bars represent the mean +/− SD.

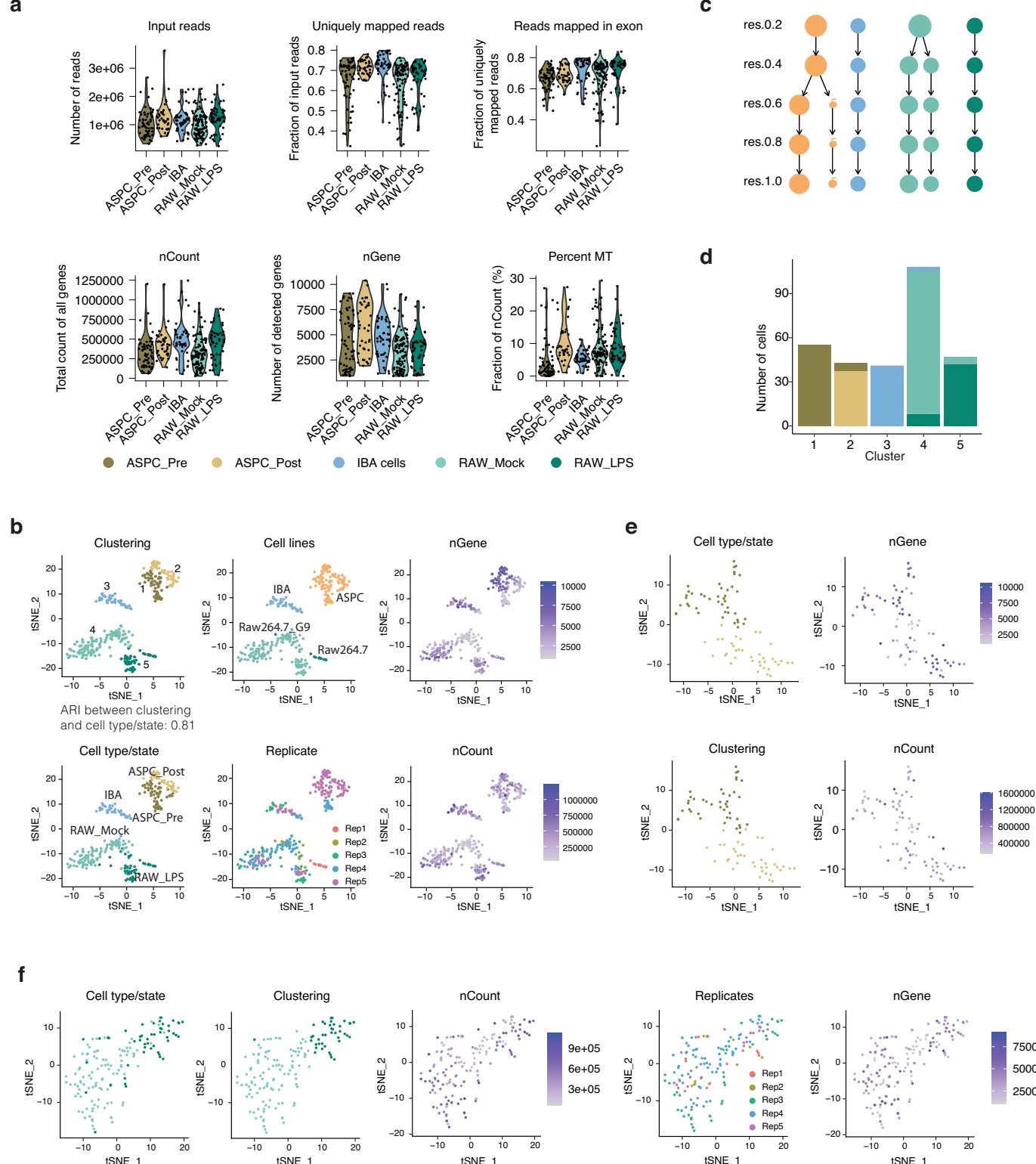

**Extended Data Fig. 2** | See next page for caption.

**Extended Data Fig. 2 | Quality control of Live-seq data, relative to Fig. 2.**
(**a**) Number of input reads (input reads), the rate of reads uniquely mapped to the genome (uniquely mapped reads), the fraction of reads mapped to exons (Reads mapped in exon), total counts of all genes (nCount), number of detected genes (nGene) and the percentage of counts from mitochondrial genes (percent MT) are shown per cell type/state for Live-seq samples/libraries passing the quality control. N = 5 replicates, a total of 294 cells. (**b**) tSNE-based visualization of clusters, cell types/states, cell lines, replicates, number of genes (nGene), and number of counts (nCount) of the Live-seq data. The Adjusted Rand Index (ARI) between the clustering and cell type/state classification is indicated. (**c**) Clustering tree of the Seurat-based clustering results of the Live-seq data. It visualizes the relationship between clustering at increasing resolutions (top to bottom). The size of the circles represents the number of cells in that cluster, while the opacity of the arrows shows the proportion of the cells passing from one cluster to another at a different resolution. Note that the ASPCs do not split by treatment due to batch effect. The clustering was therefore independently adapted for the clustered ASPCs to correctly capture their state difference (see Methods). (**d**) Barplot showing the overlap in number of cells between the clustering (x-axis) and the ground truth, i.e., cell type/state, displayed in (b). (**e-f**) tSNE-based visualization of cell type/state, nGene, Clustering, nCount and batch for (e) ASPCs and (f) RAW cells. The ASPCs only contain one batch.

**a**

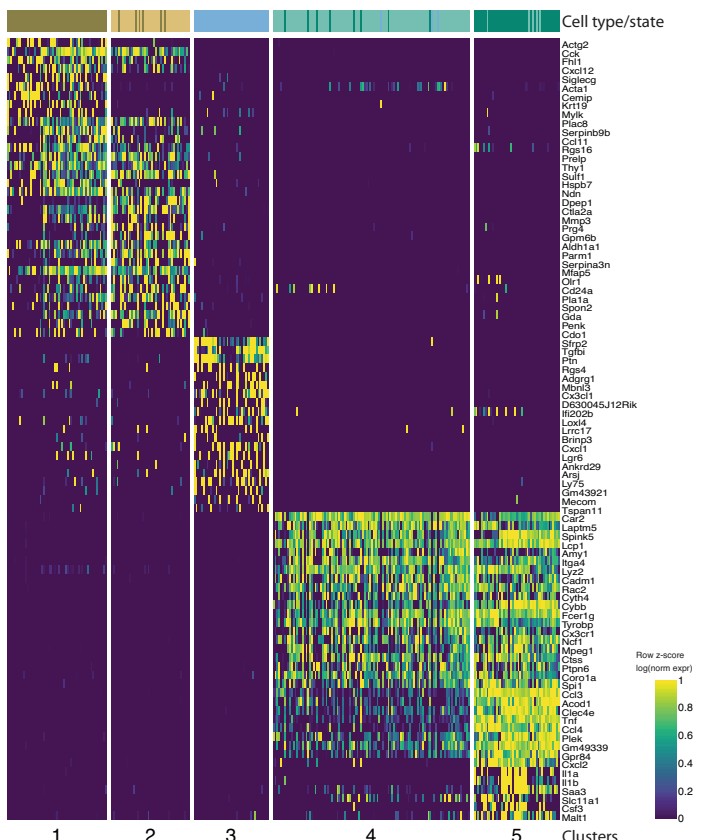

**c**

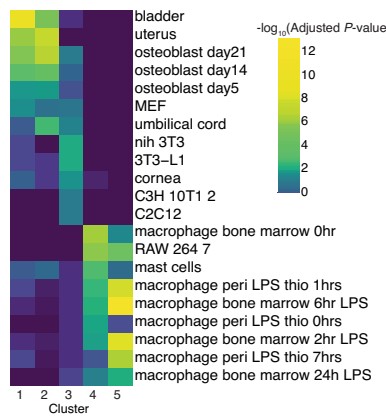

**b**

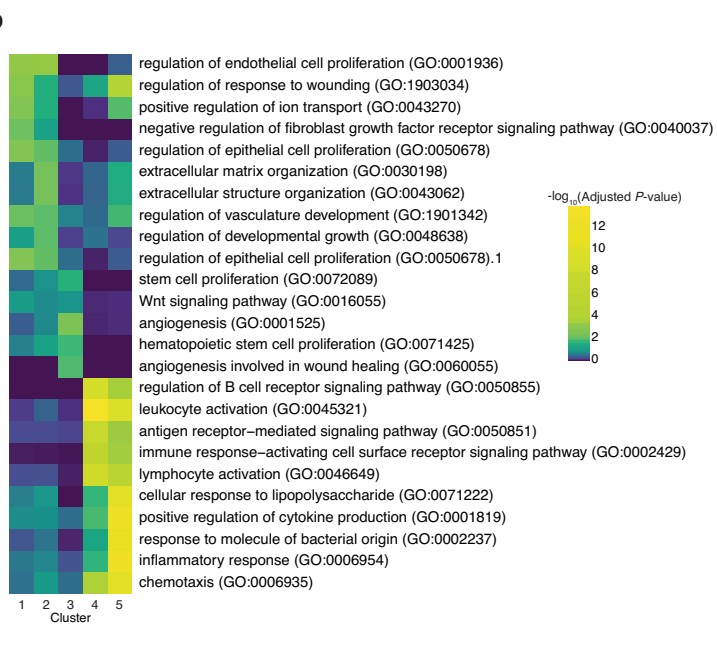

● ASPC_Pre  ● ASPC_Post  ● IBA cells  ● RAW_Mock  ● RAW_LPS

**Extended Data Fig. 3 | Evaluation of the cell identity as discovered by Live-seq, relative to Fig. 2.** (**a**) Heatmap showing the top 20 differentially expressed genes of each cluster of the Live-seq data. (**b**) GO term enrichment of each cluster using the top 100 marker genes. (**c**) Mouse gene atlas-based prediction of cell type/state of each cluster using the top 100 marker genes.

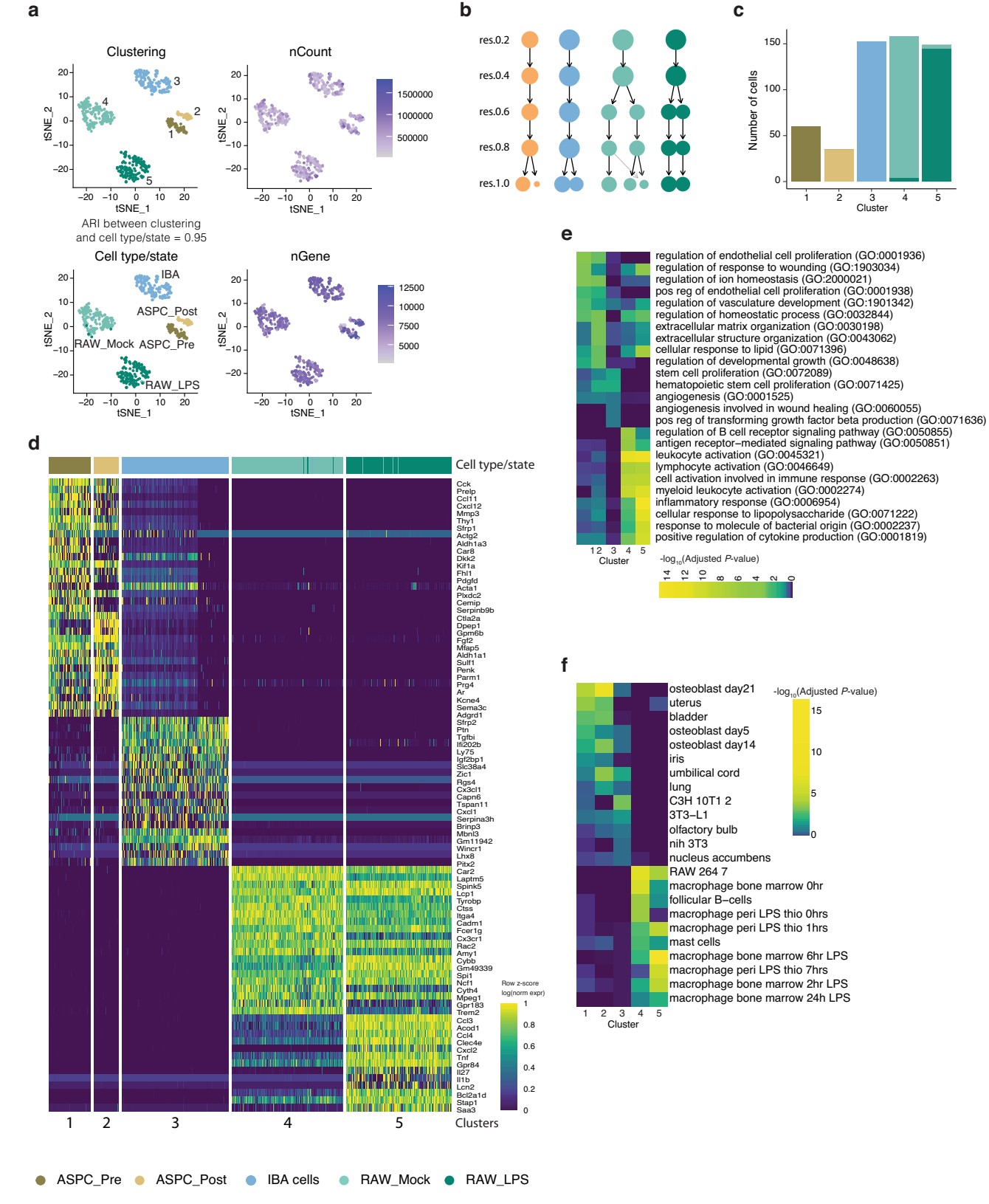

**Extended Data Fig. 4** | See next page for caption.

**Extended Data Fig. 4 | Quality control of scRNA-seq data, relative to Fig. 2.**
(**a**) tSNE-based visualization of clusters, cell types/states, number of counts (nCount) and number of genes (nGene) of the scRNA-seq data. The Adjusted Rand Index (ARI) between the clustering and cell type/state classification is indicated. (**b**) Clustering tree of the Seurat-based clustering results of the scRNA-seq data. It visualizes the relationship between clustering at increasing resolutions (top to bottom). The size of the circles represents the number of cells in that cluster, while the opacity of the arrows shows the proportion of the cells passing from one cluster to another at a different resolution. Note that the ASPCs do not split by treatment due to batch effect. The clustering was therefore independently adapted for the clustered ASPCs to correctly capture their state difference (see Methods). (**c**) Barplot showing the overlap in number of cells between the clustering (x-axis) and the ground truth, i.e. cell type/state, displayed in (a). (**d**) Heatmap showing the top differentially expressed genes stratified according to the five scRNA-seq clusters. (**e**) GO term enrichment analysis of the five scRNA-seq clusters using the top 100 differentially expressed genes. (**f**) Mouse gene atlas-based prediction of cell type/state of each cluster using the top 100 marker genes.

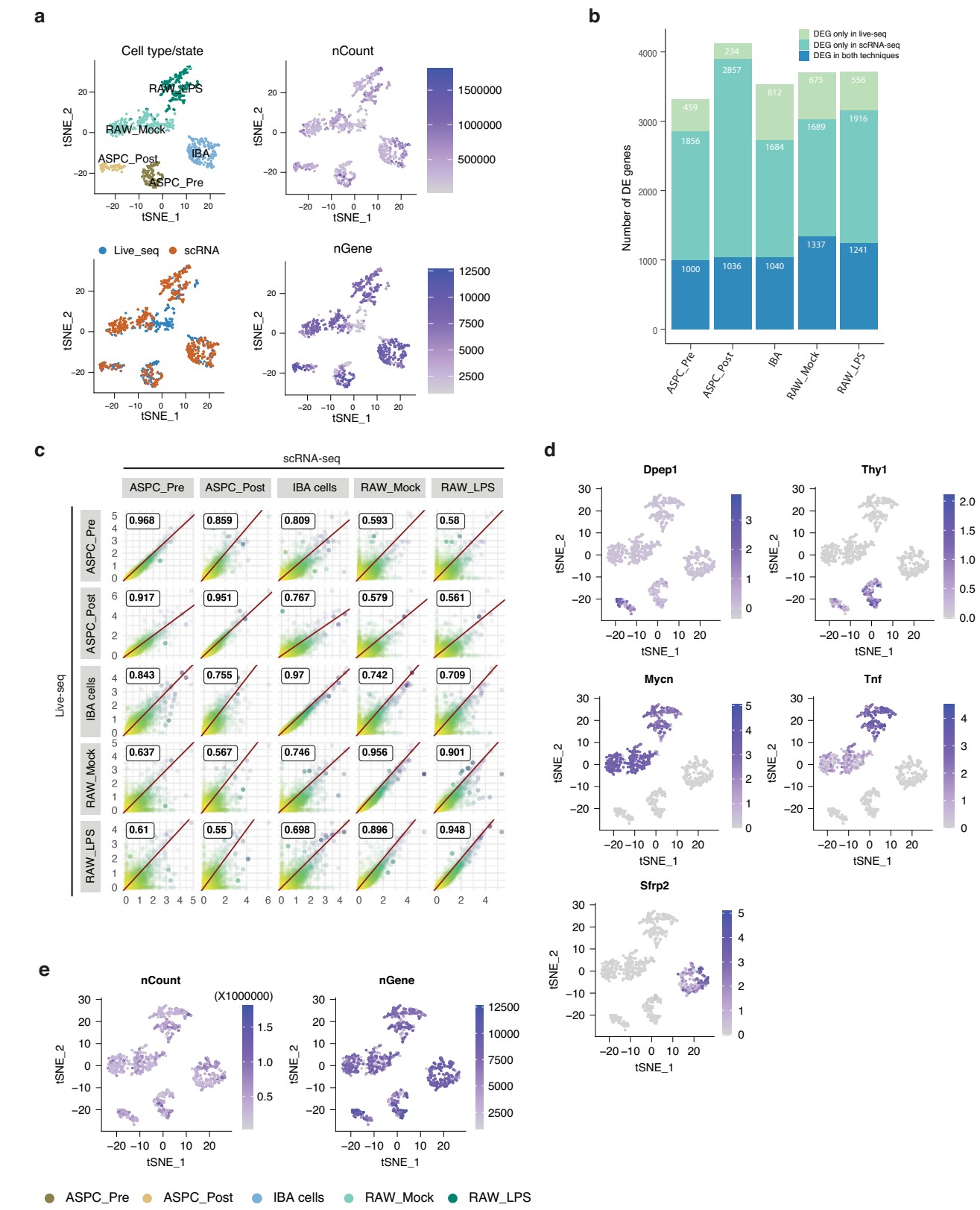

**Extended Data Fig. 5 | Comparison of Live-seq and scRNA-seq data, relative to Fig. 2.** (**a**) tSNE-based visualization of integrated scRNA-seq and Live-seq data according to cell type or treatment, approach, number of detected genes (nGenes), and total counts of all genes (nCount) without batch correction. (**b**) Barplot displaying the number of overlapping genes identified as differentially expressed in the Live-seq and scRNA-seq data for each cell type and state versus the rest. (**c**) The correlation between simulated bulk data (i.e. based on the aggregation of each scRNA-seq or Live-seq expression profile into bulk-like data) across cell type and state (treatment), which shows that the Live-seq and scRNA-seq data are highly correlated. The Pearson correlation value is shown inside each of the subpanels. (**d**) tSNE-based visualization of marker gene expression on integrated scRNA-seq and Live-seq data. (**e**) Additional quality controls relative to Fig. 2f: visualization of Live-seq and scRNA-seq data after anchor-based data integration (Methods) according to the number of detected genes and total gene expression count.

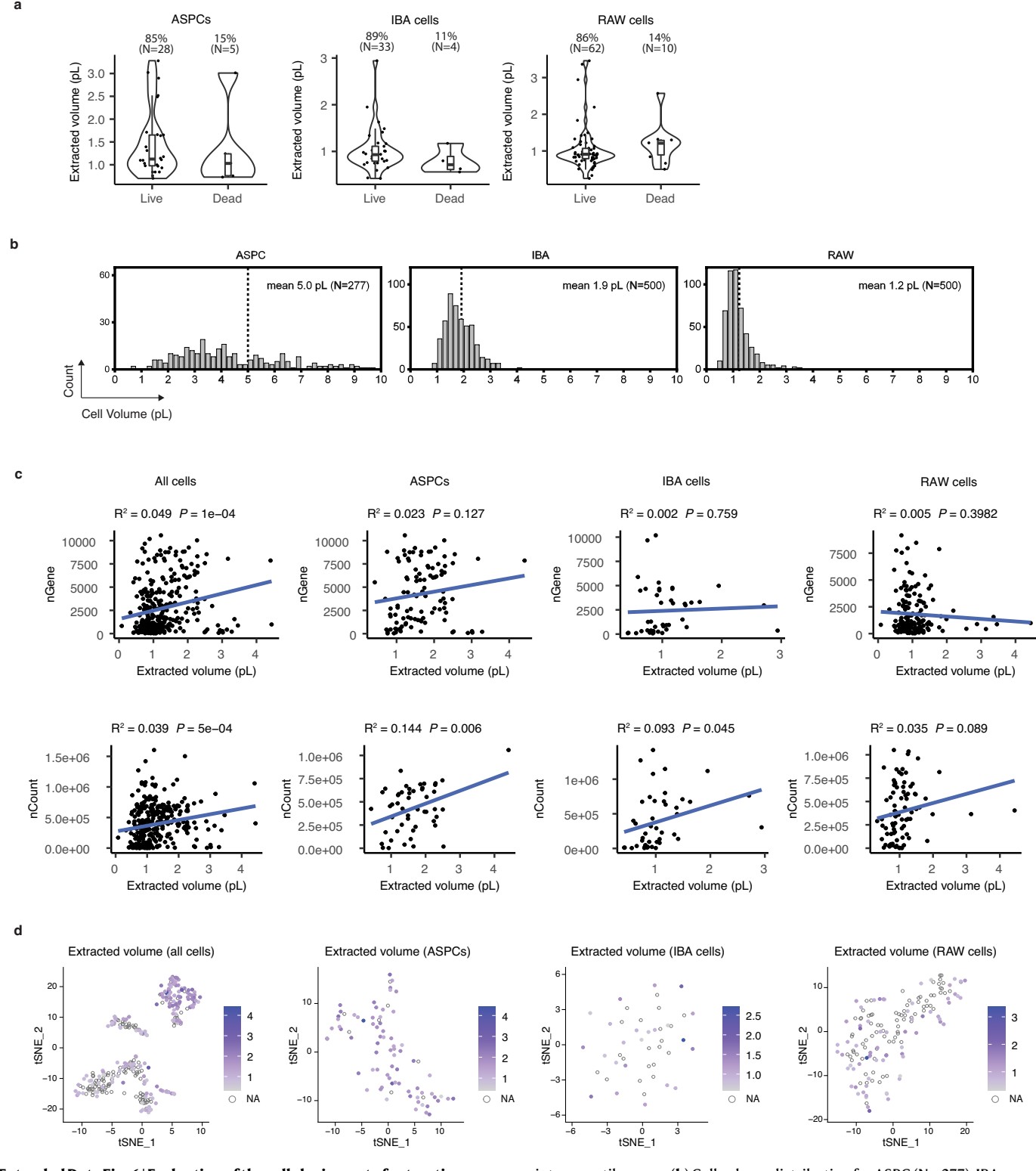

**Extended Data Fig. 6 | Evaluation of the cellular impact of extracting cytoplasm by Live-seq, relative to Fig. 3. (a)** Post-extraction viability as a function of the Live-seq-sampled volumes for ASPC (N = 33), IBA (N = 37) and RAW cells (N = 72). $P$ = 0.44, 0.20, 0.18 for ASPC, IBA and RAW cells (two-sided Wilcoxon rank-sum test), respectively. The lower, centre and upper bounds of the box correspond to the first, second and third quartiles, respectively. The whiskers extend to the largest and smallest value no further than 1.5 inter-quartile range. **(b)** Cell volume distribution for ASPC (N = 277), IBA (N = 500), and RAW (N = 500) cell populations. **(c)** The correlations ($R^2$ of linear regression and P value (two-sided F-test)) between extracted cytoplasmic volume and either the number of detected genes (nGene) or total counts (nCount) for each indicated category are shown. **(d)** tSNE plots of RAW, IBA, and ASPC cells colored by extracted cytoplasmic volume. NA: data not available. Relative to Fig. 2b and Extended Data Fig. 2e, f.

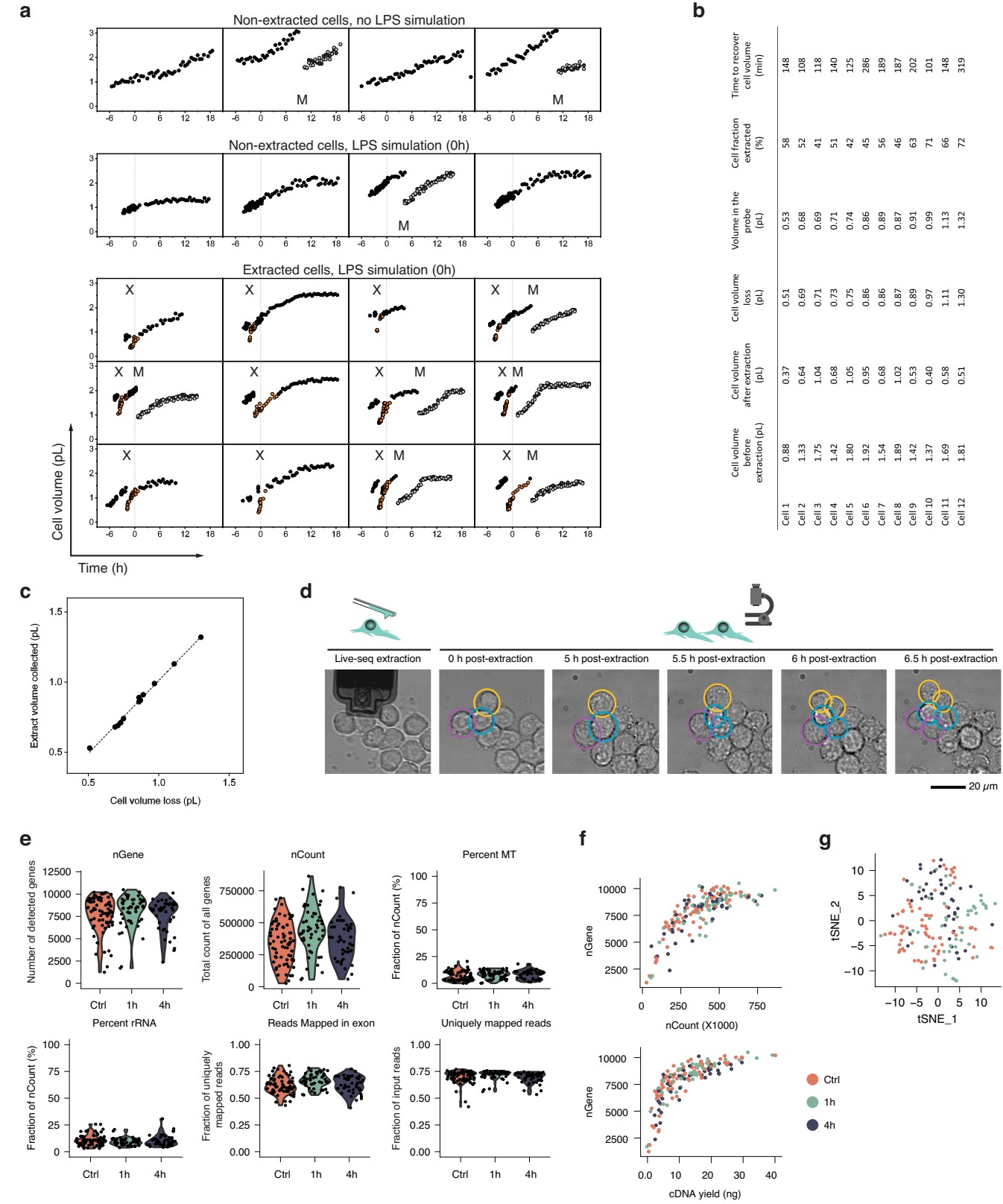

**Extended Data Fig. 7** | See next page for caption.

**Extended Data Fig. 7 | Live-seq preserves a cell's growth, with additional controls relative to Fig. 3.** (**a**) Longitudinal measurements of RAW cell volumes. Cells were exposed to LPS at time 0 (vertical line). "X" indicates an extraction and "M" mitosis. After mitosis, both daughter cells were monitored (light and dark grey points). Volumes measured during cell recovery from Live-seq extraction are labeled as orange points. (**b**) The volume changes, extracted volumes, and recovery times of RAW cells, calculated based on the profiles in (a). (**c**) Correlation between the measured cell volume loss shown in (a) and the measured cell extract volumes that were measured in the FluidFM probe (see "Determination of the extracted volumes" section in Methods) ($R^2$ = 0.99). (**d**) Representative time-lapse images showing the division of RAW cells post-Live-seq extraction. Three dividing cells are outlined in colored circles. The cell in the yellow circle was subjected to cytoplasmic extraction, while those in purple and blue were not. 49 extracted cells and 272 non-extracted cells were observed independently; 17 extracted and 54 non-extracted cells divided during 8 h of time-lapse imaging. (**e**) Quality control of the scRNA-seq data of the control IBA cells, as well as IBA cells 1 h and 4 h post Live-seq extraction, respectively. Relative to Fig. 3. (**f**) Plotted correlations between the number of detected genes on the one hand and respectively the total count of all genes (nCount, upper panel) and the cDNA yield (lower panel), were indistinguishable between the respective cell categories. (**g**) A tSNE projection of control IBA cell scRNA-seq (Smart-seq2) data (Ctrl, 70 cells) as well as 1 h (49 cells) and 4 h (43 cells) post Live-seq extraction scRNA-seq (Smart-seq2) data does not reveal clearly distinct clusters based on the top 500 most variable genes.

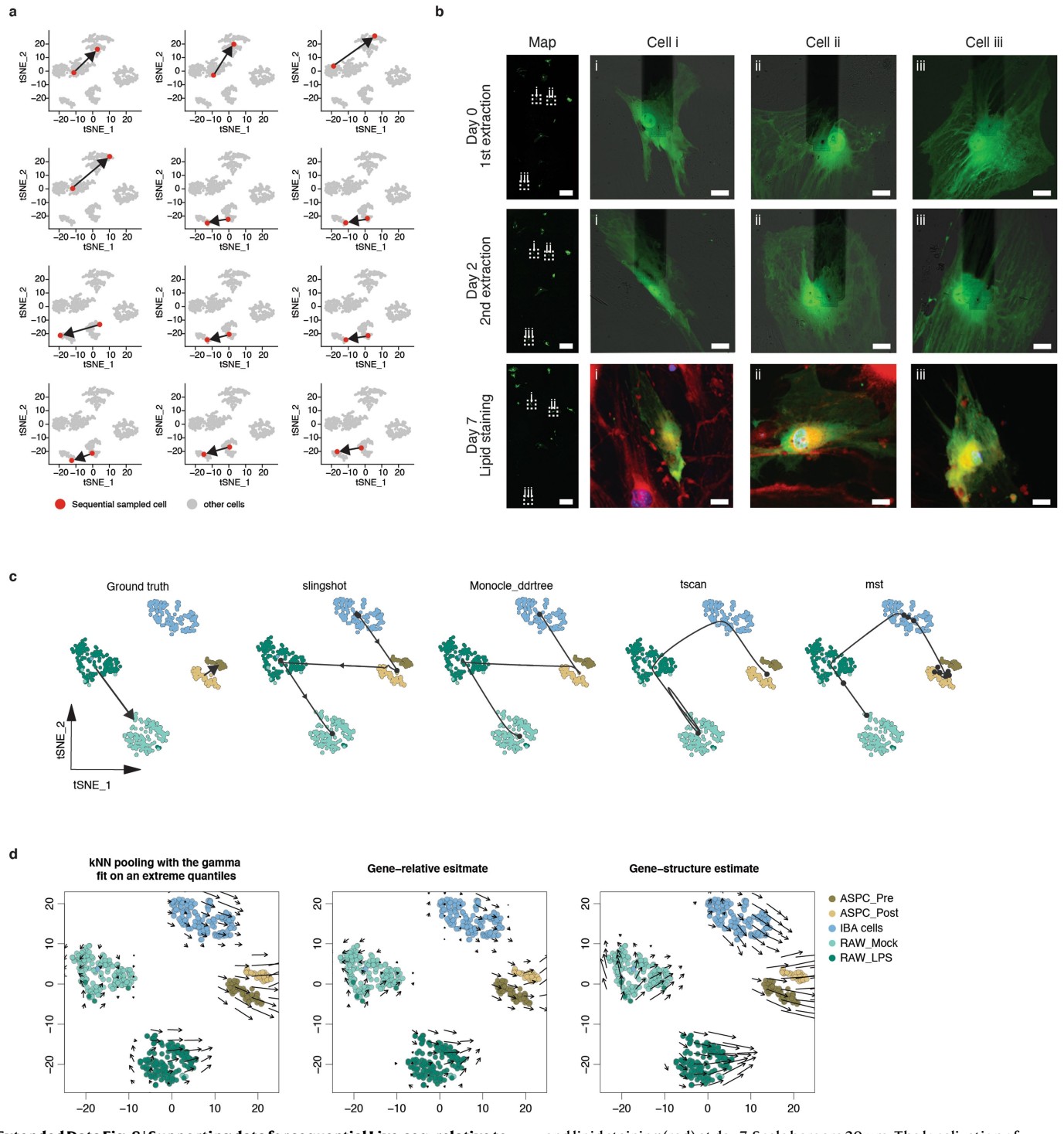

**Extended Data Fig. 8 | Supporting data for sequential Live-seq, relative to Fig. 4.** (**a**) Cell state transition trajectories as measured by sequential Live-seq. The two red dots in each panel represent the same cell. The arrow defines the respective cell state transition. (**b**) Sequential extraction of ASPCs. The left panels provide an overview of the entire cell culture area, enabling the localization of barcoded, GFP-expressing cells. Scale bars are 1000 μm. The right panels show three representative cells (out of 42 in 5 independent experiments) undergoing extraction at day 0 and day 2, and after nuclear (blue) and lipid staining (red) at day 7. Scale bars are 20 μm. The localization of each cell within the culture area (map) is indicated with a dashed square. (**c**) Trajectory predictions based on conventional scRNA-seq data using distinct approaches with default settings as contained in the dynverse package. (**d**) Trajectory prediction using the RNA velocity approach. Different strategies including "kNN pooling with gamma fit on extreme quantiles", "Gene-relative estimate", and "Gene-structure estimate" were tested.

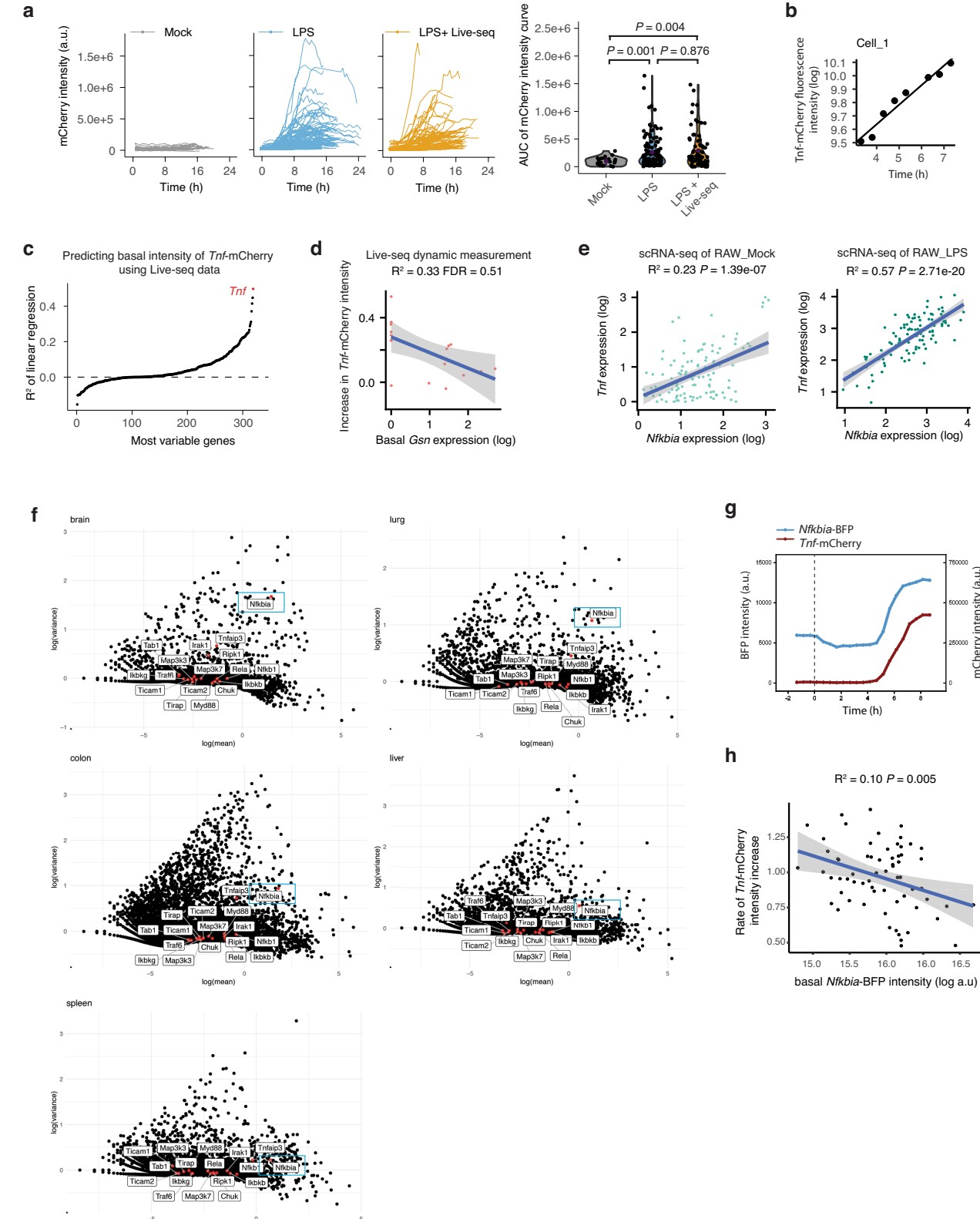

**Extended Data Fig. 9** | See next page for caption.

**Extended Data Fig. 9 | *Nfkbia* basal expression predicts LPS-induced response in RAW cells, relative to Fig. 5.** (**a**) (Left panels) Overlaid mCherry intensity profiles of individual mock-treated (N = 23), LPS-treated (N = 122), or Live-seq-sampled and then LPS-treated (N = 77) cells. (Right panel) Area Under the Curve (AUC) values for all profiles shown in the left panels. *P* values were determined by a two-sided Wilcoxon rank-sum test. (**b**) Relative to Fig. 5a, linear relationship between the time post-LPS treatment (within a 3–7.5h window) and the *Tnf*-mCherry fluorescence intensity (log transformed) in one cell (see Supplementary Fig. 7 for all other cells). (**c**) Similar to Fig. 5b, a linear regression model was used to predict the intercept (basal *Tnf*-mCherry intensity) calculated from data shown in (b and Supplementary Fig. 7) based on ground-state Live-seq-recorded expression data. The *Tnf* gene is highlighted. (**d**) Basal *Gsn* expression anticorrelates with the rate of *Tnf*-mCherry fluorescence intensity increase. $R^2$ and FDR values are listed. (**e**) Similar to Fig. 5d, but the expression correlation between *Nfkbia* and *Tnf* is shown separately for RAW_Mock and RAW_LPS cells. The $R^2$ and *P* (F test) values of the linear regression model are shown. (**f**) *Nfkbia* is among the most variably expressed LPS-NF-kB pathway genes in primary macrophage cell populations. The expression and the variance of all expressed genes are shown, with genes of the KEGG NF-κB signaling pathway (downstream of the TLR4 receptor) highlighted. (**g**) Representative *Tnf*-mCherry and *Nfkbia*-BFP profiles from a single cell. Similar to endogenous *Nfkbia*, an in-house engineered *Nfkbia*-BFP reporter is induced by LPS treatment, synchronously with the *Tnf*-mCherry reporter, acting as a proxy for *Nfkbia* expression. LPS was applied at time 0 h (vertical dashed line). (**h**) Validation of *Nfkbia* expression as a predictor of a macrophage's response to LPS (Methods). The basal *Nfkbia* level, here reflected by *Nfkbia*-BFP reporter intensity, negatively correlates with the rate of LPS-induced *Tnf*-mCherry intensity increase. Independent, biological replicate relative to Fig. 5e. $R^2 = 0.10$, $P = 0.005$, F-test; Pearson's r = −0.34, $P = 0.005$. The error band (d, e and h) represents the SD.

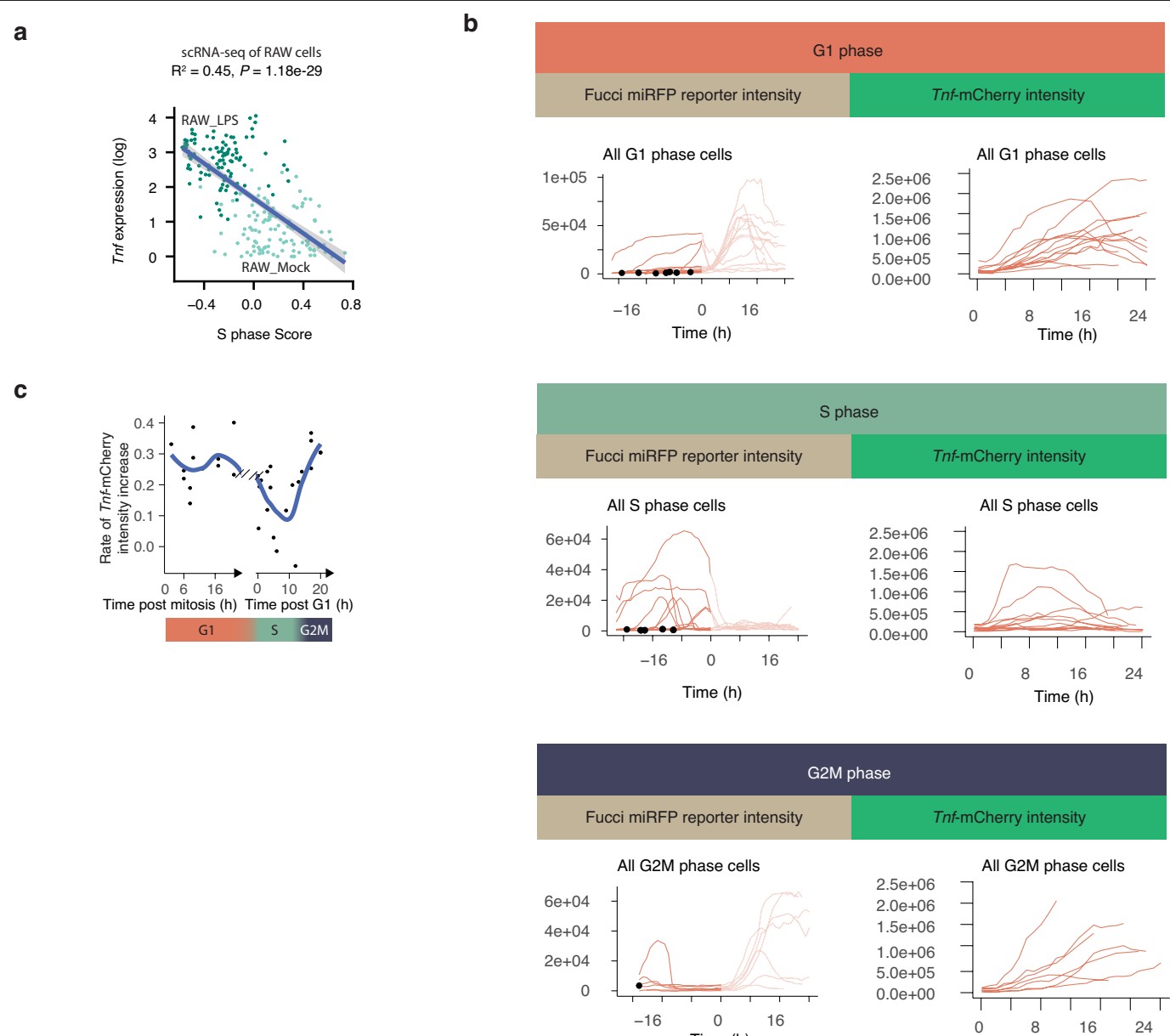

**Extended Data Fig. 10 | The LPS-induced response in RAW cells is weaker in S phase than those in other cell cycle phases, relative to Fig. 5.** (**a**) The S phase score anti-correlates with *Tnf* expression in conventional scRNA-seq data. The $R^2$ and $P$ (F test) values of the linear regression model are shown. (**b**) Cell cycle phases of individual cells were determined using the Fucci reporter miRFP-hCdt1. The Fucci reporter and LPS-induced *Tnf*-mCherry intensities of the cells assigned to a specific cell cycle phase are merged into one meta-plot, with the information of each cell shown in Supplementary Fig. 8. The time of LPS treatment is defined here as 0. Black dots indicate the mitosis time point. The curves of Fucci reporter after LPS treatment are not considered and are therefore shown in a lighter color. The G1/S boundary is inferred from the time point at which the Fucci reporter intensity drops. Cells that underwent mitosis but did not yet reach the G1/S boundary were annotated as G1 cells. Given the lack of a clearly discernable S/G2M boundary, cells were assigned to either the S or G2M phase based on the post G1/S boundary timing. (**c**) The rate of *Tnf*-mCherry fluorescence intensity increase (slope) between 3 to 7.5 h post-LPS treatment was calculated based on profiles in (b). The Fucci reporter was used to time cells based on the G1/S boundary. However, mitosis was used to specifically time G1 cells, as the latter did not yet reach the G1/S boundary (which can be detected using the Fucci reporter), rendering them more difficult to annotate.

# Reporting Summary

## Statistics

For all statistical analyses, confirm that the following items are present in the figure legend, table legend, main text, or Methods section.

| n/a | Confirmed | |
|---|---|---|
| ☐ | ☒ | The exact sample size (*n*) for each experimental group/condition, given as a discrete number and unit of measurement |
| ☐ | ☒ | A statement on whether measurements were taken from distinct samples or whether the same sample was measured repeatedly |
| ☐ | ☒ | The statistical test(s) used AND whether they are one- or two-sided *Only common tests should be described solely by name; describe more complex techniques in the Methods section.* |
| ☒ | ☐ | A description of all covariates tested |
| ☐ | ☒ | A description of any assumptions or corrections, such as tests of normality and adjustment for multiple comparisons |
| ☐ | ☒ | A full description of the statistical parameters including central tendency (e.g. means) or other basic estimates (e.g. regression coefficient) AND variation (e.g. standard deviation) or associated estimates of uncertainty (e.g. confidence intervals) |
| ☐ | ☒ | For null hypothesis testing, the test statistic (e.g. *F*, *t*, *r*) with confidence intervals, effect sizes, degrees of freedom and *P* value noted *Give P values as exact values whenever suitable.* |
| ☒ | ☐ | For Bayesian analysis, information on the choice of priors and Markov chain Monte Carlo settings |
| ☒ | ☐ | For hierarchical and complex designs, identification of the appropriate level for tests and full reporting of outcomes |
| ☐ | ☒ | Estimates of effect sizes (e.g. Cohen's *d*, Pearson's *r*), indicating how they were calculated |

*Our web collection on statistics for biologists contains articles on many of the points above.*

## Software and code

Policy information about availability of computer code

| Data collection | Basecalls performed on NextSeq 500/HiSeq 4000 result using bcl2fastq v2.19.1 |
|---|---|
| Data analysis | STAR (2.6.1c); HTseq (0.10.0); R (version 3.5.0); ggplot2 (version 3.2.1); Seurat R package (v3.0); EnrichR(3.0); ClusterProfiler(3.14.3);  msigdbr (7.1.1); org.Mm.eg.db (version 3.10.0); dynverse (0.1.2); velocyte(0.6); |

For manuscripts utilizing custom algorithms or software that are central to the research but not yet described in published literature, software must be made available to editors and reviewers. We strongly encourage code deposition in a community repository (e.g. GitHub). See the Nature Portfolio guidelines for submitting code & software for further information.

## Data

Policy information about availability of data

All manuscripts must include a data availability statement. This statement should provide the following information, where applicable:
- Accession codes, unique identifiers, or web links for publicly available datasets
- A description of any restrictions on data availability
- For clinical datasets or third party data, please ensure that the statement adheres to our policy

All Live-seq and scRNA-seq data are available in the Gene Expression Omnibus (GEO) with accession number GSE141064.

# Field-specific reporting

Please select the one below that is the best fit for your research. If you are not sure, read the appropriate sections before making your selection.

☒ Life sciences          ☐ Behavioural & social sciences          ☐ Ecological, evolutionary & environmental sciences

For a reference copy of the document with all sections, see nature.com/documents/nr-reporting-summary-flat.pdf

# Life sciences study design

All studies must disclose on these points even when the disclosure is negative.

| | |
|---|---|
| Sample size | There are 10 IBA cells and Hela cells processed through Live-seq sampling in alternative fashion. In addition, 588 Live-seq samples were prepared across five experimental replicates, with each of them containing both single and sequential sampling events. All the data from these 588 cells were used for the analyses presented in Figures 2-5. 554 cells were processed using conventional scRNA-seq as part of three experimental replicates and the data linked to these cells are presented in Figures 2-5. To evaluate the potential molecular perturbation of cytoplasmic sampling, scRNA-seq data of IBA cells 1 hour (49 cells) and 4 hours (43 cells) post extraction were generated, as well as of control cells that were not subjected to such sampling (70 cells) . To evaluate the post-extraction viability of ASPC (N=33) and IBA (N=37) cells, the cells were stained between 2 and 4 hours after extraction using a LIVE/DEAD® Cell Imaging Kit 488/570 (Invitrogen), and following the manufacturer's protocol. To evaluate the post-extraction viability of RAW264.7 cells, the extracted cells (N= 72) were monitored by time-lapse microscopy. We analysed 77 extracted cells, 122 LPS-stimulated cells that were not extracted, and 23 cells that were not extracted and not stimulated with LPS. Statistic tests were performed for all the analyses as indicated in the Figure legends and methods accordingly. While the sample size for Live-seq to couple initial transcriptome with subsequent LPS response is insufficient to explain the full variant at each single gene level, we have discussed the limitation of Live-seq at the current stage in the result and discussion section of the main text. |
| Data exclusions | No data is excluded, except for RNA-seq samples showing low quality were filtered out, with quality cutoffs being: i) the number of genes <1000, ii) the mitochondrial read ratio >30%, or iii) the uniquely mapped rate <30%. This is described in the methods. |
| Replication | Live-seq samples and conventional scRNA-seq were prepared across five and three replicates, respectively. Enhanced Smart-seq2 optimization was performed with three replicates. The calculation of cell volume is performed per cell base (N=277 for ASPC and N=500 for IBA and RAW). Time-lapse monitoring of mCherry expression was performed with three replicates. Cell viability after extraction was checked with more than 3 replicates. These replicates successfully show consistent results. |
| Randomization | No randomization was applied as only cell lines and no human or animal subjects were used. |
| Blinding | The imaging of lipid accumulation in ASPC was performed blindingly. Blinding is not necessary for other analysis as they are quantitative and no subjective interpretation is required. |

# Reporting for specific materials, systems and methods

We require information from authors about some types of materials, experimental systems and methods used in many studies. Here, indicate whether each material, system or method listed is relevant to your study. If you are not sure if a list item applies to your research, read the appropriate section before selecting a response.

## Materials & experimental systems

| n/a | Involved in the study |
|---|---|
| ☒ | ☐ Antibodies |
| ☐ | ☒ Eukaryotic cell lines |
| ☒ | ☐ Palaeontology and archaeology |
| ☐ | ☒ Animals and other organisms |
| ☒ | ☐ Human research participants |
| ☒ | ☐ Clinical data |
| ☒ | ☐ Dual use research of concern |

## Methods

| n/a | Involved in the study |
|---|---|
| ☒ | ☐ ChIP-seq |
| ☒ | ☐ Flow cytometry |
| ☒ | ☐ MRI-based neuroimaging |

# Eukaryotic cell lines

Policy information about cell lines

| | |
|---|---|
| Cell line source(s) | RAW264.7, 293T and Hela cells were obtained from ATCC. RAW264.7 cells with Tnf-mCherry reporter and relA-GFP fusion protein (RAW-G9 clone) were generated and kindly provided by Dr. Iain D.C. Fraser (NIH). The IBA cell line derived from the stromal vascular fraction (SVF) of interscapular brown adipose tissue of young male mice (C57BL6/J) was generated and kindly provided by Prof. Christian Wolfrum's laboratory (ETHZ). Primary ASPCs were isolated from subcutaneous fat tissue of C57BL/6J mice. |
| Authentication | None of the cell lines used were authenticated. |

| Mycoplasma contamination | The cell lines were not tested for mycoplasma contamination. |
| Commonly misidentified lines<br>(See ICLAC register) | No cell line is misidentified to our knowledge. |

# Animals and other organisms

Policy information about studies involving animals; ARRIVE guidelines recommended for reporting animal research

| Laboratory animals | All mice used in the remaining experiments were wild-type C57BL/6J males and females (median of age: 8.7 weeks ranging 7-11 weeks, median of weight: 22.5 g ranging 20.4-23.4 g). Mice were housed in SFP condition at 22 °C with 40%-60% humidity and a 12 light/12 dark cycle. |
| Wild animals | No wild animals were used in the study. |
| Field-collected samples | The study did not involve field-collected samples. |
| Ethics oversight | All mouse experiments were conducted in strict accordance with the Swiss law, and all experiments were approved by the ethics commission of the state veterinary office (license number VD 3406, valid from 14/10/2018 to 14/01/2022). |

Note that full information on the approval of the study protocol must also be provided in the manuscript.

