## [Peer Review File · Nature]

Manuscript Title: Live-seq enables temporal transcriptomic recording of single cells.

Reviewer Comments & Author Rebuttals

Reviewer Reports on the Initial Version:

Referees' comments:

Referee #1 (Remarks to the Author):

The paper "Genome-wide molecular recording using Live-seq" by Chen et al combines Fluid-FM, a technique that uses an AFM probe with an internal channel to manipulate fluids, and sc-RNA-seq to study the single cell response of macrophages to LPS.

The paper is well written.

I have 2 major concerns:

1) The authors overstate the technical novelty. While the group is among the world-leaders in this approach, the technique remains unfortunately only used by few. However, this that does not justify to claim the level of novelty as is done here, for what looks like a rather incremental novel application type of the existing technology. To the best of my knowledge, the technique has been introduced by Meister et al, NanoLetters 2009 (FluidFM: Combining Atomic Force Microscopy and Nanofluidics in a Universal Liquid Delivery System for Single Cell Applications and Beyond), and was then, and since then, used to inject and extract fluids into and out of cells. Later, several of the authors of this manuscript have 'introduced' in Guillaume-Gentil et al, Cell 2016 (Tunable Single-Cell Extraction for Molecular Analyses) the quantitative extraction of cytoplasm from single cells for various assays, including the extraction of RNA with subsequent PCR. Thus, while the work is technically of top-level quality in this field, there is conceptually much less novelty than the authors claim.

2) The authors state that "Live-seq enables the profiling of cell transcriptomes without imposing major perturbations on a cell's basic properties such as viability, transcriptome, or growth". Based on the cell images, eg Fig 1a, Fig 3d, 4j, and as documented in Extended Data Figure 3 - part 1, the cells (depending on the kind) have a volume of 1pL to 5pL. The RAW cells have a rather narrow volume distribution around 1pL (peaking at 1pL, with a mean of 1.2pL). According to the methods, the authors extract in average 1.1pL (line 215) or between 0.1pL and 4.4pL with mean 1.1pL (line 548, 549). The volume extraction is performed by pressure changes and estimated using geometrical considerations (no loss considered, despite the large cell injury). Altogether, especially for the RAW cells, we must assume, that the process basically depletes 100% of the cytoplasm in a highly non-specific way (see Extended Data Figure 3 - part 1a compare left panel and right panel, row RAW). It just puzzles me that the authors report a 86% viability of the RAW cells and claim 'no major perturbations'.

Minor: The force of 500nN to maintain the probe inside the cell also seems excessive. In typical AFM experiments, one would expect that such forces would damage the cells irreversibly.

Based on the analysis, the authors conclude that the expression levels of Nfkbia and Gelsolin were dominant in the macrophage response to LPS exposure. I find these findings interesting, and the approach (while not that novel) exciting, but doubt that the method does not induce significant alterations to the cells, which makes the method - at this point in time - similar to and maybe less performing than 'end-point' scRNA-seq.

Referee #2 (Remarks to the Author):

In their manuscript, Chen et al present Live-seq, a method for longitudinal single-cell transcriptomic profiling from individual cells. Live-seq utilizes fluid force microscopy (FluidFM) to perform cytoplasmic biopsies; multiple subsamples can be made from the same cell to assess cellular dynamics. The authors first describe the development of their method. After, they benchmark it and finally apply it to characterize transcriptional dynamics during the LPS response in RAW 264.7 cells.

Overall, the work describes a first-of-its-kind platform for direct, physical, cytosolic transcriptome-wide longitudinal single-cell profiling. The manuscript is methodical in the main and easy to follow – after all, it traces well-worn paths in molecular optimization and benchmarking. Nevertheless, it falls short of demonstrating the transformative promise of the technique outlined in the manuscript's introduction. While we include comments about specific elements of the work, we cannot help but feel as though the manuscript is incomplete. As it stands, with minor revision, it feels like a great fit for a more methods-focused journal like Nature Methods; with a killer app (e.g., neuron before and after plasticity protocol; some cells pre-differentiation and then after; exhausted t cells to see if they can be reactivated by I/O; etc - honestly, there are many), it could be appropriate for Nature.

Below, we provide comments by section/line/figure.

The basics of Live-seq/Fig 1/ED Fig 1

Line 94: How was cellular sampling optimized? How do assay information content and cell viability depend on sampling volume? What is the throughput of the method? If 15 minutes per sample, how many cells are sampled in each experiment listed and how many runs does each figure represent (e.g., Fig 4)? If multiple independent experiments (likely given numbers), how much of the authors' observations are driven by experiment-to-experiment variability (e.g., differences in stimulation dynamics, cell density, etc; see Line 710)? A lot remains opaque.

Line 95: How is temperature controlled? Does this impact cellular response dynamics?

Lines 103-119: This should move to the supplement. The optimization of the molecular biology follows previous descriptions (like those in SmartSeq2 and Drop-seq). What is missing is more details specific to the optimization of the FluidFM method, which is the main contribution.

Line 112: cDNA yield is not the right metric for optimization – it's information content (e.g., ED Fig 1h; see ED Fig 2e). Meanwhile, how many distinct experiments (not replicates) are represented in ED Fig 1a-g (i.e., distinct reagent mixtures for each test)? This is often the greatest source of variation.

Line 124: Fig 1b: Please adjust the plot to end at 0 on y. The axes are misleading. Also, how does quality relate to read depth? Is that what drives the variance? Same for ED Fig 2a.

Live-seq enables the stratification of cell types and states/Fig 2/ ED Fig 2

Line 142: What are cell numbers per cell type/condition?

Line 145: Why did such a low percentage of cells pass filter (compare ED Fig 2i)? This is critical (coupled with the question on Line 94) re: power.

Line 150: When and how are ERCC spike-ins added? Are they preloaded into the AFM tip or are they added during RT. This is essential to determine their utility.

Line 184: What is being correlated in ED Fig 2k? There are a few figures like this. Please make sure it is clear what is being plotted.

Line 198: What is DE between Live-seq and standard scRNA-seq? How does variation change? Also, how many of the significantly DE genes in ED Fig 2q are part of the LPS response?

Live-seq preserves cell viability, transcriptome, and growth/Fig 3/ED Fig 3

Line 212: How viable are the cells if you wait longer than 6.5h? Does the stress induce long term

changes/death? This seems critical for calibrating the method and its utility ... and for understanding the data presented in Fig. 4 ... and for distinguishing it from metabolic labeling modalities (Line 366). In truth, it's hard to pinpoint something that couldn't have been done before using existing methods (e.g., metabolic labeling, which has much, much higher throughput and can look at all cellular RNAs).

Line 215: How does are cell volume, extracted volume and viability related? Co-dependence isn't plotted. A more quantitative analysis per ED Fig. 3e would be helpful.

Lines 233-247: This is a place where significantly more data is needed. Also, is this also true for primary cells? What about post-mitotic cells? We really don't have a great sense of when and where the method is useful.

Live-seq records molecular events that are predictive of a cell's downstream phenotype/ED Fig 4/ED Fig 4

Line 261: Some of the experimental details from the methods need to come to the front to aid interpretation.

Lines 271-304: First, it is unclear why baseline RNA expression (rather than protein abundance) should be predictive (relatedly, how much does gene expression change in the 30 minutes between sampling and stimulation (Line 566))? Second, there is no validation. And, third, as highlighted, several of the hits are expected. In short, it is unclear how Live-seq uniquely enabled new biological insights here. This is the biggest shortcoming of the paper. Ditto Lines 306-325.

Sequential Live-seq cell sampling to measure cellular dynamics

Line 327-355: This analysis/section is dramatically underpowered and reads as preliminary.

Dynamics are highlighted in the introduction and motivation as a key selling point but there are only 2/14 cells whose repeated sampling passed filtering. This yield seems like a flag and directly undermines the major selling points.

Minor points:

1. We would seriously reconsider what data is in main figures and what is in the ED figures. We'd bring aspects critical to the method to the main and move some that are less relevant panels to the back.
2. It's unclear from what is provided if the full transcriptome can be sampled. We might adjust the title based on a more nuanced analysis of ED Fig. 2q

Referee #3 (Remarks to the Author):

The authors used FluidFM to sample cytoplasmic sub-cellular liquid biopsies and developed a highly sensitive scRNA-seq protocol to perform RNA-sequencing on these small inputs. They demonstrate that this method, termed Live-seq, can capture cell-type and state differences. They use Live-seq to investigate if transcriptome states, prior to LPS stimulation, can explain variable macrophage responses to the LPS stimulation, as measured by Tnf-mCherry signal. They identify genes with high predictability of LPS response (slope and intercept) and that cells in S-phase had weaker LPS responses. Finally, they show that Live-seq can enable sequential biopsies of the same cell (although only performed on two cells).

The main strength of this manuscript is the introduction of a method that can take cytoplasmic biopsies of cells in a manner compatible with cell-type and -state informative RNA-seq. Importantly, the cells having "donated" cytoplasmic biopsies mostly recovered and showed e.g. similar LPS response heterogeneity. This improvement is novel and important.

The method was used to investigate molecular determinants dictating variable LPS responses in macrophages, revealing genes and cell-cycle phase as informative in separating LPS responses.

The LPS experiments provided are however a bit preliminary (see comments below) as it was not fully clear to what extent significant gene-level effects was identified. For this manuscript to be a strong candidate in Nature, it will be important to demonstrate that Live-seq has sufficient power to identify gene-level effects, as that is how it would likely have its core applications in biology.

Major comments.

1. Clarifications of timescales in all experiments

1a) The temporal aspects of all experiments provided need to be clarified. How long time does it take to perform cytoplasmic biopsies on one (or a hundred) cells. This impact on the time difference between cells in a population if they are all exposed to these biopsies.

1b) The time after Live-seq cytoplasmic biopsies to LPS stimulation and sequential Live-seq sampling was not found. I assume LPS stimulation is provided directly after finishing the Live-seq cytoplasmic sampling? However, the time required to take cytoplasmic biopsies of tens to hundred of cells takes how long (comment 1a) and how does that relate to the timing in the LPS stimulation experiments. The timing is also relevant for the experiment linking cell-cycle phase of individual cells to the variable LPS response (the extra time for the first cytoplasmic biopsy compared to the last sampled cell). If certain times are allowed post-biopsies for cells to regain vitality, that should also be clearly stated as that impact future planning of Live-seq experiments.

2. Live-seq method development

The authors demonstrate that Live-seq is sufficiently accurate in profiling sub-cellular RNAs to enable cell-type and state separations, although the separations improve when "guided" by standard scRNA-seq (i.e. sparser Live-seq transcriptomes can be projected onto a high-quality low-dimensional representation of the standard scRNA-seq transcriptome patterns). I think the authors have done a great job of developing Live-seq and it clearly has merits. Having said that, any method aiming to profile sub-cellular RNAs would benefit from using UMIs to enable molecular counting, as amplicon effects are stronger with smaller initial starting RNA populations. Although not critical, adding UMIs to the Live-seq protocol would be highly desirable to improve the RNA counting (and can perhaps improve gene-level phenotype correlations – see comment below).

3. LPS heterogeneity experiment.

3a) Limited power to identify gene-level effects?

The authors apply Live-seq to study the heterogenous response of macrophages to LPS stimulation. Using a linear model, they investigate gene expression levels (prior to stimulation) that can predict the Tnf-mCherry intensity from the same cells. The results sections provide correlation scores (r-square and Pearson correlations), with p-values for specific gene-mCherry correlations provided in Extended Data Table 3-4. The authors report the identification of known and unknown immune regulators, including Tnf, Gelsolin, Nfkbia, that show correlation with LPS responses. Nowhere in the results nor methods are the statistics performed described. Moreover, some of the reported gene-level interactions are not significant as listed in Extended Data Table 3-4 (or am I not reading these tables correctly – the tables do lack headers and information so they are not easy to follow). The results section should clearly state whether mentioned gene-level interactions were significant (with what test) or if they were not significant. This is important, since the usefulness of Live-seq do depend on the ability to robustly estimate expression levels and if they are too noisy, they might not enable biological discoveries. If Pearson correlations were used, they are highly sensitive to outliers, and is likely not a robust method and should be complemented with rank-based approaches (or experiments showing that the used linear models are appropriate). The limited statistical significance (if lack of significance for some genes) is only followed up by discussions, as no functional follow up experiments are performed on any gene-level effect.

Permutation based significance estimation to identify gene-level predictors of Tnf-mCherry responses.

3b) Irrespective of the exact statistical model used (which should be reported), I strongly advise the authors to compare their obtained gene-level effects (for slope and intercept, respectively) against large sets of random permutations of the data (where e.g. Tnf-mCherry data attached to each cell was randomized), to further investigate significance of gene interactions in a more controlled manner. This procedure should be performed on the analysis of slope and intercept, and it would provide information on the power of the Live-seq strategy to reveal molecular determinants of variable phenotypic responses.

3c) Integration of cell-cycle phase and gene-level correlations.

Subsequent cell-cycle experiments identify cells in S-phase to have weaker LPS responses. The authors do not mention whether this cell-cycle effect explains the previous results that highlighted Nfkbia as a main regulator of Tnf-mCherry response, e.g. if Nfkbia would be expressed in a cell-cycle dependent fashion. More systematically, the authors could perform the linear modeling across genes and cell-cycle phase to comprehensively assess which genes-level correlations relate to cell-cycle and which gene-level observations are independent of cell-cycle (again, with permutation-based or more comprehensive statistical analysis).

4. Contrasting Live-seq and metabolic labeling in single cells

The authors mainly discuss Live-seq against traditional scRNA-seq analysis coupled to trajectory inference. Metabolic labeling in single cells is however the more relevant comparison as these published methods (that are referenced in the introduction) do enable two temporally linked cellular transcriptome snapshots from the same cell, and the metabolic labeling approaches is high-throughput in comparison to Live-seq. A discussion on the conceptual differences of Live-seq towards metabolic labeling methods is missing and would be highly useful to readers. Live-seq is the first method to enable sequential cellular transcriptome snapshots which is highly interesting. Yet, the ability of metabolic labeling to measure pre-existing to newly transcribed RNAs contains complementing information. For example, Live-seq is not well suited to identify transcriptional effects of perturbations (as the majority of RNA measured in the second snapshot would have been there before the initial perturbation was there), whereas metabolic labeling methods excel at that. Instead, it seems that Live-seq excel in experiments spanning longer time-periods between initial cytoplasmic sampling and downstream phenotypic recordings. I think this distinction would help readers understand when the different methods could be best applied. To fully understand the temporal resolution in Live-seq, detail clarification of the timings of the experiments presented in the present manuscript (comment 1 above) is needed.

Moreover, the authors are correct in that Live-seq offers the first strategy to sample the same cell at two different time points (with no temporal dependencies) whereas metabolic labeling do achieve a similar goal but only when the two measured time periods are within 6 or 12 hours for the strategy to be powerful. However, having said that, the authors could balance the text to acknowledge that two temporal snapshots of cellular transcriptomes has in principle been achieved with metabolic labeling.

5. Tracking of individual cells over time

Where any positive controls used during Live-cell imaging to make sure the cells were tracked correctly, and consequently that the sequential biopsies were indeed performed on the same cell? Can one add dyes or barcoded polyA containing RNAs to cells during initial biopsy to have reference points later for the controlled re-capturing of cells in the sequential experiment to make sure the cell-mapping is correct?

Rebuttal Live-seq: #2021-03-04452A;

Chen, Guillaume-Gentil et al.

Author Rebuttals to Initial Comments:

We thank all three reviewers for taking the time to carefully examine our manuscript and for sharing their expertise with us. The comments greatly helped to guide us in revising the manuscript.

Referee #1 (Remarks to the Author):

The paper "Genome-wide molecular recording using Live-seq" by Chen et al combines FluidFM, a technique that uses an AFM probe with an internal channel to manipulate fluids, and sc-RNA-seq to study the single cell response of macrophages to LPS.

The paper is well written.

I have 2 major concerns:

1) The authors overstate the technical novelty. While the group is among the world-leaders in this approach, the technique remains unfortunately only used by few. However, this that does not justify to claim the level of novelty as is done here, for what looks like a rather incremental novel application type of the existing technology. To the best of my knowledge, the technique has been introduced by Meister et al, NanoLetters 2009 (FluidFM: Combining Atomic Force Microscopy and Nanofluidics in a Universal Liquid Delivery System for Single Cell Applications and Beyond), and was then, and since then, used to inject and extract fluids into and out of cells. Later, several of the authors of this manuscript have 'introduced' in Guillaume-Gentil et al, Cell 2016 (Tunable Single-Cell Extraction for Molecular Analyses) the quantitative extraction of cytoplasm from single cells for various assays, including the extraction of RNA with subsequent PCR. Thus, while the work is technically of top-level quality in this field, there is conceptually much less novelty than the authors claim.

We recognize that the acknowledgement of the advances in this study in light of previously important achievements may not have been made sufficiently clear. We have therefore modified the introduction (**Lines 98-131**) and the results (The basics of Live-seq, **Lines 135-151**) to clarify past and present achievements. Specifically, the FluidFM technology has indeed been introduced before (2009; injection proof of concept into one cell), and then extended towards extraction of molecules with a first demonstration that the detection of a few abundant house-keeping transcripts by qPCR is possible (2016). With this study, we now reach in our opinion an entirely new level. We demonstrate that biologically relevant transcriptomes from single living cells can be generated with thousands of different transcripts that cover a range of 4 orders of magnitude in expression, which has not been achieved by any other single-cell transcriptomic method. Importantly, we demonstrate that the data are of such quality that they can distinguish cell types and states reliably and can be used to detect heterogeneity that predicts future phenotypic states and allows the generation of cell trajectories from the very same cell. This achievement is remarkable since we subsample cells rather than scarifying the

entire cell. To achieve this, it was critical to optimize the extraction protocol to maximize high-quality mRNA recovery and prevent transcript degradation during extraction, to develop a protocol for material transfer from the tip, and to enhance detection sensitivity. The conceptual advance manifests in the quality of the actual data that we are now able to routinely generate based on the technical advances of the study. The latter made it possible to robustly map different cell types and states using a transcriptome-wide readout (trajectory measurement) and to directly record "past" molecular states that influence "current" cell behaviors. As mentioned above, we rephrased the text to address the reviewer's comment on technological and conceptual advances .

2) The authors state that "Live-seq enables the profiling of cell transcriptomes without imposing major perturbations on a cell's basic properties such as viability, transcriptome, or growth". Based on the cell images, eg Fig 1a, Fig 3d, 4j, and as documented in Extended Data Figure 3 - part 1, the cells (depending on the kind) have a volume of 1pL to 5pL. The RAW cells have a rather narrow volume distribution around 1pL (peaking at 1pL, with a mean of 1.2pL). According to the methods, the authors extract in average 1.1pL (line 215) or between 0.1pL and 4.4pL with mean 1.1pL (line 548, 549). The volume extraction is performed by pressure changes and estimated using geometrical considerations (no loss considered, despite the large cell injury). Altogether, especially for the RAW cells, we must assume, that the process basically depletes 100% of the cytoplasm in a highly non-specific way (see Extended Data Figure 3 - part 1a compare left panel and right panel, row RAW). It just puzzles me that the authors report a 86% viability of the RAW cells and claim 'no major perturbations'.

Thank you for pointing out this apparent contradiction. We have modified **Fig. 3** and **Extended Data Fig. 3**, the **Methods** (Cell viability after extraction), and the **Results** (Live-seq preserves cell viability, growth, and transcriptome) to clarify this aspect.

The extracted volumes can be accurately measured in the FluidFM probe (due to the fixed geometry of the cantilever tip). We can confirm that there is no leakage/loss of cell volume during the probe insertion and withdrawal without suction into the cantilever. In fact, the membrane immediately seals upon insertion and extraction. We have performed longitudinal cell volume measurements to monitor the volume of RAW cells before and after extraction (**Fig. 3b** and **Extended Data Fig. 3a, b**). These data show that the measured cell volume loss matched the measured extracted volume for each individual cell. These data were generated with RAW cells, for which we have the opportunity to estimate cell volumes due to the spherical nature of the cells. Importantly, our data show that in most cases about half of the cell volume is withdrawn in these cells.

For the other (larger) cell types, cell volume estimates are not possible in the adherent state, which we use for cell extraction. In this case, as we do not have information on the cell volume, but only the extracted volume, we might occasionally extract up to 100% of the cellular content, as the reviewer points out. The cell volume data reported in **Fig. 3a** are based on dissociated cells. Since these are spherical objects, the volume can be inferred. The data illustrate the inherent broad distribution within all cell types used in this study.

Minor: The force of 500nN to maintain the probe inside the cell also seems excessive. In typical AFM experiments, one would expect that such forces would damage the cells irreversibly.

Indeed, 500 nN is higher than the minimal force required to insert the probe into the cell. Our force feedback data revealed that we need ~30 nN up to ~300 nN, depending on the cell type and spread extent. The application of larger forces does not result in further pressure or damage on the cells (for more details on force effects, please see Guillaume-Gentil et al. Small 2013). Setting the force to 500 nN and maintain this force ensured that it is sufficient for the various cell types used in our study. It allowed us to set a threshold and, in consequence, achieve a higher throughput. The cells are not harmed upon application of higher forces than those required to drive the probe through the membrane. This is due to the sharp tip geometry and force action on the underlying substrate rather than the cell. It is in contrast to typical cell stiffness measurements (Young modulus) where mechanical forces are applied onto cells with rounded or flat surface geometry. We have modified the Methods to clarify this point (**Lines 1100-1102**).

Based on the analysis, the authors conclude that the expression levels of Nfkbia and Gelsolin were dominant in the macrophage response to LPS exposure. I find these findings interesting, and the approach (while not that novel) exciting, but doubt that the method does not induce significant alterations to the cells, which makes the method - at this point in time - similar to and maybe less performing than 'end-point' scRNA-seq.

We agree with the reviewer that it was imperative for the functionality of Live-seq to investigate potential alterations to the cells and it resonates with us. We provide both direct and indirect data to address this concern. Specifically, to explore potential molecular effects, we sampled IBA cells and compared the single cell transcriptomes of these cells 1h and 4h post-probing to control IBA cells (i.e. unprobed cells that also did not receive a biological stimulus during this time window). As described in the Results, downstream data analysis revealed no distinct condition-related clusters (**Fig. 3e**), further supported by the observation that only 12 genes were found to be significantly differentially expressed (DE) among the three conditions (**Fig. 3f**). Based on these findings, we conclude that Live-seq does not induce major, short-term gene expression alterations. Additional molecular evidence is indirect and consists of superimposing Live-seq data with "end-point" scRNA data obtained under the same experimental conditions. These integrative analyses revealed an overall congruence of the data in that scRNA-seq and Live-seq data tends to seamlessly integrate (e.g. **Fig. 2e**). Finally, we assessed cell viability (**Fig. 3a**), cell division (**Fig. 3b, c**), and cell function ("phenotypic response to LPS", **Extended Data Fig. 4e.**) as well as the newly added "adipogenic capacity" (**Fig. 4c** and **Extended Data Fig. 4b; Lines 710-725**). While perhaps unexpected, all these data are consistent and suggest that Live-seq does not impose major perturbations on the function of the probed cells.

Referee #2 (Remarks to the Author):

In their manuscript, Chen et al present Live-seq, a method for longitudinal single-cell transcriptomic profiling from individual cells. Live-seq utilizes fluid force microscopy (FluidFM) to perform cytoplasmic biopsies; multiple subsamples can be made from the same cell to assess cellular dynamics. The authors first describe the development of their method. After, they benchmark it and finally apply it to characterize transcriptional dynamics during the LPS response in RAW 264.7 cells.

Overall, the work describes a first-of-its-kind platform for direct, physical, cytosolic transcriptome-wide longitudinal single-cell profiling. The manuscript is methodical in the main and easy to follow – after all, it traces well-worn paths in molecular optimization and benchmarking. Nevertheless, it falls short of demonstrating the transformative promise of the technique outlined in the manuscript's introduction. While we include comments about specific elements of the work, we cannot help but feel as though the manuscript is incomplete. As it stands, with minor revision, it feels like a great fit for a more methods-focused journal like Nature Methods; with a killer app (e.g., neuron before and after plasticity protocol; some cells pre-differentiation and then after; exhausted t cells to see if they can be reactivated by I/O; etc - honestly, there are many), it could be appropriate for Nature.

We thank the reviewer for appreciating the value of our work. The reviewer is correct, many transformative applications are now possible, even beyond the molecular recording as a novelty that we already provided in the original version. We agree with the provided, constructive suggestions and have now conducted an entirely new experiment on cell differentiation ("cells pre-differentiation and then after"), as one of the prime applications of interest. To demonstrate the potential of Live-seq in this respect, we chose an adipogenic differentiation model of primary adipose stem and progenitor cells (ASPCs) (**Lines 710-725**). To do so, we GFP-labeled a subpopulation of ASPCs with each cell containing a unique barcode, and biopsied them to profile their transcriptome in the pre-differentiated state. We then induced their differentiation with a chemical cocktail and sampled the same cells a second time, two days later, to profile their transcriptome in a differentiating state (see **Lines 710-725, Methods** (Cytoplasmic biopsies), and **Fig. 4c**). Using these strategies, we succeeded to sequentially sample 44 cells and obtained eight paired, quality control-passing gene expression profiles from ASPCs, as confirmed by the recovery of the correct, respective barcodes (**Fig. 4e**). Further monitoring of the cells for up to seven days after the second extraction revealed that cell viability was not compromised (we lost 2 cells out of 44 compared to 3 cells from 41 non-extracted control cells). In addition, we observed lipid droplets in these sequentially probed cells indicative of their retained adipogenic differentiation capacity (for representative images, see **Extended Data Fig. 4b**). Projection of the retrieved ASPC transcriptomes onto the integrated Live-seq and scRNA-seq data revealed the correct transition from pre- to differentiating ASPCs (**Fig. 4d** and **Extended Data Fig. 4a**). Thus, our results demonstrate that for both rapid (macrophage response to LPS) and slower (adipogenic differentiation of ASPCs) transition models, Live-seq data can be exploited to unambiguously establish the "correct" trajectory of cells that were

processed using conventional scRNA-seq. This is in our opinion an unprecedented, technological achievement.

In parallel, we used Live-seq's molecular recording capacity to investigate the factors that determine LPS response heterogeneity of macrophages. We found that the basal expression level of *Nfkb1a* is a negative predictor of LPS-induced *Tnf* expression (**Fig. 4h**). In our revised manuscript, we now validate this finding using a newly created *Nfkb1a*-BFP reporter cell line (**Fig. 4j and Extended Data Fig. 4j**). We also found and validated that cells in S phase respond weaker to LPS stimulation (**Fig. 4k-l**). Together, these data provide new insights into how stochastic variation in inflammatory response among macrophages arises, which, we believe, could not be captured by a snapshot scRNAs-seq type of approach (**Fig. 4i**).

Below, we provide comments by section/line/figure.

1. The basics of Live-seq/Fig 1/ED Fig 1

1-1) Line 94: How was cellular sampling optimized?

We have amended the text to better describe our efforts to optimize cellular sampling. Specifically, our efforts to optimize cellular sampling are described in the first part of the results (**Lines 135-151**, detailed in **Methods**, Cytoplasmic biopsies), which, together with the Introduction (**Lines 98-131**), we have now modified to better explain the past and present technological developments enabling biopsy sampling for downstream transcriptome profiling.

How do assay information content and cell viability depend on sampling volume?

Cell viability as a function of extracted volume is provided in **Fig. 3a** (former Extended Data Fig. 3a). Within the investigated volume range, we could not detect a dependence of the information content on the volume.

What is the throughput of the method? If 15 minutes per sample, how many cells are sampled in each experiment listed and how many runs does each figure represent (e.g., Fig 4)?

The entire procedure takes approximately 15 min per extraction, with the cytoplasmic extraction itself lasting up to 5 min. We performed 43 extraction experiments in total, sampling 10 to 20 cells in each experiment, which were further processed in five batches of library preparation. In the future, we anticipate that the throughput of Live-seq will be improved with automation. However, we like to point out that for our study, sampling a greater number of cells was not limited by the extraction itself but rather by the entire workflow which also involved the continuous monitoring of cell fate as well as imaging of cells with single cell resolution to provide cell viability and differentiation data. At present, data analysis (image and RNA-seq) has also been a limiting factor and in fact tends to surpass the time required for actual experimental handling. We are confident that sustained efforts and automation on pre-set parameters will enhance throughput. This is true in particular for studies focused on one particular Live-seq

application in the future. We have now modified the Methods to provide more details as requested (**Lines 1117-1121**).

If multiple independent experiments (likely given numbers), how much of the authors' observations are driven by experiment-to-experiment variability (e.g., differences in stimulation dynamics, cell density, etc; see Line 710)? A lot remains opaque.

While the batch effect is always a source of technical variation in single cell-based approaches, we could show that the cell identity rather than the batch is the main driver of variation (**Fig. 2b** and **Extended Data Fig. 2f**). The seamless integration of scRNA-seq and Live-seq data (**Fig. 2e**) further indicates that the batch effects are minor compared to the observed biological heterogeneity.

1-2) Line 95: How is temperature controlled? Does this impact cellular response dynamics?

For FluidFM manipulations and for live cell imaging, the temperature is controlled using a temperature-controlled incubation chamber (Zeiss) (See **Methods**, Optical microscopy setup). Therefore, we do not expect undesired temperature shift responses. In addition, all the experiments in this study were performed under the same environmental conditions. The cell samples in our study features cells that were targeted for extraction but also includes non-extracted neighboring cells that were used as controls. Dynamic cellular responses that were investigated and presented in this study included these control cells and were subjected to the same environmental conditions as the extracted cells.

1-3) Lines 103-119: This should move to the supplement. The optimization of the molecular biology follows previous descriptions (like those in SmartSeq2 and Drop-seq). What is missing is more details specific to the optimization of the FluidFM method, which is the main contribution.

We have amended the text and details (**Lines 135-142**) illustrated in **Fig. 1a** and **Methods** (Cytoplasmic biopsies). The main optimizations are i) reducing the extraction time, ii) lowering the temperature, iii) implementing a preloading of the FluidFM probe with sampling buffer with the goal of immediately mixing the extracted cytoplasmic fluid with RNase inhibitors, iv) releasing the extract into a microliter droplet containing buffer that is compatible with downstream RNA-seq, v) introducing a washing step to avoid cross contamination, and vi) implementing image-based cell tracking for sequential extraction. We thereby believe that also the part on the optimization of the underlying biochemistry is of similar importance, as delineated in the current manuscript.

1-4) Line 112: cDNA yield is not the right metric for optimization – it's information content (e.g., ED Fig 1h; see ED Fig 2e). Meanwhile, how many distinct experiments (not replicates) are represented in ED Fig 1a-g (i.e., distinct reagent mixtures for each test)? This is often the greatest source of variation.

We agree with the reviewer that the library information content is the golden metric for experimental optimization. Specific to former line 112 (**Extended Data Fig. 1e**), our main goal was to reduce primer concatemers, which can be observed in both the cDNA profile (**Extended Data Fig. 1f**) and the cDNA yield from real RNA and negative controls (0 pg input RNA) in highly cost- and time-efficient fashion. Thereafter, we indeed validated the information content of the libraries in the optimized condition (**Extended Data Fig. 1h**). As the reviewer points out, variation between different batches can be extensive in general for various scRNA-seq methods. The variation inside one batch is smaller, which is also shown in **Extended Data Fig. 2e**. As such, replicates were shown in **Extended Data Fig. 1a-g**, with 2-5 distinct experiments yielding congruent results. This information has been added to the legend of **Extended Data Fig. 1** of the revised manuscript (**Lines 1918-1919**).

1-5) Line 124: Fig 1b: Please adjust the plot to end at 0 on y. The axes are misleading. Also, how does quality relate to read depth? Is that what drives the variance? Same for ED Fig 2a.

We have modified the figure as requested. As data in **Extended Data Fig. 2a** involved more cells, we used it to investigate the impact of read depth. We found that read depth is indeed a source of variance (Left panel, Figure below), but downsampling to the same read depth revealed that it is in fact not the only factor (Right panel, Figure below). Since these results are consistent with observations for conventional scRNA-seq (Luecken and Theis, MSB, 2019), we opted not to include these analyses in the revised manuscript, but would be happy to do so upon the reviewer's / editor's request.

Figure Legend: (Left panel) Correlation between the number of input reads and the data quality (nGene, the number of genes detected). $R^2 = 0.24$, $P = 5.2e-33$, F-test. (Right panel) The number of detected genes is still variable when the data were downsampled to feature an equal amount of sequencing reads per cell.

2. Live-seq enables the stratification of cell types and states/Fig 2/ ED Fig 2

2-1) Line 142: What are cell numbers per cell type/condition?

There are 61 not-differentiated ASCs, 37 differentiated ASCs, 55 IBA cells, 102 mock-treated RAW cells and 50 LPS-treated RAW cells. To improve clarity, we have now extended the description of the **Fig. 2b** legend in the revised manuscript.

2-2) Line 145: Why did such a low percentage of cells pass filter (compare ED Fig 2i)? This is critical (coupled with the question on Line 94) re: power.

We extracted on average 1.1 pL of cytoplasm, as detailed in **Lines 1105-1106**. This represents a very low amount of input, rendering the standard scRNA-seq biochemistry not sufficient to amplify the RNA in all cases. It is in fact the main reason why we have modified the Smart-seq2 approach to increase overall detection sensitivity. We expect that additional improvements in sensitivity will be offered by rapidly evolving biochemical scRNA-seq approaches such as the Smart-seq3 biochemistry. As a consequence, we anticipate that a higher percentage of cells (currently around 40%) will eventually pass the stringent filtering criteria that we applied in this study. To make this important point clearer, we have amended the Discussion in our revised manuscript (**Lines 879-881**).

2-3) Line 150: When and how are ERCC spike-ins added? Are they preloaded into the AFM tip or are they added during RT. This is essential to determine their utility.

We add the ERCC during the RT (see **Methods** for further details). Adding ERCC to the preloaded sampling buffer provides an opportunity to gage overall sample loss during the sampling process. However, as the volume of preloaded sampling buffer is variable, preloading the ERCC into the tip is less practical at the current stage.

2-4) Line 184: What is being correlated in ED Fig 2k? There are a few figures like this. Please make sure it is clear what is being plotted.

Extended Data Fig. 2k shows the gene expression correlation between each cell and the average of cells in the indicated groups. For example, each IBA cell correlates to the average of all IBA cells. The reviewer might also refer to **Extended Data Fig. 2c**. It is similar to **Extended Data Fig. 2k**, but the ERCC rather than gene expression was correlated. We have clarified this in the revised Figure legends accordingly (**Lines 1967 and 1979**).

2-5) Line 198: What is DE between Live-seq and standard scRNA-seq? How does variation change? Also, how many of the significantly DE genes in ED Fig 2q are part of the LPS response?

We have included additional analyses in the revised manuscript to address these questions. As shown in **Extended Data Fig. 2q**, 72% of the Live-seq DE genes are shared with scRNA-seq and 46% of scRNA-seq DE genes are shared with Live-seq. Further analysis showed that the gene expression change among Live-seq cell populations is highly correlated with that of scRNA-seq across all sampled cell populations (**Fig. 2d**). For the LPS-treated RAW cells, we show in **Extended Data Figs. 2h and 2i** that LPS treatment-related genes are highly enriched, as expected, with around 40% of the top 100 DE genes of both Live-seq and scRNA-seq cells being directly involved (**Extended Data Tables 1 and 2**). We refer to this new analysis in the revised manuscript (**Lines 342-345**).

3. Live-seq preserves cell viability, transcriptome, and growth/ED Fig 3

3-1) Line 212: How viable are the cells if you wait longer than 6.5h? Does the stress induce long term changes/death? This seems critical for calibrating the method and its utility ... and

for understanding the data presented in Fig. 4 ... and for distinguishing it from metabolic labeling modalities (Line 366). In truth, it's hard to pinpoint something that couldn't have been done before using existing methods (e.g., metabolic labeling, which has much, much higher throughput and can look at all cellular RNAs).

1) The viability of RAW cells was monitored for ~10 hours post extraction (See **Methods** – Cell viability after extraction). We note here that all the cells that died after undergoing extraction were already dead when starting the time-lapse monitoring, i.e. within 1 to 4 hours after extraction. No cell death was observed at later time points compared to control cells. The viability of IBA and ASPC cells was assayed 2-4 hours post extraction, with similar results as for the RAW cells. In the LPS treatment experiments with RAW cells, we observed the cells for 18-24 hours and did not observe additional cell death events. Perhaps most convincingly (and also surprisingly), we showed that there are only few gene expression changes between probed and control IBA cells within 1h and 4h after the cytoplasmic biopsy, a time window that corresponds to stress / immediate response changes (e.g. serum, dissociation etc.). Given the limited gene expression differences within this immediate response timeframe, we deem it highly unlikely that cells would still respond beyond this time window, consistent with our morphological and functional cell observations.

Finally, our new sequential sampling analysis of differentiating ASPCs allowed us to monitor the possible phenotypic impact of cellular biopsies over an extended time window (seven days), revealing that 95% (N=44) of the sequentially probed cells appeared viable compared to 90% for control, non-extracted cells (N=41), as mentioned in the general comments above (**Lines 718-723**). In addition, we observed lipid droplets in these Live-seq-subjected cells indicative of their retained adipogenic differentiation capacity (for representative images, see **Extended Data Fig. 4b**), together providing further support for our observation that Live-seq does not appear to induce major phenotypic changes in the sampled cells.

2) As for the comparison to metabolic labelling technologies, the latter indeed enable the detection of old and new transcripts, and undoubtedly contribute to the inference of cellular trajectories, which we acknowledge in the original manuscript. However, we hope that the reviewer will appreciate that these technologies are subject to the same limitation as conventional scRNA-seq methods, namely that they only allow the cell to be profiled once. A full reconstruction of a "past" molecular state using a "present" transcriptome thus requires knowledge of both the synthesis and degradation rates of each RNA species. While the RNA synthesis rate can be successfully recorded thanks to metabolic labelling, the degradation rate needs to be modeled / inferred. This is why such approaches tend to still be regarded as "inference" tools rather than "direct measurement" assays. This notion is acknowledged by the developers of such labelling technologies. For example, the developers of the sci-fate assay specifically state in their paper that they are "inferring single-cell transcriptional dynamics with sci-fate" (Cao et al., Nature Biotech, 2020). In addition, they describe their workflow as one that "captures information analogous to RNA velocity". In contrast, Live-seq allows the direct measurement of the molecular state of a cell without lysing it, which we now show both for rapid (LPS response) and slower (ASPC adipogenesis) cell state transitions. This constitutes a

fundamental difference to existing scRNA-seq methods including metabolic labelling technologies, which we have now better highlighted in the revised manuscript (**Lines 866-878**).

3-2) Line 215: How does are cell volume, extracted volume and viability related? Co-dependence isn't plotted. A more quantitative analysis per ED Fig. 3e would be helpful.

The general co-dependence between the extracted volume and the viability is shown in **Fig. 3a**. In most cases, while the extracted volume can be reliably measured from microscopy images of the microchannel after extraction, the volume of a particular cell undergoing extraction is unknown. Adherent cells indeed require dissociation to assume a spherical morphology and enable an estimation of their volume, which prevents measuring their volume before and after extraction. In the case of the semi-adherent RAW cells however, the measurement became feasible, as shown in former **Extended Data Fig. 3d, e**. We have now extended these data (**Fig. 3b, Extended Data Fig. 3a and 3b**) to clarify these results. We implemented the volume growth profiles of 4 control cells without LPS stimulation, 4 control cells with LPS stimulation, and 4 additional extracted cells stimulated with LPS (12 profiles of extracted cells in total). Such longitudinal volume growth was not measured for dead cells, as those did not present a spherical morphology as required for the measurement. Please see also our response to reviewer #1, comment 2) and the response to the next comment.

3-3) Lines 233-247: This is a place where significantly more data is needed. Also, is this also true for primary cells? What about post-mitotic cells? We really don't have a great sense of when and where the method is useful.

While extraction is possible with any type of cell, the approximation of cell volume is limited to non- or semi-adherent cells, which have a spherical shape. In our study, we used semi-adherent RAW cells to convey a sense of what is the fraction of the cell volume that is effectively extracted. Measuring the volumes of adherent cells is currently not possible in living cells without perturbation. We believe however that the presented data are consistent and adequately demonstrate how the volume drops upon extraction, and how the cell recovers the volume change thereafter. We have now added more longitudinal volume profiles for extracted RAW cells (with LPS stimulation), and profiles of RAW cells that were not extracted (with and without LPS stimulation) (**Fig. 3b, Extended Data Fig. 3 a, b**).

4. Live-seq records molecular events that are predictive of a cell's downstream phenotype/Fig 4/ED Fig 4

4-1) Line 261: Some of the experimental details from the methods need to come to the front to aid interpretation

In our revised manuscript, we have adjusted the **Methods** section to improve overall clarity (**Lines 1132-1137**).

4-2) Lines 271-304: First, it is unclear why baseline RNA expression (rather than protein abundance) should be predictive (relatedly, how much does gene expression change in the 30

minutes between sampling and stimulation (Line 566)? Second, there is no validation. And, third, as highlighted, several of the hits are expected. In short, it is unclear how Live-seq uniquely enabled new biological insights here. This is the biggest shortcoming of the paper. Ditto Lines 306-325.

We agree with the reviewer that it is unclear whether baseline RNA expression can be predictive of the downstream phenotype or not. This is a fundamentally unsolved question in many biological systems, largely because there is no transcriptome-wide molecular recording method that links the baseline RNA profile with downstream phenotypes. We hope that the reviewer will appreciate that this is exactly what Live-seq aims for.

Furthermore, and as indicated above, we intentionally studied LPS macrophage response heterogeneity, since this system has been well described, allowing us to benchmark Live-seq. We thus agree and also expected to "rediscover" genes described in the context of LPS macrophage response and find this reassuring and important. Regarding new candidate genes, we found that RAW cells in S phase respond weaker to LPS. We validated this interesting finding experimentally. Our results also point to NFKBIA as the principal baseline determinant of downstream heterogeneity. The uniqueness here is that Live-seq allowed us to generate a transcriptome-wide gene ranking, which cannot be derived based on targeted approaches. Moreover, even conventional scRNA-seq does not support such inference, as shown in **Fig. 4h, i**. To further strengthen the finding that is a major determinant, we generated a RAW-G9 line containing a BFP reporter under the control of the *Nfkb* promoter and were able to confirm that BFP fluorescence is induced by LPS treatment like the endogenous *Nfkb* expression, synchronously with the induction of the *Tnf*-mCherry reporter (**Extended Data Fig. 4j**). The use of the reporter line allowed us to demonstrate that, as hypothesized, basal *Nfkb*-BFP intensity is a negative predictor ($R^2 = 0.12$, $P = 0.003$, F test) of the rate of *Tnf*-mCherry intensity increase (the slope) (**Fig. 4j**). We have included this additional validation in the revised manuscript (**Lines 827-834**) and also expanded the **Methods** section (*Nfkb* reporter analyses).

5. Sequential Live-seq cell sampling to measure cellular dynamics

Line 327-355: This analysis/section is dramatically underpowered and reads as preliminary. Dynamics are highlighted in the introduction and motivation as a key selling point but there are only 2/14 cells whose repeated sampling passed filtering. This yield seems like a flag and directly undermines the major selling points.

We have substantially restructured the manuscript due to the addition of a new section on cell state transitions as a consequence of cell differentiation (**Fig. 4c-e**). While it is correct that we still have relatively few cell pairs (four for macrophages (two in the original manuscript) and eight for ASPCs), in our opinion, our results with the present throughput indicate that it is often already sufficient to probe the cell using Live-seq just once as phenotypes of the very same cell can be observed later. A second Live-seq sampling from the same cell is only desirable when further probing of the cell is required (e.g. over a longer timeframe) and was included here as a proof of concept. In addition, our sequential Live-seq data demonstrate that the probing of relatively few cells can already contribute to resolving complex biological processes

in ways that snapshot scRNA-seq data cannot. These include the empirical determination of trajectories through Live-seq and conventional scRNA-seq data integration as well as the prediction of heterogeneous phenotypic behavior. Thus, we deem conventional scRNA-seq and Live-seq to be highly complementary in that scRNA-seq can define the manifold in high-dimensional space, while Live-seq provides guidance with respect to the information flow within that space. For such a hybrid approach, the monitoring of fewer cells is possible and therefore fully aligned with Live-seq's current throughput. However, we acknowledge that technological innovations will still be required to standardize such analyses and perform them at a larger scale (**Lines 816-819, 878-885, and 967-969**).

Minor points:

1. We would seriously reconsider what data is in main figures and what is in the ED figures. We'd bring aspects critical to the method to the main and move some that are less relevant panels to the back.

We modified and expanded Figures and Extended Data. Specifically, Figs. 2d, 4c, 4e, 4j, and Extended Data Figs. 2q, 2s, 3b, 4b, 4j represent newly added data, Figs. 3a, 3b were transferred to the main Figures and Extended Data Figs. 4e, 4m were transferred to Extended Data Figures. Please also note that most panels of Figs. 2-4 and Extended Data Figs. 2-4 have also been updated as new samples were added.

2. It's unclear from what is provided if the full transcriptome can be sampled. We might adjust the title based on a more nuanced analysis of ED Fig. 2q

While the number of detected genes by Live-seq is lower than that by conventional scRNA-seq, as expected given the lower RNA input (shown in **Fig. 3a**), the seamless integration of Live-seq with canonical scRNA-seq data supports the conclusion that Live-seq comprehensively samples the transcriptome of individual cells (**Fig. 2e**). Nevertheless, and as the reviewer suggested, we have performed additional analyses to directly compare Live-seq and scRNA-seq data in the revised manuscript (**Fig. 2d** and **Extended Data Fig. 2q-r**). Taken together, we believe that these data validate the quality of Live-seq data.

Referee #3 (Remarks to the Author):

The authors used FluidFM to sample cytoplasmic sub-cellular liquid biopsies and developed a highly sensitive scRNA-seq protocol to perform RNA-sequencing on these small inputs. They demonstrate that this method, termed Live-seq, can capture cell-type and state differences. They use Live-seq to investigate if transcriptome states, prior to LPS stimulation, can explain variable macrophage responses to the LPS stimulation, as measured by Tnf-mCherry signal. They identify genes with high predictability of LPS response (slope and intercept) and that cells in S-phase had weaker LPS responses. Finally, they show that Live-seq can enable sequential biopsies of the same cell (although only performed on two cells).

The main strength of this manuscript is the introduction of a method that can take cytoplasmic biopsies of cells in a manner compatible with cell-type and -state informative RNA-seq. Importantly, the cells having “donated” cytoplasmic biopsies mostly recovered and showed e.g. similar LPS response heterogeneity. This improvement is novel and important.

The method was used to investigate molecular determinants dictating variable LPS responses in macrophages, revealing genes and cell-cycle phase as informative in separating LPS responses. The LPS experiments provided are however a bit preliminary (see comments below) as it was not fully clear to what extent significant gene-level effects was identified. For this manuscript to be a strong candidate in Nature, it will be important to demonstrate that Live-seq has sufficient power to identify gene-level effects, as that is how it would likely have its core applications in biology.

Major comments.

1. Clarifications of timescales in all experiments

1a) The temporal aspects of all experiments provided need to be clarified. How long time does it take to perform cytoplasmic biopsies on one (or a hundred) cells. This impact on the time difference between cells in a population if they are all exposed to these biopsies.

We have improved our manuscript to enhance overall clarity. We now detail the temporal aspects in the **Methods** for the time required for a biopsy (**Lines 1117-1121**), and the time intervals during which the cells were sampled before and after application of a stimulus (LPS: **Lines 1131-1146**; adipogenic differentiation: **Lines 1148-1173**).

1b) The time after Live-seq cytoplasmic biopsies to LPS stimulation and sequential Live-seq sampling was not found. I assume LPS stimulation is provided directly after finishing the Live-seq cytoplasmic sampling? However, the time required to take cytoplasmic biopsies of tens to hundred of cells takes how long (comment 1a) and how does that relate to the timing in the LPS stimulation experiments. The timing is also relevant for the experiment linking cell-cycle phase of individual cells to the variable LPS response (the extra time for the first cytoplasmic biopsy compared to the last sampled cell). If certain times are allowed post-biopsies for cells to

regain vitality, that should also be clearly stated as that impact future planning of Live-seq experiments.

We have improved our manuscript to clarify the timescales in all experiments (**Methods, Lines: 1117-1173**). For the hours' time-scale studies that are presented in this work, 4 to 5 cells were extracted per individual sample run within a time window of 1 h. The cells were monitored for 30 minutes before adding the LPS to record a *Tnf* reporter signal baseline before LPS stimulation. For the days' time-scale study involving ASPC differentiation, cells were extracted during 3-4 hours, and the chemical cocktail to induce differentiation added directly after the last extraction. The number of samples and the number of extractions per sample can be adapted as desired to fulfill specific time-scale requirements of the biological system under investigation.

2. Live-seq method development

The authors demonstrate that Live-seq is sufficiently accurate in profiling sub-cellular RNAs to enable cell-type and state separations, although the separations improve when “guided” by standard scRNA-seq (i.e. sparser Live-seq transcriptomes can be projected onto a high-quality low-dimensional representation of the standard scRNA-seq transcriptome patterns). I think the authors have done a great job of developing Live-seq and it clearly has merits. Having said that, any method aiming to profile sub-cellular RNAs would benefit from using UMIs to enable molecular counting, as amplicon effects are stronger with smaller initial starting RNA populations. Although not critical, adding UMIs to the Live-seq protocol would be highly desirable to improve the RNA counting (and can perhaps improve gene-level phenotype correlations – see comment below).

We agree with the reviewer that the incorporation of UMIs could reduce the application bias. However, it is well established that full length-based Smart-seq2 has much greater sensitivity than UMI counting-based approaches using a similar biochemistry (Ding et al., Nature Biotech, 2020). Even in the recently released Smart-seq3 workflow, reads bearing a UMI take up only a minor proportion of total reads. Given the small mRNA amounts in the Live-seq procedure, we anticipate that such reads would be even fewer and thus provide only limited information at present. However, we agree that it is indeed desirable to improve detection sensitivity and throughput such that the application of UMIs can be implemented.

3. LPS heterogeneity experiment.

3a) Limited power to identify gene-level effects?

The authors apply Live-seq to study the heterogenous response of macrophages to LPS stimulation. Using a linear model, they investigate gene expression levels (prior to stimulation) that can predict the *Tnf*-mCherry intensity from the same cells. The results sections provide correlation scores (r-square and Pearson correlations), with p-values for specific gene-mCherry correlations provided in Extended Data Table 3-4. The authors report the identification of known and unknown immune regulators, including *Tnf*, *Gelsolin*, *Nfkb*, that show correlation with LPS responses. Nowhere in the results nor methods are the statistics performed described. Moreover, some of the reported gene-level interactions are not significant as listed in Extended

Data Table 3-4 (or am I not reading these tables correctly – the tables do lack headers and information so they are not easy to follow). The results section should clearly state whether mentioned gene-level interactions were significant (with what test) or if they were not significant. This is important, since the usefulness of Live-seq do depend on the ability to robustly estimate expression levels and if they are too noisy, they might not enable biological discoveries. If Pearson correlations were used, they are highly sensitive to outliers, and is likely not a robust method and should be complemented with rank-based approaches (or experiments showing that the used linear models are appropriate). The limited statistical significance (if lack of significance for some genes) is only followed up by discussions, as no functional follow up experiments are performed on any gene-level effect.

Permutation based significance estimation to identify gene-level predictors of Tnf-mCherry responses.

3b) Irrespective of the exact statistical model used (which should be reported), I strongly advise the authors to compare their obtained gene-level effects (for slope and intercept, respectively) against large sets of random permutations of the data (where e.g. Tnf-mCherry data attached to each cell was randomized), to further investigate significance of gene interactions in a more controlled manner. This procedure should be performed on the analysis of slope and intercept, and it would provide information on the power of the Live-seq strategy to reveal molecular determinants of variable phenotypic responses.

We realize that the Extended Data Table was difficult to read in the presented PDF form. We have now provided the Extended Data Table in a more readable form.

For the linear regression model, we used an F-test to test the significance of the coefficient from linear regression, which we corrected for multiple testing (FDR). We acknowledge that most of our hits are only barely significant (e.g. the false-discovery rate of *Nfkb1a* is 0.099). We have also performed a permutation-based analysis as the reviewer suggested, and here all genes have an FDR above 0.4 (these values are included in the Extended Data Table). This difference is as expected as these non-parametric tests have a lower power. To assess whether our original approach using the F-test is correct, we verified for our top hits whether the assumptions made by the linear model are invalid, namely the normality and homoscedacity of the data. This did not seem to be the case for the top 10 genes, although we here also acknowledge that this may be hard to determine using the limited number of processed cells. The qq-plot and scale-location plots are provided below for *Nfkb1a*:

As we emphasize in our revised manuscript (**Lines 816-819**), we believe that the main mission of Live-seq in its current format is to generate leads that can then be validated using targeted approaches. Our data show that Live-seq revealed a link to the cell cycle (see also our reply below), which we were able to validate. To further demonstrate the power of our approach, we now provide additional validation on *Nfkbia*, by generating and testing a BFP reporter (see **Fig. 4j** and **Extended Data Fig. 4j**), revealing that the basal expression level of *Nfkbia* can indeed act as a negative predictor of LPS-induced *Tnf* expression (**Fig. 4j**), as hypothesized (**Fig. 4h**). We have now more specifically included a statement on the power of our current analyses in the revised manuscript (**Lines 967-969**), refined the method (**Lines 1488-1510**) and also inserted the *Nfkbia* validation data, **Lines 827-834, Fig. 4j** and **Extended Data Fig. 4j**).

3c) Integration of cell-cycle phase and gene-level correlations.

Subsequent cell-cycle experiments identify cells in S-phase to have weaker LPS responses. The authors do not mention whether this cell-cycle effect explains the previous results that highlighted *Nfkbia* as a main regulator of *Tnf*-mCherry response, e.g. if *Nfkbia* would be expressed in a cell-cycle dependent fashion. More systematically, the authors could perform the linear modeling across genes and cell-cycle phase to comprehensively assess which genes-level correlations relate to cell-cycle and which gene-level observations are independent of cell-cycle (again, with permutation-based or more comprehensive statistical analysis).

To explore whether the two processes, namely *Nfkbia* expression and cell cycle, are related, we calculated the S and G2M scores using Seurat's CellCycleScoring, and determined whether these scores are predictive for normalized *Nfkbia* expression using standard linear regression. This does not seem to be the case for both the scRNA-seq and Live-seq data ($p > 0.1$, with respectively $n = 190$ and $n = 50$), which suggests that both processes are unrelated and contribute to the lower sensitivity of a cell to LPS. We did not include this analysis in the revised manuscript; however, we would be happy to do so upon request of the reviewer / editor.

4. Contrasting Live-seq and metabolic labeling in single cells

The authors mainly discuss Live-seq against traditional scRNA-seq analysis coupled to trajectory inference. Metabolic labeling in single cells is however the more relevant comparison as these published methods (that are referenced in the introduction) do enable two temporally linked cellular transcriptome snapshots from the same cell, and the metabolic labeling approaches is high-throughput in comparison to Live-seq. A discussion on the conceptual differences of Live-seq towards metabolic labeling methods is missing and would be highly useful to readers. Live-seq is the first method to enable sequential cellular transcriptome snapshots which is highly interesting. Yet, the ability of metabolic labeling to measure pre-existing to newly transcribed RNAs contains complementing information. For example, Live-seq is not well suited to identify transcriptional effects of perturbations (as the majority of RNA measured in the second snapshot would have been there before the initial perturbation was there), whereas metabolic labeling methods excel at that. Instead, it seems that Live-seq excel in experiments spanning longer time-periods between initial cytoplasmic sampling and

downstream phenotypic recordings. I think this distinction would help readers understand when the different methods could be best applied. To fully understand the temporal resolution in Live-seq, detail clarification of the timings of the experiments presented in the present manuscript (comment 1 above) is needed.

Moreover, the authors are correct in that Live-seq offers the first strategy to sample the same cell at two different time points (with no temporal dependencies) whereas metabolic labeling do achieve a similar goal but only when the two measured time periods are within 6 or 12 hours for the strategy to be powerful. However, having said that, the authors could balance the text to acknowledge that two temporal snapshots of cellular transcriptomes has in principle been achieved with metabolic labeling.

We have now included a more comprehensive discussion about the technical differences between Live-seq and metabolic labeling technologies in our revised manuscript (**Lines 866-878**). We would thereby like to emphasize that there are fundamental differences between Live-seq and metabolic labeling methods, see also our response to the metabolic labeling comment by reviewer 2 above (point 3-1). Finally, the comment by the reviewer that “Live-seq is not well suited to identify transcriptional effects of perturbations” is well taken as we wondered about this as well. This is yet another reason why we decided to study macrophage LPS response heterogeneity given that the perturbation (i.e. LPS exposure) needed to be assessed within a relatively short timeframe. We believe that the results from these analyses show that Live-seq can adequately detect these perturbation-driven gene expression changes, also in light of our control data from our dual sampling without LPS treatment experiments (**Fig. 3d-f**).

5. Tracking of individual cells over time

Where any positive controls used during Live-cell imaging to make sure the cells were tracked correctly, and consequently that the sequential biopsies were indeed performed on the same cell? Can one add dyes or barcoded polyA containing RNAs to cells during initial biopsy to have reference points later for the controlled re-capturing of cells in the sequential experiment to make sure the cell-mapping is correct?

The cells were monitored continuously, before, during and after the extraction, by optical microscopy, at time intervals optimized to non-ambiguously track the individual cells within the field of view (≤ 30 min intervals, **Methods, Lines 1131-1146**). We did not require additional cell markers to unambiguously follow individual cells in the presented experiments, at the hours' time-scale. However, the use of dyes or molecular barcodes as suggested by the reviewer is well taken. In fact, to perform the newly added sequential sampling of ASPCs pre- and post-differentiation, whereby the cells had to be relocated after 2 days, we introduced a unique barcode in the 3' UTR of the GFP reporter that was lentivirally transduced in a subset of ASPCs. The barcode information could then be retrieved from the cDNA of each Live-seq sample and be used to evaluate the accuracy of the image-based tracking result such that samples can be confidently paired (**Fig. 4e**). These new experiments have been included in the revised manuscript (**Lines 710-725, Methods: Lines 1148-1232, Fig. 4c-e, Extended Data Fig. 4a-b**).

Reviewer Reports on the First Revision:

Referees' comments:

Referee #1 (Remarks to the Author):

Regarding point 1 from my first review, the authors have provided an improved introduction, mentioning refs 29-32 and what has been achieved before. Also, toning down some of the claims of novelty of the approach. I think the authors could still remove some of the qualitative terms of new/novel regarding the approach, as I have been thinking that their 2016 Cell paper provided this type of technical breakthrough. In the response, the authors have pointed out that they reached a 'new level', analyzing transcriptomes of relevant range, which seems to me like a fair description that should be stronger reflected in the manuscript (without the new).

Regarding point 2 from my first review, the authors admit in their response that the extracted volumes represent significant percentages of the entire cell volume ("occasionally up to 100%"), but maintain, based on their data, that the approach has a high cell viability rate and that even the cell trajectories are preserved.

In summary, I appreciated the care with which the authors have responded to my technical concerns and refer to the other reviewers to assess how strong the paper is within the field of transcriptomics, in light of the much lesser technical novelty and the potential invasiveness of the approach. Anyway, the authors are leaders in the use of FluidFM and have assembled a nice manuscript highlighting the state-of-the-art of the technique.

Referee #2 (Remarks to the Author):

In this new version of manuscript, the authors introduce several modifications to the text, figures, and data. In particular, analysis of a new Live-seq ASPC differentiation dataset is presented, as well as experimental validations of Nfkb expression and cell cycle state as predictors of LPS response in RAW 264.7 cells.

Overall, we find these new additions interesting and pertinent. However, we feel these new results still fall short of demonstrating Live-seq's promised potential at the level required for publication in Nature (e.g., answering a low-hanging, outstanding biological question). This seems to be a concern shared by Reviewers 1 and 3 as well.

Many of our previous comments were addressed. Below, we highlight remaining issues by author response (initial comments and author responses in quotes), and also provide thoughts on the manuscript's new additions.

Initial Author response:

We thank the reviewer for appreciating the value of our work. The reviewer is correct, many transformative applications are now possible, even beyond the molecular recording as a novelty that we already provided in the original version. We agree with the provided, constructive suggestions and have now conducted an entirely new experiment on cell differentiation ("cells pre-differentiation and then after"), as one of the prime applications of interest. To demonstrate the potential of Live-seq in this respect, we chose an adipogenic differentiation model of primary adipose stem and progenitor cells (ASPCs) (Lines 710-725). To do so, we GFP-labeled a subpopulation of ASPCs with each cell containing a unique barcode, and biopsied them to profile their transcriptome in the pre-differentiated state. We then induced their differentiation with a chemical cocktail and sampled the same cells a second time, two days later, to profile their transcriptome in a differentiating state (see Lines 710-725, Methods (Cytoplasmic biopsies), and Fig. 4c). Using these strategies, we succeeded to sequentially sample 44 cells and obtained eight

paired, quality control-passing gene expression profiles from ASPCs, as confirmed by the recovery of the correct, respective barcodes (Fig. 4e). Further monitoring of the cells for up to seven days after the second extraction revealed that cell viability was not compromised (we lost 2 cells out of 44 compared to 3 cells from 41 non-extracted control cells). In addition, we observed lipid droplets in these sequentially probed cells indicative of their retained adipogenic differentiation capacity (for representative images, see Extended Data Fig. 4b). Projection of the retrieved ASPC transcriptomes onto the integrated Live-seq and scRNA-seq data revealed the correct transition from pre- to differentiating ASPCs (Fig. 4d and Extended Data Fig. 4a). Thus, our results demonstrate that for both rapid (macrophage response to LPS) and slower (adipogenic differentiation of ASPCs) transition models, Live-seq data can be exploited to unambiguously establish the "correct" trajectory of cells that were processed using conventional scRNA-seq. This is in our opinion an unprecedented, technological achievement."

We agree that the ASPC differentiation assay is a pertinent application of the Live-seq method and we understand the technical feat. However, the number of paired cells is small and the results are in line with what is expected. In short, it is hard to see what Live-seq has taught us that is new. Further experiments and/or additional analyses (with validations) are needed to show that Live-seq can answer an outstanding biological question (what can you find that you could not using existing state-of-the-art methods for inference and subsequent validation). For example, are only some stem cells primed for differentiation? Is differentiation terminal for all forms of stem cells and differentiated cells? Something about the events that dictate a specific cell's decision (need to validate to establish Live-seq does not perturb this). Repeated sampling during differentiation as opposed to simply looking at two points might yield deeper insights into the process and actually enable tracking of the events that drive transitions.

We note that it is unclear why the authors consistently choose to analyze different systems (ASPC, RAW, IBA) together. This should be fixed, especially around lines 727-736. Also, please do not mix your data when reporting metrics (e.g., ED Fig 2) - it's misleading and inappropriate. Also, why do some of the IBA and RAW cells mix in ED Fig 2g?

"1-1 point 2 How do assay information content and cell viability depend on sampling volume?

Cell viability as a function of extracted volume is provided in Fig. 3a (former Extended Data Fig. 3a). Within the investigated volume range, we could not detect a dependence of the information content on the volume."

This does not answer the question. How does extracted volume impact information content (i.e., genes detected, counts, etc)? Please provide a plot.

"1-1 point 4 If multiple independent experiments (likely given numbers), how much of the authors' observations are driven by experiment-to-experiment variability (e.g., differences in stimulation dynamics, cell density, etc; see Line 710)? A lot remains opaque.

While the batch effect is always a source of technical variation in single cell-based approaches, we could show that the cell identity rather than the batch is the main driver of variation (Fig. 2b and Extended Data Fig. 2f). The seamless integration of scRNA-seq and Live-seq data (Fig. 2e) further indicates that the batch effects are minor compared to the observed biological heterogeneity."

We do not find the authors' response convincing as Fig 2b,e and ED Fig 2f do not address our concern. All analyses should be done on a cell-type-by-cell-type basis. Calling variable genes across all of the systems at once is inappropriate as it masks critical differences within a system (RAW cells are very different than ASPCs) and can easily yield the results shown. NB there appears to be separation in ED Fig 2p among RAW cells.

Please quantify your sources of variation, and provide appropriate statistics.

"2-3) Line 150: When and how are ERCC spike-ins added? Are they preloaded into the AFM tip or are they added during RT. This is essential to determine their utility.

We add the ERCC during the RT (see Methods for further details). Adding ERCC to the preloaded sampling buffer provides an opportunity to gage overall sample loss during the sampling process. However, as the volume of preloaded sampling buffer is variable, preloading the ERCC into the tip is less practical at the current stage."

As the authors acknowledge, ERCC spike-ins at RT likely yield less reliable normalizations. Given the volume variability in preloaded sampling buffer, could UMIs be used instead? Contrary to the response to Reviewer 3, this does not reduce quality or power. This would get a more accurate estimate of the information content extracted. A few simple control experiments could go a long way here.

"2-5) Line 198: What is DE between Live-seq and standard scRNA-seq? How does variation change? Also, how many of the significantly DE genes in ED Fig 2q are part of the LPS response?

We have included additional analyses in the revised manuscript to address these questions. As shown in Extended Data Fig. 2q, 72% of the Live-seq DE genes are shared with scRNA-seq and 46% of scRNA-seq DE genes are shared with Live-seq. Further analysis showed that the gene expression change among Live-seq cell populations is highly correlated with that of scRNA-seq across all sampled cell populations (Fig. 2d). For the LPS-treated RAW cells, we show in Extended Data Figs. 2h and 2i that LPS treatment-related genes are highly enriched, as expected, with around 40% of the top 100 DE genes of both Live-seq and scRNA-seq cells being directly involved (Extended Data Tables 1 and 2). We refer to this new analysis in the revised manuscript (Lines 342-345)."

Can you explain the discrepancy? The limited degree of overlap is troubling (suggests the technique is not representative or not minimally perturbative). Does nuclear vs cytoplasmic localization of the RNA help explain the differences? If driven by power, what happens if you downsample and bootstrap? A more detailed analysis of what is happening here on a cell-type-by-cell-type basis should be done. For example, what's off diagonal in ED Fig 2r and Fig 2d (also, how is fold change calculated, compared to what; please make sure all of your data and legends are well referenced)? What arises when you explicitly do a differential expression analysis within group?

Relatedly, in Fig 3e, how variable is the sampling time for each cell given the time it takes to run Live-seq - i.e., what does 1h and 4h actually represent? Relatedly, what these data show is recovery, not lack of perturbation contrary to the assertions in lines 469-478.

Also, a direct comparison of the Live-seq and single-cell RNA-seq metrics (ED Fig 2j) would be an important addition.

"3-2) Line 215: How does cell volume, extracted volume and viability related? Co-dependence isn't plotted. A more quantitative analysis per ED Fig. 3e would be helpful.

The general co-dependence between the extracted volume and the viability is shown in Fig. 3a. In most cases, while the extracted volume can be reliably measured from microscopy images of the microchannel after extraction, the volume of a particular cell undergoing extraction is unknown. Adherent cells indeed require dissociation to assume a spherical morphology and enable an estimation of their volume, which prevents measuring their volume before and after extraction. In the case of the semi-adherent RAW cells however, the measurement became feasible, as shown in

former Extended Data Fig. 3d, e. We have now extended these data (Fig. 3b, Extended Data Fig. 3a and 3b) to clarify these results. We implemented the volume growth profiles of 4 control cells without LPS stimulation, 4 control cells with LPS stimulation, and 4 additional extracted cells stimulated with LPS (12 profiles of extracted cells in total). Such longitudinal volume growth was not measured for dead cells, as those did not present a spherical morphology as required for the measurement. Please see also our response to reviewer #1, comment 2) and the response to the next comment."

We think a plot showing viability as a function of cell volume and extracted volume would be useful to the reader to understand if these are related.

"4-2) Lines 271-304: First, it is unclear why baseline RNA expression (rather than protein abundance) should be predictive (relatedly, how much does gene expression change in the 30 minutes between sampling and stimulation (Line 566))? Second, there is no validation. And, third, as highlighted, several of the hits are expected. In short, it is unclear how Live-seq uniquely enabled new biological insights here. This is the biggest shortcoming of the paper. Ditto Lines 306-325.

We agree with the reviewer that it is unclear whether baseline RNA expression can be predictive of the downstream phenotype or not. This is a fundamentally unsolved question in many biological systems, largely because there is no transcriptome-wide molecular recording method that links the baseline RNA profile with downstream phenotypes. We hope that the reviewer will appreciate that this is exactly what Live-seq aims for.

Furthermore, and as indicated above, we intentionally studied LPS macrophage response heterogeneity, since this system has been well described, allowing us to benchmark Live-seq. We thus agree and also expected to "rediscover" genes described in the context of LPS macrophage response and find this reassuring and important. Regarding new candidate genes, we found that RAW cells in S phase respond weaker to LPS. We validated this interesting finding experimentally. Our results also point to NFKBIA as the principal baseline determinant of downstream heterogeneity. The uniqueness here is that Live-seq allowed us to generate a transcriptome-wide gene ranking, which cannot be derived based on targeted approaches. Moreover, even conventional scRNA-seq does not support such inference, as shown in Fig. 4h, i. To further strengthen the finding that is a major determinant, we generated a RAW-G9 line containing a BFP reporter under the control of the Nfkb1a promoter and were able to confirm that BFP fluorescence is induced by LPS treatment like the endogenous Nfkb1a expression, synchronously with the induction of the Tnf-mCherry reporter (Extended Data Fig. 4j). The use of the reporter line allowed us to demonstrate that, as hypothesized, basal Nfkb1a-BFP intensity is a negative predictor ($R^2 = 0.12$, $P = 0.003$, F test) of the rate of Tnf-mCherry intensity increase (the slope) (Fig. 4j). We have included this additional validation in the revised manuscript (Lines 827-834) and also expanded the Methods section (Nfkb1a reporter analyses)."

Does baseline RNA expression correlate with protein abundance? This would provide support to findings of correlation between RNA expression of some genes and response to LPS. The data sets in Jovanovic et al, Science 2015 (DOI: 10.1126/science.1259038) should be instructive.

Regarding the difference in LPS response by cell cycle phase, this too is expected. Please see Allen et al, Science Signaling 2019 (DOI: 10.1126/scisignal.aau1851).

Lines 804-807: Adding the LPS data to this plot and comparing to pre-LPS Live-seq is inappropriate. There doesn't seem to be a positive correlation between Tnf and Nfkb1a at baseline (driven by the LPS stimulation).

Regarding the Nfkb1a-BFP/Tnf-mCherry validation experiment, the correlation plot would benefit from the indication of an R-squared, Spearman correlation coefficient, and associated p-value. Given that the association is not strong, it is our impression that the pattern is driven mostly by

few datapoints with high basal BFP intensity. These results would be more convincing if additional datapoints could be obtained, with more points with high intensity.

Finally, we agree with the authors that the anticorrelation of Nfkbia basal expression and LPS response/NFKB upregulation is an expected finding. In short, this is a validation of the method as pointed out above, rather than an indication of the full potential of Live-seq. The lack of new biological insights remains the biggest shortcoming of the paper.

"5. Sequential Live-seq cell sampling to measure cellular dynamics

Line 327-355: This analysis/section is dramatically underpowered and reads as preliminary. Dynamics are highlighted in the introduction and motivation as a key selling point but there are only 2/14 cells whose repeated sampling passed filtering. This yield seems like a flag and directly undermines the major selling points.

We have substantially restructured the manuscript due to the addition of a new section on cell state transitions as a consequence of cell differentiation (Fig. 4c-e). While it is correct that we still have relatively few cell pairs (four for macrophages (two in the original manuscript) and eight for ASPCs), in our opinion, our results with the present throughput indicate that it is often already sufficient to probe the cell using Live-seq just once as phenotypes of the very same cell can be observed later. A second Live-seq sampling from the same cell is only desirable when further probing of the cell is required (e.g. over a longer timeframe) and was included here as a proof of concept. In addition, our sequential Live-seq data demonstrate that the probing of relatively few cells can already contribute to resolving complex biological processes in ways that snapshot scRNA-seq data cannot. These include the empirical determination of trajectories through Live-seq and conventional scRNA-seq data integration as well as the prediction of heterogeneous phenotypic behavior. Thus, we deem conventional scRNA-seq and Live-seq to be highly complementary in that scRNA-seq can define the manifold in high-dimensional space, while Live-seq provides guidance with respect to the information flow within that space. For such a hybrid approach, the monitoring of fewer cells is possible and therefore fully aligned with Live-seq's current throughput. However, we acknowledge that technological innovations will still be required to standardize such analyses and perform them at a larger scale (Lines 816-819, 878-885, and 967-969)."

We understand that sequential sampling of the same cells favorably impacts type 1 and type 2 error allowing for smaller sample sizes. However, given stochastic, inherent variability in biological processes, a sample size of 4 appears small. Statistics should be used to guide the validity of all results discussed.

A scRNA-seq / Live hybrid approach appears promising. Could it be a way to obtain new biological insights with the existing data? We understand the conceptual promise of the approach but are having trouble seeing its proof.

"Minor point 2. It's unclear from what is provided if the full transcriptome can be sampled. We might adjust the title based on a more nuanced analysis of ED Fig. 2q

While the number of detected genes by Live-seq is lower than that by conventional scRNA-seq, as expected given the lower RNA input (shown in Fig. 3a), the seamless integration of Live-seq with canonical scRNA-seq data supports the conclusion that Live-seq comprehensively samples the transcriptome of individual cells (Fig. 2e). Nevertheless, and as the reviewer suggested, we have performed additional analyses to directly compare Live-seq and scRNA-seq data in the revised manuscript (Fig. 2d and Extended Data Fig. 2q-r). Taken together, we believe that these data validate the quality of Live-seq data."

These comparisons are instructive. Our comment refers to the potential spatial biases in RNA of different genes given that biopsies are cytoplasmic. Is the difference between DEG in Live-seq and scRNA-seq belong to a certain category of genes (e.g., nuclear vs not nuclear) or are they purely a consequence of sampling rate? This would help delineate how Live-seq signal can be interpreted.

For example, what genes are sampled in 1b relative to matched whole cells? How do the metrics compare? Ditto 2a. Fig 3 hints at this in one systems for cells at rest.

Similarly, can you show matched comparisons of Live-seq and total cell RNA-seq for ED Fig 2j? For Fig 2d/ED Fig 2r, we'd like to see enrichments of what's off-diagonal and square axes.

Referee #3 (Remarks to the Author):

The authors have addressed many of the comments from the initial round, and the manuscript has improved. It is clear that Live-seq can accurately sample cytoplasm from cells, without detrimental damage to cells, and that the sampled RNAs can provide a decent transcriptional profile that e.g. identifies correct cell types (i.e. large biological differences). To this end, it is the first transcriptome-wide demonstration of multiple scRNA-seq measurement from the same cell (Figure 2 – the strongest figure of the manuscript).

However, the biological experiments in this manuscript are still weak. The one insight generated from the LPS heterogeneity experiment was the identification of *Nfkb1a* (Fig 4h, non-significant) that was validated with borderline significance (dependent on a few outlier observations, Fig 4j), and the S-phase effect.

The new analysis (Fig 3e) that investigated the effect of cytoplasm sampling on the cell is confusing. It seems very dangerous to here claim negative results – i.e. that the cells do not separate in tSNE - since often proper separation depend on accurate identification of the biologically variable genes (tSNE parameters etc). With 18 DE genes, in my experience, the cells would be highly likely to cluster.

The discussion on metabolic labeling still misses the main point – in my opinion – that the real advantage of Live-seq lies in phenotypically linking cells over larger time intervals (>8 hours to days), as metabolic labeling will excel at shorter time points. Thus, the genes detected as differentially expressed (Fig 3f) are limited to those initially expressed at low (or no) levels, in order to be detected as differentially expressed in Live-seq (whereas metabolic labeling could detect all differentially expressed genes). On this topic, the LPS experiment would have much higher power if studied with metabolic labeling instead of Live-seq.

Altogether, I applaud the authors efforts to establish this technology, but I am worried that at its current power and cellular throughput, there will be limited interest to establish this technology in other labs. Nevertheless, I could see increased interest in Live-seq if the method could be scaled to hundreds of cells (within a reasonable time frame), although I completely understand this is currently not achievable. The rather weak biological insights provided in the manuscript makes it still less compelling for Nature.

Minor issues:

I think certain analysis in the manuscript are rather biased towards promoting Live-seq. For example, the comparisons with RNA-velocity like trajectory inferences all use inference tools that will connect all cell types, although the experiment performed is using two different experimental model systems. Proper use of inference tools on each experiment alone would likely provide the correct flow.

Figure 1b. The authors describe these QC as stringent, although they seem lenient?

Ext Data Figure 2q: The number of differentially expressed genes are more than the number of genes identified in the cell types? Must be a typo somewhere, or erroneous gene set summation.

Ext Data Figure 1b,g: show increased cDNA yields without RNA input after optimization? In fact, same cDNA yields were obtained from 0 as with 1 pg of RNA? In general, optimizations were done with very few cells per condition (2 to 3 cells per condition, that does not sound very robust).

Details on how the short time-period experiments are performed is still lacking. For 1-2 hour treatment experiments, are individual cells sampled and treated alone? And the experimental setup repeated n times (every time stimulating one sampled cell?).

All figures still look preliminary and I agree with reviewer 1 that the organization of results in main and extended data figures could be much improved.

Rebuttal for Manuscript #2021-03-04452B

Chen*, Guillaume-Gentil* et al.

Author Rebuttals to First Revision:

We would like to thank the reviewers for taking the time to carefully re-evaluate our work, their insightful comments and for acknowledging that the manuscript has been strengthened. We address the remaining technical and conceptual concerns in our detailed point-by-point reply below, striving to further clarify our conclusions regarding the novelty and uniqueness of Live-seq and thus the first transcriptome-wide demonstration of multiple scRNA-seq measurements from the same cell, as already acknowledged by the reviewers.

Referee #1 (Remarks to the Author):

Regarding point 1 from my first review, the authors have provided an improved introduction, mentioning refs 29-32 and what has been achieved before. Also, toning down some of the claims of novelty of the approach. I think the authors could still remove some of the qualitative terms of new/novel regarding the approach, as I have been thinking that their 2016 Cell paper provided this type of technical breakthrough. In the response, the authors have pointed out that they reached a 'new level', analyzing transcriptomes of relevant range, which seems to me like a fair description that should be stronger reflected in the manuscript (without the new). Regarding point 2 from my first review, the authors admit in their response that the extracted volumes represent significant percentages of the entire cell volume ("occasionally up to 100%"), but maintain, based on their data, that the approach has a high cell viability rate and that even the cell trajectories are preserved. In summary, I appreciated the care with which the authors have responded to my technical concerns and refer to the other reviewers to assess how strong the paper is within the field of transcriptomics, in light of the much lesser technical novelty and the potential invasiveness of the approach. Anyway, the authors are leaders in the use of FluidFM and have assembled a nice manuscript highlighting the state-of-the-art of the technique.

We thank the reviewer for appreciating our revision efforts. To directly state the novelty instead of naming it as such, we removed "new ground" and "novel avenue" from the first sentence of the conclusion section. It now reads (lines 535-538): "In summary, Live-seq breaks new ground by enables single cell transcriptome profiling as well as downstream molecular and functional analyses on the same cell at distinct time points (Fig. 4), providing a novel avenue unprecedented opportunities to address some of the long-standing biological questions pertaining to cell dynamics or cellular phenotypic variation...."

Referee #2 (Remarks to the Author):

In this new version of manuscript, the authors introduce several modifications to the text, figures, and data. In particular, analysis of a new Live-seq ASPC differentiation dataset is presented, as well as experimental validations of Nfkb expression and cell cycle state as predictors of LPS response in RAW 264.7 cells.

1) Overall, we find these new additions interesting and pertinent. However, we feel these new results still fall short of demonstrating Live-seq's promised potential at the level required for publication in Nature (e.g., answering a low-hanging, outstanding biological question). This seems to be a concern shared by Reviewers 1 and 3 as well.

We thank the reviewer for appreciating our revision efforts. We believe that the LPS macrophage response study featured in our manuscript demonstrates the potential of Live-seq to provide both

benchmarking and biological advances. For example, it has not previously been shown that *Nfkbia* expression is the strongest predictor of the LPS response, which is a biological finding that we have independently confirmed. The reviewer might find the insight intuitive; however, we point out that it was not self-evident. In addition, please see also our response to **point 18**, essentially illuminating why predicting LPS response variation is even today a pertinent challenge that is of interest to the field.

Many of our previous comments were addressed. Below, we highlight remaining issues by author response (initial comments and author responses in quotes), and also provide thoughts on the manuscript's new additions.

“Initial Author response: We thank the reviewer for appreciating the value of our work. The reviewer is correct, many transformative applications are now possible, even beyond the molecular recording as a novelty that we already provided in the original version. We agree with the provided, constructive suggestions and have now conducted an entirely new experiment on cell differentiation (“cells pre-differentiation and then after”), as one of the prime applications of interest. To demonstrate the potential of Live-seq in this respect, we chose an adipogenic differentiation model of primary adipose stem and progenitor cells (ASPCs) (Lines 710-725). To do so, we GFP-labeled a subpopulation of ASPCs with each cell containing a unique barcode, and biopsied them to profile their transcriptome in the pre-differentiated state. We then induced their differentiation with a chemical cocktail and sampled the same cells a second time, two days later, to profile their transcriptome in a differentiating state (see Lines 710-725, Methods(Cytoplasmic biopsies), and Fig. 4c). Using these strategies, we succeeded to sequentially sample 44 cells and obtained eight paired, quality control-passing gene expression profiles from ASPCs, as confirmed by the recovery of the correct, respective barcodes (Fig. 4e). Further monitoring of the cells for up to seven days after the second extraction revealed that cell viability was not compromised (we lost 2 cells out of 44 compared to 3 cells from 41 non-extracted control cells). In addition, we observed lipid droplets in these sequentially probed cells indicative of their retained adipogenic differentiation capacity (for representative images, see Extended Data Fig. 4b). Projection of the retrieved ASPC transcriptomes onto the integrated Live-seq and scRNA-seq data revealed the correct transition from pre- to differentiating ASPCs (Fig. 4d and Extended Data Fig. 4a). Thus, our results demonstrate that for both rapid (macrophage response to LPS) and slower (adipogenic differentiation of ASPCs) transition models, Live-seq data can be exploited to unambiguously establish the “correct” trajectory of cells that were processed using conventional scRNA-seq. This is in our opinion an unprecedented, technological achievement.”

2) We agree that the ASPC differentiation assay is a pertinent application of the Live-seq method and we understand the technical feat. However, the number of paired cells is small and the results are in line with what is expected.

We agree that the number of paired cells is small; however, the purpose of the paired transcriptomes was to show that they cluster together with state-of-the-art end point transcriptomes and that the technology is applicable beyond short term assessments of transcriptomes of individual cells. Thus, our choice fits its purpose. In fact, we would have considered it problematic if the results would have been divergent. In this regard, the “expected” is a reassurance and makes our approach trustworthy, while it is still the first demonstration of direct trajectory read-outs of differentiating cells.

3) In short, it is hard to see what Live-seq has taught us that is new. Further experiments and/or additional analyses (with validations) are needed to show that Live-seq can answer an outstanding biological question (what can you find that you could not using existing state-of-the-art methods for inference and subsequent validation). For example, are only some stem cells primed for differentiation? Is differentiation terminal for all forms of stem cells and differentiated cells? Something about the events that dictate a specific cell's decision (need to validate to establish Live-

seq does not perturb this). Repeated sampling during differentiation as opposed to simply looking at two points might yield deeper insights into the process and actually enable tracking of the events that drive transitions

We agree that, similar to the scSIAM-seq paper (Erhard et al., Nature, 2019), the focus of our study is on technological innovation including comprehensive benchmarking to demonstrate the applicability of the technology. The Live-seq approach itself is orthogonal to any other approach: *live* versus *dead*; *direct measurement* versus *statistical inference*. We show that Live-seq can be used to sequentially sample and profile the same cell twice; and we demonstrate that it can be used as a molecular recorder to couple the transcriptome-wide ground state of a cell to downstream molecular and/or phenotypic events, which has not been possible by any other method. For the latter, we further show that the method provides new gene-level, biological insights on the widely studied LPS macrophage response. The questions raised by the reviewer are indeed pertinent; however, tackling these would be a whole new research program on itself and not feasible for a revision.

4) We note that it is unclear why the authors consistently choose to analyze different systems (ASPC, RAW, IBA) together. This should be fixed, especially around lines 727-736.

We included different systems to demonstrate the capacity of Live-seq to resolve cell types and states, as it otherwise may have been criticized that Live-seq appears restricted in scope to one specific cell type. Specifically for the trajectory analyses described in lines 354-358 (linked to ED Fig. 4c,d), we chose to analyze different cell types to mimic natural systems which typically contain more than one cell type. Consequently, we do not deem this a weakness but rather a strength of our study.

Nevertheless, we understand from the reviewer's comment the request to also perform trajectory analyses on each cell type (and transition) individually, which we have now performed and show below (Rebuttal Fig. 1). For differentiating ASPCs, the pseudotime-based trajectories varied between approaches (Rebuttal Fig. 1a), while the inferred trajectory was largely consistent among distinct approaches for RAW cells (Rebuttal Fig. 1b). RNA velocity analyses with gene-structure estimate parameters revealed the correct transition of ASPCs from a pre- to post-differentiation state (Rebuttal Fig. 1c); however, other parameters did not. A similar RNA velocity analysis on RAW cells did not identify the correct transition of RAW cells responding to LPS stimulation (Rebuttal Fig. 1d). These results support our original conclusion and are consistent with our previous discussion (i.e. in the Introduction of our original and revised manuscripts) that the accuracy of trajectory inference depends on the dataset, methods, and parameter settings. As such, the results of these methods need to be interpreted as statistical expectations which may reflect different aspects of biological properties, but not necessarily the true transition path taken by the cell. Given i) that our analysis on a mixture of cell types makes a similar point, ii) that this mixture mimics natural systems more closely and iii) that trajectory analysis is already widely appreciated (and acknowledged) as an inferring rather than a direct cell state transition measurement tool, we decided not to include these "single system" trajectory analyses in the re-revised manuscript. However, we would be happy to do so if the reviewer would deem them nevertheless insightful.

Rebuttal Fig. 1. (a) Trajectory prediction of ASPCs before and after differentiation based on conventional scRNA-seq data using distinct approaches with default settings as contained in the dynverse package. (b) Trajectory analysis of RAW cells before and after LPS stimulation similar to (a). (c) Trajectory prediction of ASPC cells using the RNA velocity approach. Different strategies including “kNN pooling with gamma fit on extreme quantiles”, “Gene-relative estimate”, and “Gene-structure estimate” were used. (d) Similar to (c), but on RAW cells before and after LPS stimulation.

5) Also, please do not mix your data when reporting metrics (e.g., ED Fig 2) - it's misleading and inappropriate.

We now provide the metrics for each cell type separately (**ED Fig. 2a, d and l**), as requested by the reviewer.

6) Also, why do some of the IBA and RAW cells mix in ED Fig 2g?

In **ED Fig. 2i** (previous ED Fig. 2g), 42 out of 294 cells are misclustered, constituting a clustering accuracy of around 86%. It is well recognized that single-cell based transcriptomic data are inherently noisy. In a benchmarking study (Mereu et al. Nature Biotech, 2020), the accuracy of different scRNA-seq methods on recognizing known cell types has been shown to be variable, e.g. around 90% for Smart-seq2 and 85% for 10X Chromium. These metrics are in our opinion thus consistent with our Live-seq data, as well as with our scRNA-seq (Smart-seq2) data (**ED Fig. 2n**).

“How do assay information content and cell viability depend on sampling volume?”

Cell viability as a function of extracted volume is provided in Fig. 3a (former Extended Data Fig. 3a). Within the investigated volume range, we could not detect a dependence of the information content on the volume.”

7) This does not answer the question. How does extracted volume impact information content (i.e., genes detected, counts, etc)? Please provide a plot.

We now provide the plots requested by the reviewer (**Rebuttal Fig. 2**). We detect a weak correlation between the extracted volume and number of detected genes and counts when including all the data. This correlation was however weaker or absent when analyzing each cell type individually, likely due to the lower cell numbers. Please note that scRNA-seq approaches exhibit variability (to different extents), for example, in the number of genes detected on a per assay basis, even for "easy-to-handle cells" such as HEK293T cells and using whole cells (e.g. Fig. 2d; Mereu et al., Nature Biotech, 2020). This points to technical cDNA generation and sequencing limitations as the primary issue. In his respect, our subcellular sampling seems on par with conventional scRNA-seq approaches. We agree that an assessment of our data in relation to extracted volume is information that we should provide to the reader. To do so, we have now mapped “extracted cytoplasmic volume” as a feature on the tSNE plots of the different, sampled cell types (i.e. ASPCs, IBA and RAW cells). This did not reveal any obvious pattern or bias, suggesting that cell state (treatment) largely drives the observed clustering. We have included a short statement on these observations as well as the new plots themselves in the re-revised manuscript (**lines 255-262 and ED Fig. 3b, c**).

Rebuttal Fig. 2. The correlation (R^2 of linear regression and P value (F test)) between extracted cytoplasmic volume and either the number of detected genes (nGene) (a) or total counts (nCount) (b) for each indicated category are shown. (c) tSNE plots of distinct cell types colored by extracted cytoplasmic volume (upper panels) and cell state (lower panels). NA: not available.

“If multiple independent experiments (likely given numbers), how much of the authors’ observations are driven by experiment-to-experiment variability (e.g., differences in stimulation dynamics, cell density, etc; see Line 710)? A lot remains opaque.

While the batch effect is always a source of technical variation in single cell-based approaches, we could show that the cell identity rather than the batch is the main driver of variation (Fig. 2b and Extended Data Fig. 2f). The seamless integration of scRNA-seq and Live-seq data (Fig. 2e) further indicates that the batch effects are minor compared to the observed biological heterogeneity.”

8) We do not find the authors’ response convincing as Fig 2b,e and ED Fig 2f do not address our concern. All analyses should be done on a cell-type-by-cell-type basis. Calling variable genes across all of the systems at once is inappropriate as it masks critical differences within a system (RAW cells are very different than ASPCs) and can easily yield the results shown. NB there appears to be separation in ED Fig 2p among RAW cells. Please quantify your sources of variation, and provide appropriate statistics.

We have now analyzed the data on a cell type-by-cell type basis for the ASPC and RAW systems given that both allow us to compare (and contrast) two conditions (non-treated vs treated)(Rebuttal Figs. 3,

4). First, for both cell types, the clustering analysis matched the cell treatment and no bias due to library size, number of features or batch was observed (ED Fig. 2g, h, and Rebuttal Fig. 3; lines 172-174 of the revised manuscript). Using these normalized data, the mean gene expression levels between the scRNA-seq and Live-seq data were highly correlated for each cell type and treatment (Pearson $r > 0.95$, Rebuttal Fig. 4), as already shown in ED Fig. 2w.

Rebuttal Fig. 3. tSNE colored by the indicated parameters of RAW cells profiled with Live-seq (a) or scRNA-seq (b), or of ASPCs profiled with Live-seq (c) or scRNA-seq (d). ASPCs were processed in one batch.

Rebuttal Fig. 4. Normalized gene expression averaged across RAW (a) or ASPC (b) cells of scRNA-seq versus Live-seq data for the specified conditions.

In sum, these new analyses are consistent with our previous results. Please also see our related response to **point 10** below and note that **ED Fig. 2q** (previous ED Fig. 2p) shows the merger of Live-seq and scRNA-seq data without any batch correction. The separation of RAW cells is mainly driven by sampling methods (the right panel). This effect is largely corrected when Live-seq and scRNA-seq data are integrated (**Fig. 2f**). We hope that we have now adequately addressed the concern.

“Line 150: When and how are ERCC spike-ins added? Are they preloaded into the AFM tip or are they added during RT. This is essential to determine their utility.

We add the ERCC during the RT (see Methods for further details). Adding ERCC to the preloaded sampling buffer provides an opportunity to gage overall sample loss during the sampling process. However, as the volume of preloaded sampling buffer is variable, preloading the ERCC into the tip is less practical at the current stage.”

9) As the authors acknowledge, ERCC spike-ins at RT likely yield less reliable normalizations. Given the volume variability in preloaded sampling buffer, could UMIs be used instead? Contrary to the response to Reviewer 3, this does not reduce quality or power. This would get a more accurate estimate of the information content extracted. A few simple control experiments could go a long way here.

We use the total library size instead of ERCCs for normalization (see Methods section). Per the reviewer’s request, we have now also tested oligos that contain UMIs and cell barcodes. As shown below (**Rebuttal Fig. 5**), the cDNA yield from 5 pg total RNA is lower using UMI-containing oligos and even further reduced when both a UMI and barcode are included. This is consistent with common knowledge in the field that reverse transcription is less efficient when using oligos featuring longer, non-pairing nucleotide stretches (e.g. Sasagawa et al., *Genome Biology*, 2018), which is obviously an important issue given the very low RNA input that is typically processed with Live-seq. Nevertheless, and as acknowledged in our last revision, we agree that it would indeed be desirable to include UMIs providing the sensitivity of the RT step can be improved.

Rebuttal Fig. 5. The cDNA yield using different oligo-dT types (X-axis).

“Line 198: What is DE between Live-seq and standard scRNA-seq? How does variation change? Also, how many of the significantly DE genes in ED Fig 2q are part of the LPS response?”

We have included additional analyses in the revised manuscript to address these questions. As shown in Extended Data Fig. 2q, 72% of the Live-seq DE genes are shared with scRNA-seq and 46% of scRNA-seq DE genes are shared with Live-seq. Further analysis showed that the gene expression change among Live-seq cell populations is highly correlated with that of scRNA-seq across all sampled cell populations (Fig. 2d). For the LPS-treated RAW cells, we show in Extended Data Figs. 2h and 2i that LPS treatment-related genes are highly enriched, as expected, with around 40% of the top 100 DE genes of both Live-seq and scRNA-seq cells being directly involved (Extended Data Tables 1 and 2). We refer to this new analysis in the revised manuscript (Lines 342-345).”

10) Can you explain the discrepancy? The limited degree of overlap is troubling (suggests the technique is not representative or not minimally perturbative). Does nuclear vs cytoplasmic localization of the RNA help explain the differences? If driven by power, what happens if you downsample and bootstrap? A more detailed analysis of what is happening here on a cell-type-by-cell-type basis should be done. For example, what’s off diagonal in ED Fig 2r and Fig 2d (also, how is fold change calculated, compared to what; please make sure all of your data and legends are well referenced)? What arises when you explicitly do a differential expression analysis within group?

We agree with the reviewer that the exploration of a potential bias in Live-seq data is important. We have now performed additional analyses to address this point (Rebuttal Fig. 6, 7 and ED Fig 2s-v).

To more objectively evaluate whether the overlap shown in Fig. 2e (previous Fig. 2d) is limited, we kindly refer the reviewer to a study that has systematically evaluated different scRNA-seq methods (Mereu et al. Nature Biotechnology, 2020). We found that the correlation between our Live-seq and scRNA-seq data is comparable, if not better, to those between different scRNA-seq methods. For example, the correlation of gene expression levels between Live-seq and Smart-seq2 is > 0.95 (Rebuttal Fig. 4), whereas the correlations between different state-of-the-art scRNA-seq approaches range from 0.6 to 0.95, depending on the comparison and thus approaches. As for the overlap of DEGs across batches or methods, the same study (Mereu et al.) showed that the value ranges from 10% to 70% depending on the cell type and method, which is further supported by our own findings from a

previous study (Schwale et al., 2018, Nature, Fig. 1f), revealing around 40% overlap when comparing Fluidigm C1 and 10X scRNA-seq data aimed at profiling adipose stromal cells. Furthermore, in our review on adipose stromal cell heterogeneity (Ferrero et al., Trends in Cell Biology, 2020; Supp. Fig. 1c and i), we compared the detected DEGs from our integration of four datasets to the markers of the original publications, revealing a similar overlap range. These results support our observations regarding the comparison of datasets and approaches, and, in our opinion, show that the observed degree of overlap is acceptable.

That said, the reviewer rightfully pointed out that the limited power of Live-seq due to low RNA input amounts might explain in part the DEGs that were not detected by Live-seq. Furthermore, the reviewer highlighted that the DE analysis should also be performed by cell type. To address these requests, we first compared DEG results evaluated per cell type. Specifically, we compared non-treated versus treated cells, as performed on Live-seq or scRNA-seq data (ED Fig. 2s and Rebuttal Fig. 6a, b), revealing a reasonable logFC correlation that was comparable to that for DEGs of adipose stromal cells when comparing two distinct scRNA-seq experiments (Rebuttal Fig. 6g). We then performed the same comparison, but this time using a down-sampled version of the scRNA-seq datasets (i.e. the scRNA-seq count matrix was reduced to reach a similar mean number of genes as Live-seq data and then scaled up to keep the library size similar to that before down-sampling, please see updated Methods, lines 1032-1041). We noted that a large proportion of genes (82% for ASPCs and 78% for RAW cells) that had previously been identified as DEGs only in the original scRNA-seq datasets were no longer detected as such in the down-sampled data, implying that Live-seq has indeed reduced power compared to conventional scRNA-seq to detect DEGs, as expected (Rebuttal Fig. 6c-f). We have acknowledged this issue in the Discussion, stating that “With currently around 40% of the samples passing our data quality control criteria, an increase in mRNA detection sensitivity may further increase Live-seq’s efficiency” (lines 483-485). Nevertheless, and also as expected, some genes remained detected as DEGs by one or the other approach (Live-seq vs scRNA-seq) independent of down-sampling. We performed Gene Ontology enrichment on these genes and found a few terms that were shared by both cell types (e.g. “integral component of plasma membrane” and “plasma membrane”), but (perhaps surprisingly), only for scRNA-seq-specific differentially expressed genes. We therefore conclude that Live-seq does not appear to show any bias with respect to specific biological processes or cellular components across the distinct analyses (ED Fig. 2v, Rebuttal Fig. 7, lines 217-229).

Rebuttal Fig. 6. (a-b) Correlation of logFC between non-treated versus treated cells of scRNA-seq data vs Live-seq data (a: RAW cells, b: ASPCs). (c-d) Correlation of logFC between non-treated versus treated cells of down-sampled scRNA-seq data vs Live-seq data (c: RAW cells, d: ASPCs). (e-f) Barplot showing the number of genes shared between the different categories: None (not identified as differentially expressed (DE) genes), Both (identified as DE both in scRNA-seq and Live-seq data), Only Live-seq (only identified as DE in Live-seq), Only scRNA-seq (only identified as DE in scRNA-seq). The x-axis shows the number of genes belonging to each category derived from the comparison between Live-seq and scRNA-

seq and the colors show how these genes overlap with those derived from the comparison between Live-seq and down-sampled scRNA-seq. **(g)** Correlation of the logFC between subpopulations of mouse adipose stromal populations (Adipose Stem Cells (ASCs) versus Adipogenesis regulatory cells (Aregs), ASCs versus PreAdipocytes (PreAs) and PreAs versus Aregs) of two published datasets (Schwalie et al., 2018, Nature; Merrick et al., 2019, Science). The clustering and subpopulations are described in our review on the heterogeneity of adipose stromal cells (Ferrero et al., TCB, 2020).

Rebuttal Fig. 7. Biological process (BP) and Cellular Component (CC) GO enrichment for genes detected as DE either using Live-seq or conventional scRNA-seq for both ASPCs and RAW cells. The size of each red circle also denotes the significance.

11) Relatedly, in Fig 3e, how variable is the sampling time for each cell given the time it takes to run Live-seq - i.e., what does 1h and 4h actually represent? Relatedly, what these data show is recovery, not lack of perturbation contrary to the assertions in lines 469-478.

Related to Figure 3e, the cells were extracted during a 1h time-window, before processing for scRNA-seq 1 or 4h later. 1h and 4h thus represent 1 ± 0.5 and 4 ± 0.5 h after extraction. We have adjusted the text in the Methods (lines 860-866) to further clarify.

Whether the obtained data show a “lack of perturbation” or “recovery” is a rather semantic discussion in our opinion; most importantly, the data show that there are very few expression differences between extracted (1h and 4h earlier) and non-extracted cells, indicating that cells do not seem to be majorly affected by Live-seq sampling. This is why we prefer to use the term “lack of perturbation” over “recovery”.

12) Also, a direct comparison of the Live-seq and single-cell RNA-seq metrics (ED Fig 2j) would be an important addition.

We agree and have now added a direct comparison (ED Fig. 2l), as suggested by the reviewer.

“Line 215: How does are cell volume, extracted volume and viability related? Co- dependence isn’t plotted. A more quantitative analysis per ED Fig. 3e would be helpful.

The general co-dependence between the extracted volume and the viability is shown in Fig. 3a. In most cases, while the extracted volume can be reliably measured from microscopy images of the microchannel after extraction, the volume of a particular cell undergoing extraction is unknown. Adherent cells indeed require dissociation to assume a spherical morphology and enable an estimation of their volume, which prevents measuring their volume before and after extraction. In the case of the semi-adherent RAW cells however, the measurement became feasible, as shown in former Extended Data Fig. 3d, e. We have now extended these data (Fig. 3b, Extended Data Fig. 3a and 3b) to clarify these results. We implemented the volume growth profiles of 4 control cells without LPS stimulation, 4 control cells with LPS stimulation, and 4 additional extracted cells stimulated with LPS (12 profiles of extracted cells in total). Such longitudinal volume growth was not measured for dead cells, as those did not present a spherical morphology as required for the measurement. Please see also our response to reviewer #1, comment 2) and the response to the next comment.”

13) We think a plot showing viability as a function of cell volume and extracted volume would be useful to the reader to understand if these are related.

We have now modified Fig. 3a and the main text (lines 249-255) to improve clarity and show the relation between the extracted volume and the cell viability. Specifically, Fig. 3a (and Rebuttal Fig. 8 below) shows the cell viability as a function of the volume extracted (violin plots instead of former distribution plots). Cell viability did not scale with the extracted volume, which itself ranged from 0.2 to 3.5 pL with an average of 1.1 pL (two-sided Wilcoxon rank-sum test: $P = 0.44$, 0.20, and 0.18 for ASPC, IBA and RAW cells, respectively). ED Fig. 3a (formerly included in Fig. 3a) shows the indicative cell volume distribution for the three different cell types (measurements of the volumes of dissociated cells, independent of Live-seq sampling experiments). It shows considerable variation in cell volume between the three different cell types, as well as within a given cell type. The cell volume before Live-seq sampling is generally unknown, at the exception of the few RAW cells for which we measured longitudinal volume profiles (Fig. 3b, live RAW cells).

Rebuttal Fig. 8. Cell status (dead or alive) for distinct cell types in function of extracted cytoplasmic volume. $P = 0.44, 0.20, 0.18$ for ASPC, IBA and RAW cells (two-sided Wilcoxon rank-sum test), respectively.

“Lines 271-304: First, it is unclear why baseline RNA expression (rather than protein abundance) should be predictive (relatedly, how much does gene expression change in the 30 minutes between sampling and stimulation (Line 566))? Second, there is no validation. And, third, as highlighted, several of the hits are expected. In short, it is unclear how Live-seq uniquely enabled new biological insights here. This is the biggest shortcoming of the paper. Ditto Lines 306-325.

*We agree with the reviewer that it is unclear whether baseline RNA expression can be predictive of the downstream phenotype or not. This is a fundamentally unsolved question in many biological systems, largely because there is no transcriptome-wide molecular recording method that links the baseline RNA profile with downstream phenotypes. We hope that the reviewer will appreciate that this is exactly what Live-seq aims for. Furthermore, and as indicated above, we intentionally studied LPS macrophage response heterogeneity, since this system has been well described, allowing us to benchmark Live-seq. We thus agree and also expected to “rediscover” genes described in the context of LPS macrophage response and find this reassuring and important. Regarding new candidate genes, we found that RAW cells in S phase respond weaker to LPS. We validated this interesting finding experimentally. Our results also point to *NFKBIA* as the principal baseline determinant of downstream heterogeneity. The uniqueness here is that Live-seq allowed us to generate a transcriptome-wide gene ranking, which cannot be derived based on targeted approaches. Moreover, even conventional scRNA-seq does not support such inference, as shown in Fig. 4h, i. To further strengthen the finding that *Nfkb1a* is a major determinant, we generated a RAW-G9 line containing a BFP reporter under the control of the *Nfkb1a* promoter and were able to confirm that BFP fluorescence is induced by LPS treatment like the endogenous *Nfkb1a* expression, synchronously with the induction of the *Tnf-mCherry* reporter (Extended Data Fig. 4j). The use of the reporter line allowed us to demonstrate that, as hypothesized, basal *Nfkb1a*-BFP intensity is a negative predictor ($R^2 = 0.12, P = 0.003, F$ test) of the rate of *Tnf-mCherry* intensity increase (the slope) (Fig. 4j). We have included this additional validation in the revised manuscript (Lines 827-834) and also expanded the Methods section (*Nfkb1a* reporter analyses).”*

14) Does baseline RNA expression correlate with protein abundance? This would provide support to findings of correlation between RNA expression of some genes and response to LPS. The data sets in Jovanovic et al, Science 2015 (DOI: 10.1126/science.1259038) should be instructive.

Thank you for this interesting suggestion. We looked into the Jovanovic et al. (Science, 2015) work where the authors studied the relationship between protein abundance and mRNA expression, translation rates and degradation rates in mouse dendritic cells, both at baseline and after stimulation with LPS. The authors found that, at baseline before stimulation with LPS, mRNA explained 66% - 68% percent of protein abundance, compared to only 21%-26% of variance explained by translation rates and 8%-11% by degradation rates. The authors also provided per-gene model parameters for the contribution of individual regulatory mechanisms to the respective protein levels. Although there are limitations when comparing these gene-wise parameters at the bulk level to Live-seq single cell data,

we looked into the raw data provided by Jovanovic et al.; however, *Nfkb1a*, which we identified as the strongest negative predictor of LPS-induced *Tnf* expression, was not included in the dataset.

The dominant role of mRNA expression in determining protein levels at steady state and the fact that not all functional RNAs encode proteins (e.g. lncRNA, endogenous microRNA sponges) makes it plausible, in our opinion, to find a predictive role of gene expression at baseline for response heterogeneity upon exposure to LPS. We have included a statement emphasizing this point in the revised manuscript (discussion) (lines 521-523).

15) Regarding the difference in LPS response by cell cycle phase, this too is expected. Please see Allen et al, Science Signaling 2019 (DOI: 10.1126/scisignal.aau1851).

We double-checked this paper. The study revealed that the circadian cycle has an impact on LPS macrophage response, yet the cell cycle was not studied.

16) Lines 804-807: Adding the LPS data to this plot and comparing to pre-LPS Live-seq is inappropriate. There doesn't seem to be a positive correlation between *Tnf* and *Nfkb1a* at baseline (driven by the LPS stimulation).

We color-coded the plot (Figure 4i) to provide more information, so that the readers can choose their own point-of-view. We have now included an analysis on each condition in ED Fig. 4i, which shows a positive correlation between *Tnf* and *Nfkb1a* expression at baseline ($R^2 = 0.23$, $P = 1.4e-07$, F test).

17) Regarding the *Nfkb1a*-BFP/*Tnf*-mCherry validation experiment, the correlation plot would benefit from the indication of an R-squared, Spearman correlation coefficient, and associated p-value. Given that the association is not strong, it is our impression that the pattern is driven mostly by few datapoints with high basal BFP intensity. These results would be more convincing if additional datapoints could be obtained, with more points with high intensity.

As per the reviewer's request, we have now performed another independent experiment that included more observed cells. Consistent with our previous results (now in ED Fig. 4l), the basal *Nfkb1a*-BFP reporter negatively predicts the rate of *Tnf*-mCherry intensity increase (Fig. 4j, $R^2 = 0.11$, $P = 0.0008$, F-test, Pearson's $r = -0.34$, $P = 0.0008$).

18) Finally, we agree with the authors that the anticorrelation of *Nfkb1a* basal expression and LPS response/NFkB upregulation is an expected finding. In short, this is a validation of the method as pointed out above, rather than an indication of the full potential of Live-seq. The lack of new biological insights remains the biggest shortcoming of the paper.

Please note that we have not claimed that this anticorrelation was expected. There are many genes involved in the LPS-induced inflammatory response (a few are highlighted in ED Fig. 4j). Accordingly, all would be "expected" to contribute to LPS response variation. Based on our Live-seq data, we are now able to show that, except for *Nfkb1a*, none of the other "expected" factors contribute to the phenotypic outcome, at least to an extent that is comparable to that of *Nfkb1a* (i.e. the latter gene tops the transcriptome-wide list). Interestingly, the paper referred to by the reviewer above (Allen et al. Science Signaling, 2019) aimed to better understand what the historical molecular events are that determine downstream LPS response phenotypic heterogeneity. Specifically, as stated in their study, the authors "sought to identify heterogeneity in the innate immune response among a phenotypically homogeneous population" (i.e. bone marrow-derived macrophages, ed.), implying that this is still an unresolved question of interest to the field. Moreover, and importantly, Allen and colleagues did so in a trial-and-error fashion, whereas Live-seq revealed targets in an unbiased, transcriptome-wide manner.

*“Sequential Live-seq cell sampling to measure cellular dynamics
Line 327-355: This analysis/section is dramatically underpowered and reads as preliminary. Dynamics are highlighted in the introduction and motivation as a key selling point but there are only 2/14 cells whose repeated sampling passed filtering. This yield seems like a flag and directly undermines the major selling points.*

We have substantially restructured the manuscript due to the addition of a new section on cell state transitions as a consequence of cell differentiation (Fig. 4c-e). While it is correct that we still have relatively few cell pairs (four for macrophages (two in the original manuscript) and eight for ASPCs), in our opinion, our results with the present throughput indicate that it is often already sufficient to probe the cell using Live-seq just once as phenotypes of the very same cell can be observed later. A second Live-seq sampling from the same cell is only desirable when further probing of the cell is required (e.g. over a longer timeframe) and was included here as a proof of concept. In addition, our sequential Live-seq data demonstrate that the probing of relatively few cells can already contribute to resolving complex biological processes in ways that snapshot scRNA-seq data cannot. These include the empirical determination of trajectories through Live-seq and conventional scRNA-seq data integration as well as the prediction of heterogeneous phenotypic behavior. Thus, we deem conventional scRNA-seq and Live-seq to be highly complementary in that scRNA-seq can define the manifold in high-dimensional space, while Live-seq provides guidance with respect to the information flow within that space. For such a hybrid approach, the monitoring of fewer cells is possible and therefore fully aligned with Live-seq’s current throughput. However, we acknowledge that technological innovations will still be required to standardize such analyses and perform them at a larger scale (Lines 816-819, 878-885, and 967-969).”

19) We understand that sequential sampling of the same cells favorably impacts type 1 and type 2 error allowing for smaller sample sizes. However, given stochastic, inherent variability in biological processes, a sample size of 4 appears small. Statistics should be used to guide the validity of all results discussed.

The purpose of these experiments was to demonstrate that Live-seq can be applied over shorter and longer time periods. We used two cell systems at two different time scales and demonstrate the wide applicability of Live-seq for sequential sampling, which resulted in four pairs of RAW cells and eight pairs of ASPCs. The cluster analysis validated the approach.

20) A scRNA-seq / Live hybrid approach appears promising. Could it be a way to obtain new biological insights with the existing data? We understand the conceptual promise of the approach but are having trouble seeing its proof.

We fully agree with the reviewer that a scRNA-seq / Live-seq hybrid approach is promising. In fact, we believe that we have demonstrated this with our study on macrophage LPS response heterogeneity. We mention this strategy in the Discussion (lines 492-502). We deem this an important strategic route going forward to explore uncharacterized systems.

“Minor point 2. It’s unclear from what is provided if the full transcriptome can be sampled. We might adjust the title based on a more nuanced analysis of ED Fig. 2q.

While the number of detected genes by Live-seq is lower than that by conventional scRNA-seq, as expected given the lower RNA input (shown in Fig. 3a), the seamless integration of Live-seq with canonical scRNA-seq data supports the conclusion that Live-seq comprehensively samples the transcriptome of individual cells (Fig. 2e). Nevertheless, and as the reviewer suggested, we have performed additional analyses to directly compare Live-seq and scRNA-seq data in the revised

manuscript (Fig. 2d and Extended Data Fig. 2q-r). Taken together, we believe that these data validate the quality of Live-seq data.”

21) These comparisons are instructive. Our comment refers to the potential spatial biases in RNA of different genes given that biopsies are cytoplasmic. Is the difference between DEG in Live-seq and scRNA-seq belong to a certain category of genes (e.g., nuclear vs not nuclear) or are they purely a consequence of sampling rate? This would help delineate how Live-seq signal can be interpreted.

For example, what genes are sampled in 1b relative to matched whole cells? How do the metrics compare? Ditto 2a. Fig 3 hints at this in one systems for cells at rest.

Similarly, can you show matched comparisons of Live-seq and total cell RNA-seq for ED Fig 2j? For Fig 2d/ED Fig 2r, we'd like to see enrichments of what's off-diagonal and square axes.

We have now updated ED Fig. 2l (previous ED Fig. 2j) to include a direct metric comparison between scRNA-seq and Live-seq. As expected, due to the smaller sampling volume and as shown before, Live-seq detects in general less genes compared to scRNA-seq.

With respect to the other questions, we kindly refer the reviewer to our answer to point 10, as well ED Figs. 2s-v and lines 217-229 in the re-revised manuscript, which revealed no particular biases with respect to specific biological processes or cellular components for genes that were detected only by Live-seq (ED Fig. 2v, Rebuttal Fig. 7).

Referee #3 (Remarks to the Author):

The authors have addressed many of the comments from the initial round, and the manuscript has improved. It is clear that Live-seq can accurately sample cytoplasm from cells, without detrimental damage to cells, and that the sampled RNAs can provide a decent transcriptional profile that e.g. identifies correct cell types (i.e. large biological differences). To this end, it is the first transcriptome-wide demonstration of multiple scRNA-seq measurement from the same cell (Figure 2 – the strongest figure of the manuscript).

We thank the reviewer for the appraisal of our work.

1) However, the biological experiments in this manuscript are still weak. The one insight generated from the LPS heterogeneity experiment was the identification of *Nfkbia* (Fig 4h, non-significant) that was validated with borderline significance (dependent on a few outlier observations, Fig 4j), and the S-phase effect.

We agree with the reviewer that these are new insights on a system that has already been well studied, allowing first of all to benchmark the data obtained with Live-seq and secondly to contribute to biological advances. Nevertheless, to strengthen the validation of *Nfkbia* expression in predicting the LPS-induced response, we have now conducted another independent experiment that included more observed cells. Consistent with our previous results (now in ED Fig. 4l), the basal *Nfkbia*-BFP reporter negatively predicts the rate of *Tnf*-mCherry intensity increase (Fig. 4j, $R^2 = 0.11$, $P = 0.0008$, F-test; Pearson's $r = -0.34$, $P = 0.0008$).

2) The new analysis (Fig 3e) that investigated the effect of cytoplasm sampling on the cell is confusing. It seems very dangerous to here claim negative results – i.e. that the cells do not separate in tSNE - since often proper separation depend on accurate identification of the biologically variable genes (tSNE parameters etc). With 18 DE genes, in my experience, the cells would be highly likely to cluster.

The reviewer makes a valuable point and we agree that the data should be interpreted with care. This is why we now state in our manuscript: “Thus, these experiments suggest that Live-seq does not

induce major, short-term gene expression alterations”. We believe that this is still an accurate representation of our data, since, using the most sensitive scRNA-seq method, Smart-seq2, we detected only 12 (not 18) genes that exhibited mild differential expression, which did not impose on cell clustering. Compared to the vast effects observed upon tissue dissociation (e.g. van den Brink et al., *Nature Methods*, 2017 or Denisenko et al., *Genome Biology*, 2020), we hope that the reviewer will agree that the impact of live cell sampling appears modest, even though it is indeed a surprising result. However, the molecular data is consistent with our phenotypic data, which provides further support to our claim that Live-seq’s impact is not major. Nevertheless, we agree that the tSNE Figure panel might confuse the reader (who might try to find a pattern while there is none) and we therefore moved it to the Extended Data section (ED Fig. 3h).

3) The discussion on metabolic labeling still misses the main point – in my opinion – that the real advantage of Live-seq lies in phenotypically linking cells over larger time intervals (>8 hours to days), as metabolic labeling will excel at shorter time points. Thus, the genes detected as differentially expressed (Fig 3f) are limited to those initially expressed at low (or no) levels, in order to be detected as differentially expressed in Live-seq (whereas metabolic labeling could detect all differentially expressed genes).

To briefly clarify, Fig. 3e (previous Fig. 3f) shows differential gene expression as measured by Smart-seq2 (not Live-seq) and is intended to show the possible molecular impact of live cell sampling.

We thank the reviewer for the appreciation of the novelty of our approach. As for detecting differentially expressed genes across time points, we see no conceptual reason why Live-seq would be restricted to genes that are very lowly or not expressed in the original cells, given that we show that Live-seq’s transcriptomes act as suitable representations of full cell transcriptomes and would thus be subject to the same advantages and restrictions. Indeed, supported by new analyses and detailed in our re-revised manuscript (lines 217-229), we show that the differentially expressed genes derived from the Live-seq and scRNA-seq datasets overlap to a large extent (Extended Data Fig. 2r), supported by a high correlation in fold changes when comparing each cell state to the rest of all the cells (Fig. 2d) and within each cell type (Extended Data Fig. 2s). However, compared to Live-seq, scRNA-seq yielded a larger number of differentially expressed genes that were not detected by Live-seq. Down-sampling the scRNA-seq data to a library complexity that was similar to that of Live-seq reduced the number of differentially expressed genes that were detected by scRNA-seq only (Extended Data Fig. 2t, u). These findings suggest that, as expected, scRNA-seq - at present and with current state-of-the-art RNA detection capacities - provides greater power than Live-seq. We have acknowledged this issue in the Discussion, stating that “With currently around 40% of the samples passing our data quality control criteria, an increase in mRNA detection sensitivity may further increase Live-seq’s efficiency” (lines 483-485). Finally, GO analysis on the differentially expressed genes found a few terms that were shared by both cell types (e.g. “integral component of plasma membrane” and “plasma membrane”), but (perhaps surprisingly), only for scRNA-seq-specific differentially expressed genes. We therefore conclude that Live-seq does not appear to show any bias with respect to specific biological processes or cellular components across the distinct analyses (ED Fig. 2v, Rebuttal Fig. 7, lines 217-229).

We agree with the reviewer that it will help the reader to place Live-seq in the context of the strengths and weaknesses of metabolic labeling. As argued earlier, we deem Live-seq truly novel in that it preserves cell viability and thus allows cells to be continuously monitored (e.g. for live cell imaging-based phenotyping) or even sampled again to profile pre- and post-transition transcriptomes on the same cell, both of which we show in our manuscript. This live cell preservation provides unique opportunities to link the molecular ground state of a cell to downstream molecular, cellular or functional properties, while metabolic labeling approaches are by definition restricted to transcriptomic changes.

As the reviewer points out, an unmatched advantage of Live-seq lies in phenotypically linking cells over larger time intervals. Here, metabolic labelling is not applicable. Regarding the investigation of short-term transcriptomic dynamics, both approaches might be perceived as alternative approaches. However, we would like to argue that even for short time scales (<8h), Live-seq is superior over metabolic labeling in characterizing the original state of the cell. This is because Live-seq provides directly measured expression data, whereas metabolic labeling approaches need to infer expression levels, given that they all still require cell lysis and can thus probe each cell only once. This results in ambiguous, difficult-to-interpret results, even at shorter time scales than 8 hours, in particular given the short half-life of many transcripts. In brief, the cells are lysed at one time point and the transcripts of an earlier time point need to be approximated because, while mRNA synthesis can be determined using metabolic labeling, mRNA degradation cannot. The latter issue is bypassed by assuming constant (across time) and homogenous (across cells) degradation rates. This explains why metabolic labeling may work for shorter time scales, since over longer ones, it would be very difficult to estimate degradation dynamics. However, mRNA degradation is dynamically regulated, even at short time scales, and depends on the biological system as well as on the presence of specific mRNA stability-controlling elements (e.g. AU-rich elements), which can be affected by exogenous stimuli (Schoenberg and Maquat, *Nature Rev. Genet.*, 2012). For example, in the pioneering study by the Regev lab which introduced metabolic labeling for bulk transcriptomics (Rabani et al., *Nature Biotech.*, 2011), the authors used an LPS-stimulated mouse dendritic cell model (comparable to some extent to the LPS-stimulated RAW cells used in our study) and found that the constant degradation hypothesis needed to be rejected for 6% of the genes. Importantly, the latter genes, which do not follow the assumption of constant degradation, were enriched for inflammatory and immune signaling functions and NF κ B signaling targets. This suggests that inferring their past expression levels by metabolic labeling is bound to be error-prone. This concern thus also applies to *Nfkb1a*, which we identified in our study and is a direct NF κ B target. Thus, using Live-seq, we were able to bypass this issue by *directly* measuring its expression level in the ground state, without the need to *infer* it. This example provides another important conceptual difference between Live-seq and metabolic labeling. In our view, this shows that Live-seq offers advantages that are relevant also for shorter time periods (<8 h). We have now updated the discussion in **lines 471-483**.

4) On this topic, the LPS experiment would have much higher power if studied with metabolic labeling instead of Live-seq.

We would like to emphasize that we were able to couple a transcriptional read out to a phenotypic one. It is doubtful that we would have identified *Nfkb1a* by metabolic labeling for the reasons outlined above. Indeed, particularly unstable transcripts might be crucial to drive cells into divergent populations. The use of inference rather than direct measurement methods could therefore represent a conceptual obstacle for future advances and, in our view, warrants exploration of novel and complementary approaches.

However, to avoid making this argument purely hypothetical, we now went one step further and investigated how we could use metabolic labeling data to explore how the initial transcriptomic state of a cell determines the magnitude of its immune response, as we did with Live-seq for macrophages responding to LPS. We first assessed whether this could be possible theoretically, using a set of ordinary differential equations that model the dynamics of labelled and unlabeled mRNA, and how they are affected by degradation and initial state. Based on that, we observed that it is indeed possible to infer either a difference in initial state or in degradation rate between cells, but not both. Importantly, these same conceptual limitations emerged when analyzing the data from the scSLAM-seq paper (Erhard et al., 2019). This is relevant in the context of our work given that in this paper, mouse fibroblasts were infected with mCMV after which new and old mRNA levels were determined after 2 hours. In other words, when a change in response is observed, we found that it is not possible

using metabolic labeling data to see whether this is due to differences in degradation or differences in initial state. Finally, we also explored additional challenges with metabolic labeling data (e.g. the requirement to estimate the detection rate of newly labeled mRNA), which, we found, also impact the ability of metabolic approaches to make unambiguous and robust gene-level predictions on the phenotypic behavior of individual cells.

Taken together, our analyses highlight that metabolic labeling approaches infer rather than directly observe the original molecular state, as provided by Live-seq. That said, Live-seq also makes an assumption, namely that the extraction of a cell's cytoplasm does not perturb its state or phenotype (note that metabolic labeling actually also assumes that the label has no impact on mRNA stability, which, especially for transcripts with a high number of "U"s or short-lived transcripts, may not be the case (Schott et al., *Nature Methods*, 2021)). To address Live-seq's potential perturbation caveat, we directly measured the impact of the sampling itself and found only a few genes being differentially expressed after extraction. Furthermore, and importantly, this assumption can be validated by a user of Live-seq when probing different cell types or systems. Please note that we have briefly summarized the results of our analyses and thus our responses to points 3 and 4 by the reviewer in our revised Discussion (lines 471-483).

Altogether, I applaud the authors efforts to establish this technology, but I am worried that at its current power and cellular throughput, there will be limited interest to establish this technology in other labs. Nevertheless, I could see increased interest in Live-seq if the method could be scaled to hundreds of cells (within a reasonable time frame), although I completely understand this is currently not achievable. The rather weak biological insights provided in the manuscript makes it still less compelling for Nature.

Please note that in our study, we chose to investigate several cell types to demonstrate broad applicability, since, contrary to metabolic labeling, live cell transcriptomic profiling had not been performed before on any cell type. It is evident that the throughput can be substantially increased when focusing on one cell type/system, as the set-up of automated protocols is readily feasible in future studies. We thus agree that there is room for higher throughput and we state this openly in the discussion. Furthermore, we benchmarked our study with the scSLAM-seq paper, which was published in Nature (2019). We and many others certainly consider this study groundbreaking, since the authors demonstrate that their metabolic labeling approach for bulk RNA-seq, "SLAM-seq" (Herzog et al., *Nature Methods*, 2017), can also be used to analyze single cells. To demonstrate this, the authors profiled 107 (compared to 294 Live-seq transcriptomes) single mouse fibroblast cells (of which 94 were functional) infected with mouse cytomegalovirus to study the onset of infection with lytic virus. Both in number of profiled cells and scope, our manuscript seems therefore at least at the same level, if not larger and broader. Moreover, the scSLAM-seq paper remained at the level of processes in terms of infection response modulation and did not explore individual gene determinants. We can only speculate that, as argued above (points 3 and 4), this might be due to technical limitations including sensitivity, throughput and general issues with correctly inferring the "past" transcriptome of cells prior to infection. Why this nevertheless was compelling to Nature is (in our opinion), because it provided a clear proof-of-concept for a novel, powerful approach with, providing further technological and analytical improvements, clear downstream applications. We hope that the reviewer will appreciate that our Live-seq study is in this regard highly complementary to the scSLAM-seq paper and we therefore hope that it will be evaluated accordingly.

Minor issues:

I think certain analysis in the manuscript are rather biased towards promoting Live-seq. For example, the comparisons with RNA-velocity like trajectory inferences all use inference tools that will connect

all cell types, although the experiment performed is using two different experimental model systems. Proper use of inference tools on each experiment alone would likely provide the correct flow.

The reviewer points out a fundamental issue of trajectory inference methods: they explicitly look for connections between any cell type that is provided to them, without necessarily reflecting true biological dynamics or relevance. Many natural systems consist of several different cell types with different lineages/origins. We therefore believe that the analysis, as shown, correctly conveys the limitation of trajectory inference compared to the type of data that Live-seq can provide. As the reviewer also points out, inference tools need prior information before they can be applied. In contrast, Live-seq's measurements are direct and thus are, we are convinced, complementary.

To address the concern more directly, we have now also performed trajectory analyses on each cell type (and transition) individually. For differentiating ASPCs, the pseudotime-based trajectories varied between approaches (**Rebuttal Fig. 1a**), while the inferred trajectory was largely consistent among distinct approaches for RAW cells (**Rebuttal Fig. 1b**). RNA velocity analyses with gene-structure estimate parameters revealed the correct transition of ASPCs from a pre- to post-differentiation state (**Rebuttal Fig. 1c**); however, other parameters did not. A similar RNA velocity analysis on RAW cells did not identify the correct transition of RAW cells responding to LPS stimulation (**Rebuttal Fig. 1d**). These results support our original conclusion and are consistent with our previous discussion (i.e. in the Introduction of our original and revised manuscripts) that the accuracy of trajectory inference depends on the dataset, methods, and parameter settings. As such, the results of these methods need to be interpreted as statistical expectations which may reflect different aspects of biological properties, but not necessarily the true transition path taken by the cell. Given i) that our analysis on a mixture of cell types makes a similar point (**lines 354-358** (linked to **ED Fig. 4c,d**)), ii) that this mixture mimics natural systems more closely and iii) that trajectory analysis is already widely appreciated (and acknowledged) as an inferring rather than a direct cell state transition measurement tool, we decided not to include these "single system" trajectory analyses in the re-revised manuscript. However, we would be happy to do so if the reviewer would deem them nevertheless insightful for the reader.

Figure 1b. The authors describe these QC as stringent, although they seem lenient?

Please note that the metrics on all cells were shown without any filter in **Figure 1b**. We thus speculate that the reviewer is referring to our statement in **line 720** of the previous manuscript where we mention "stringent" QC filtering. We feel that the definition of "stringent" versus "lenient" in the scRNA-seq field is ambiguous, depending on the used systems, scores etc. To avoid any preconception, we decided to remove the term "stringent" and refer directly to the **Methods** section which contains all the details on how the QC was performed (**lines 948-952**).

Ext Data Figure 2q: The number of differentially expressed genes are more than the number of genes identified in the cell types? Must be a typo somewhere, or erroneous gene set summation.

The provided number referred to all the five cell types together. We agree that this would have confused the reader and we now provide the numbers for each cell type separately (**ED Fig. 2r**, previous **ED Fig. 2q**).

Ext Data Figure 1b,g: show increased cDNA yields without RNA input after optimization? In fact, same cDNA yields were obtained from 0 as with 1 pg of RNA? In general, optimizations were done with very few cells per condition (2 to 3 cells per condition, that does not sound very robust).

We have specifically addressed this issue in **ED Fig. 1j, k**. Specifically, we observed that the reads from the negative control (without input RNA) are overrepresented by poly A and TSO sequence stretches (**ED Fig. 1j**) and that they map to only few genes (**ED Fig. 1k**). While three cells per condition in one experiment were shown, we performed between two to five distinct experiments per assay, yielding consistent results. We kindly refer the reader to the legend of **ED Fig. 1**.

Details on how the short time-period experiments are performed is still lacking. For 1-2 hour treatment experiments, are individual cells sampled and treated alone? And the experimental setup repeated n times (every time stimulating one sampled cell?).

We have amended the **Methods** section to improve clarity. The details for Live-seq experiments with 4 hours LPS treatment are described in the **Methods** section **lines 637-640** and **697-704**; details for scRNA-seq experiments of cells that underwent Live-seq sampling 1 hour and 4 hours earlier are described in the **Methods** section **lines 862-866**.

All figures still look preliminary and I agree with reviewer 1 that the organization of results in main and extended data figures could be much improved.

We have added, polished and re-arranged the Figures to better reflect the main messages of our study. For example, we have added new figures in **ED Fig. 2g, 2h, 2t, 2u, 2v, 3a, 3b, 4i, 4l**, re-arranged **Fig. 3a** into **Fig. 3a** and **ED fig. 3a**, moved **Fig. 3e** to **ED Fig. 3g**, and updated **Fig. 4h, 4i, 4j, 4k** and **ED Fig. 2a, 2d, 2l, 2m, 2s, 4h,4m**. Please note that reviewer 1's comment was on the original manuscript, not on the revised version, which was already significantly modified.

Reviewer Reports on the Second Revision:

Referees' comments:

Referee #2 (Remarks to the Author):

In this new version of the manuscript, the authors include modifications to their analyses, addition of statistics, an expanded TNF-mCherry experiment, and some clarifications in the text. While some of our concerns are addressed, the lack of novel biological insight remains a significant shortcoming of the manuscript. We suggest, once again, that the authors add an entirely new final section to their paper, answering one of the multitudes of impactful biological questions that could benefit from their technology. As pointed out by reviewer 3, this should perhaps involve sequential sampling over longer periods of time (days) to make full use of the technology over metabolic labeling. Short of this, could a different journal be a better fit, for example Nature Methods or Nature Biotechnology?

We respond to specific comments below:

Regarding Point 1. As pointed out in the manuscript (line 405), NFKBIA is a known inhibitor of NFkB, and NFkB is linked to TNF expression. Therefore, it appears to us that the anticorrelation of the baseline NFKBIA and the TNF response is not a surprising finding but rather a confirmation of the expected finding and a validation of the method (which the authors acknowledge in their comments). Indeed it is interesting that the inhibitory effect of NFKBIA on NFkB can be observed unbiasedly at a single-cell level, but we believe our understanding of the NFkB signaling pathway or its heterogeneity in cells has not changed.

Regarding Point 2. Given that longitudinal sampling is live-seq's biggest innovation, paired cells should perhaps be at the crux of the manuscript. From that lens, the few cells that have paired measurements are essentially the only cells that were fully assessed by live-seq; this limits the conclusions that can be drawn from the biology that is observed. From this point of view, this further highlights the secondary role of biological observations in this manuscript.

Regarding Point 3. While we appreciate the potential of the method and the comparison to scSLAM-seq (Erhard et al., Nature, 2019), we maintain that a conclusive example of how it can answer an outstanding biological question is missing for publication in this journal, especially given the authors' previous Cell paper.

Regarding Point 4. Analyses performed separately by system should be the ones included in the manuscript. Also, for RNA velocity analyses, the genes chosen as input have a large impact on the results. Only genes relevant for a given system and its process of interest should be chosen – consider describing how this is done. Combining all systems together for gene selection and beyond has a strong potential for bias.

Regarding Point 6. Clustering accuracy depends on cell types measured. For example, in Mereu et al (Nat Biotech 2020), HEK cells clustered separately from immune cells more readily than did different types of immune cells to each other. We would expect RAW and IBA to readily cluster separately given their drastic difference in transcriptome. Could a comparison to the clustering obtained from the scRNAseq be done?

Regarding Point 9. We suggest the authors look to SMART-Seq3 (<https://www.nature.com/articles/s41587-020-0497-0>) for guidance.

Regarding Point 10. How does downsampling improve the overlap between scRNAseq and Live-

seq? Could the authors quantify this?

Regarding Point 19. Consider specifying this is the manuscript – it was not clear to us that the goal of sequential sampling here was purely demonstrative given that this is one of the promises of the technique.

Referee #3 (Remarks to the Author):

The authors have improved the Nfkbia validation experiment with a second experiment that show more convincing significance. I would advise the authors, however, to tone down the language around this observation. Currently, the abstract list Nfkbia as a “major phenotypic determinant” and the main text says “principal driver” and “negative predictor”. Yet, the R2 of 0.1 is a very weak, although significant, interaction.

Metabolic labeling vs. Live-seq

I will spell out my point here – take it or leave it – I still think the current discussion (and rebuttal text) is highly misleading and should be corrected.

If a researcher would like to study the effect of a perturbation on the transcriptome, say the direct effects of LPS on gene expression. A Live-seq user would then sample a part of the cytoplasm before the perturbation (timepoint t0) and then collect the cell or cytoplasm after 1 hour exposure to the perturbation (timepoint t1). The problem here is that the 1 hour timepoint would be highly confounded by RNAs present before the perturbation. In fact, in a typical in vitro grown cell only 5% of the transcriptome is transcribed within the last 1hr, making 95% of the RNA profiled at timepoint t1 being actually the remaining pre-existing RNA at timepoint t0 that was not sampled in the first Live-seq experiment. Therefore, the power to detect a change at 1hr relative to t0 will be highly limited. Most power would lie in identifying genes that were not expressed/present at t0 as they would have expression at t1 that is clearly detectable and different from t0. However, if a gene was abundant at t0 and say 2-fold induced at t1, that signal would be lost due to fact that a 2-fold change occurring in only 5% of the RNA would be diluted away. Therefore, I withhold that Live-seq will have limited power to detect shorter-timescales direct effects of perturbations onto gene expression.

A metabolic labeling user would provide the cells 4sU/EdU at t0 together with the perturbation, and collect the cells at t1 for analysis. Comparing separate cells that were exposed to perturbation or not, the analysis can focus on the RNAs transcribed only since the perturbation therefore avoiding the dilution pitfalls of the Live-seq scenario above. Therefore, differential expression can be detected even for genes that were highly expressed at t0 in contrast to Live-seq. Therefore, metabolic labeling will have drastically improved power for any short-term experiment that aim to identify the direct effects of a perturbation on transcription. Here, Live-seq (like a standard total RNA comparison on unmatched samples) will have low power to find any difference) – as shown in the manuscript :) .

The parts about degradation can be removed as degradation has a minimal role compared to transcription, in particular within a 1-hour window, which is the conclusion of the Rabani paper.

I am just trying to help the authors from grossly over-stating what can be achieved with Live-seq over metabolic labelling. Having the ability with Live-seq to sample the same cell twice is unique and offers new experimental strategies but it will not have any impact on studying direct effects of perturbations on transcription – there metabolic labelling is drastically better – yet reading this manuscript gives the complete opposite impression. Like I said in the first comments, Live-seq is better at longer timescales – which is great.

Rebuttal for Manuscript #2021-03-04452D

Chen*, Guillaume-Gentil* et al.

Author Rebuttals to Second Revision:

Reviewer 2

Regarding Point 1. As pointed out in the manuscript (line 405), NFKBIA is a known inhibitor of NFkB, and NFkB is linked to TNF expression. Therefore, it appears to us that the anticorrelation of the baseline NFKBIA and the TNF response is not a surprising finding but rather a confirmation of the expected finding and a validation of the method (which the authors acknowledge in their comments). Indeed it is interesting that the inhibitory effect of NFKBIA on NFkB can be observed unbiasedly at a single-cell level, but we believe our understanding of the NFkB signaling pathway or its heterogeneity in cells has not changed.

As Point 1 is highly related to Point 3, we kindly refer the reviewer to our response to Point 3 below.

Regarding Point 2. Given that longitudinal sampling is live-seq's biggest innovation, paired cells should perhaps be at the crux of the manuscript. From that lens, the few cells that have paired measurements are essentially the only cells that were fully assessed by live-seq; this limits the conclusions that can be drawn from the biology that is observed. From this point of view, this further highlights the secondary role of biological observations in this manuscript.

We thank the reviewer for this feedback. In our view, the biggest innovation of Live-seq is the profiling of the transcriptome of a cell without killing it, which then allows us to use this same cell for further phenotypic or molecular (such as with sequential sampling) profiling. Sequential sampling is thus a direct application of this innovation.

To avoid a misunderstanding and to make sure the main achievement is conveyed more clearly, we have now edited the abstract and restructured and rewritten parts of our introduction and discussion. We now explicitly define the main innovation (as described above), and then delineate the different downstream profiling strategies that can be followed, with the caveat that sequential profiling is currently still a proof-of-concept given its lack of scale (and thus power) (and as explicitly stated in the original and revised manuscript: "we sought to establish a proof-of-principle, sequential Live-seq sampling approach..", **lines 321-322**). Finally, revising the introduction and discussion also allowed us to better contrast Live-seq with other single cell transcriptomics methods, aiming to guide the reader when to use one over the other, depending on the desired application.

Regarding Point 3. While we appreciate the potential of the method and the comparison to scSLAM-seq (Erhard et al., Nature, 2019), we maintain that a conclusive

example of how it can answer an outstanding biological question is missing for publication in this journal, especially given the authors' previous Cell paper.

We maintain that we have provided such an example, showing how low *Nfkb1a* primes the macrophage cell line to respond stronger to LPS. It is of course true that a regulator of the NF- κ B pathway is not too unexpected in this process. However, as far as we know, it has never been described that especially heterogeneity of expression of this gene in steady state (as compared to the many other genes involved in the NF κ B pathway / LPS response) can affect the rate by which this cell responds. This is exactly why we believe that this finding showcases Live-seq very well: it is not too exotic, so it can be understood by a broad audience, while at the same time demonstrating that differences in gene expression - not visible as a separate cluster upon dimensionality reduction - can have a functional effect, and can be detected by Live-seq. In addition, we not only uncovered that *Nfkb1a* affects the macrophage LPS response, but also demonstrated that it is *the* most important modulator of this response, at least on an individual gene basis, since we were able to use Live-seq to generate a genome-wide ranking of the extent to which each detected gene contributes to this phenotype, which is also a novel achievement.

Regarding Point 4. Analyses performed separately by system should be the ones included in the manuscript. Also, for RNA velocity analyses, the genes chosen as input have a large impact on the results. Only genes relevant for a given system and its process of interest should be chosen – consider describing how this is done. Combining all systems together for gene selection and beyond has a strong potential for bias.

We now include both the “per cell type” and “all combined” analyses in the manuscript (**Extended Data Fig. 4e, f** and **lines 361-362**). For the RNA velocity analyses, we provide additional detail in the **Methods (lines 1053-1066)**.

Regarding Point 6. Clustering accuracy depends on cell types measured. For example, in Mereu et al (Nat Biotech 2020), HEK cells clustered separately from immune cells more readily than did different types of immune cells to each other. We would expect RAW and IBA to readily cluster separately given their drastic difference in transcriptome. Could a comparison to the clustering obtained from the scRNAseq be done?

Legend:

- (A) Clustering tree of the Seurat-based clustering results of the scRNA-seq or Live-seq data. It visualizes the relationship between clustering at increasing resolutions (top to bottom). The size of the circles represents the number of cells in that cluster, while the opacity of the arrows shows the proportion of cells from one cluster to another at different resolutions. (B) tSNE plots of the scRNA-seq or Live-seq data showing the final clustering (left) and the ground truth (right). The Adjusted Rand Index (ARI) comparing the two is each time specified. (C) Barplots showing the overlap in number of cells between the clustering (x-axis) and the ground truth (colors).

We found that cell types, including RAW and IBA, clustered separately both in Live-seq and scRNA-seq, even for low resolutions of the graph-based clustering implemented in Seurat (**Rebuttal Fig. A** and **Extended Data Fig. 2g, 2p**) and with very high accuracy (99.8% and 99.0% of the cells were assigned to the correct cell types for scRNA-seq and Live-seq respectively). We note that ASPCs were probed two days post-adipogenic cocktail induction to minimize sampling interference by lipid droplets, meaning that the cells were still early in their differentiation trajectory. Due to this molecular similarity between ASPC_Pre and ASPC_Post compared to the other probed cell types/states, it is only at high resolution that the ASPCs split. We therefore decided to use low-resolution clustering and independently adapted the clustering for the clustered ASPCs to correctly capture their state difference. We calculated the Adjusted Rand Index comparing our final clustering and the ground truth and obtained good clustering accuracy for both techniques (**Rebuttal Fig. B** and **Extended Data Fig. 2f, 2o**). Both for Live-seq and scRNA-seq, a few cells were misassigned to the correct treatment (**Rebuttal Fig. C** and **Extended Data Fig. 2h, 2q**). For combined ASPCs and RAW cells, these represented 7.6% and 2% of the cells for Live-seq and scRNA-seq, respectively. Thus, the misassignment was slightly higher for Live-seq, potentially due to its lower sensitivity compared to scRNA-seq.

Please note that the comment of the reviewer on clustering accuracy made us revise the differential expression analysis. Indeed, it was previously based on the cell type/state assignment and, to be more unsupervised, it is now based on the clustering results. While the general message / conclusion does not change, some plots were added/modified accordingly, explaining subtle changes (**Fig. 2e** and **ED Fig. 2f, 2k-m, 2r-t, 2v**).

Regarding Point 9. We suggest the authors look to SMART-Seq3 (<https://www.nature.com/articles/s41587-020-0497-0>) for guidance.

Thank you for the suggestion. We had already previously referred to this paper in our discussion, but now also explicitly mention the name of the method.

Regarding Point 10. How does downsampling improve the overlap between scRNAseq and Live-seq? Could the authors quantify this?

This is an expected result in our opinion. Given that down-sampling will lower the power of scRNA-seq to the same level as that of Live-seq, only the highest expressed genes, or those with the highest effect size, will remain differentially expressed after down-sampling. The consequence is that the relative overlap increased (from 20.3% to 28.5% overlap for the ASPCs and from 14.6% to 19% overlap for RAW cells), but the absolute number of genes decreased (from 270 to 183 genes for the ASPCs and from 410 to 256 genes for the RAW cells).

Regarding Point 19. Consider specifying this is the manuscript – it was not clear to us that the goal of sequential sampling here was purely demonstrative given that this is one of the promises of the technique.

We kindly refer the reviewer to our response to point 2. We aimed to make this clearer now in the abstract, introduction, and discussion.

Reviewer 3

The authors have improved the *Nfkb* validation experiment with a second experiment that show more convincing significance. I would advise the authors, however, to tone down the language around this observation. Currently, the abstract list *Nfkb* as a “major phenotypic determinant” and the main text says “principal driver” and “negative predictor”. Yet, the R^2 of 0.1 is a very weak, although significant, interaction.

We thank the reviewer for acknowledging the improvement of the *Nfkb* validation experiment. In our Live-seq data, *Nfkb* is the top hit to predict the heterogeneity of the LPS response. Nevertheless, it is possible that there are other factors that contribute little to the heterogeneity but collectively do. We therefore followed the

reviewer's instruction and modified the abstract (line 48) and main text (line 92, 402, 420, 424) accordingly.

Metabolic labeling vs. Live-seq

I will spell out my point here – take it or leave it – I still think the current discussion (and rebuttal text) is highly misleading and should be corrected.

If a researcher would like to study the effect of a perturbation on the transcriptome, say the direct effects of LPS on gene expression. A Live-seq user would then sample a part of the cytoplasm before the perturbation (timepoint t0) and then collect the cell or cytoplasm after 1 hour exposure to the perturbation (timepoint t1). The problem here is that the 1 hour timepoint would be highly confounded by RNAs present before the perturbation. In fact, in a typical in vitro grown cell only 5% of the transcriptome is transcribed within the last 1hr, making 95% of the RNA profiled at timepoint t1 being actually the remaining pre-existing RNA at timepoint t0 that was not sampled in the first Live-seq experiment. Therefore, the power to detect a change at 1hr relative to t0 will be highly limited. Most power would lie in identifying genes that were not expressed/present at t0 as they would have expression at t1 that is clearly detectable and different from t0. However, if a gene was abundant at t0 and say 2-fold induced at t1, that signal would be lost due to fact that a 2-fold change occurring in only 5% of the RNA would be diluted away. Therefore, I withhold that Live-seq will have limited power to detect shorter-timescales direct effects of perturbations onto gene expression.

A metabolic labeling user would provide the cells 4sU/EdU at t0 together with the perturbation, and collect the cells at t1 for analysis. Comparing separate cells that were exposed to perturbation or not, the analysis can focus on the RNAs transcribed only since the perturbation therefore avoiding the dilution pitfalls of the Live-seq scenario above. Therefore, differential expression can be detected even for genes that were highly expressed at t0 in contrast to Live-seq. Therefore, metabolic labeling will have drastically improved power for any short-term experiment that aim to identify the direct effects of a perturbation on transcription. Here, Live-seq (like a standard total RNA comparison on unmatched samples) will have low power to find any difference) – as shown in the manuscript :).

The parts about degradation can be removed as degradation has a minimal role compared to transcription, in particular within a 1-hour window, which is the conclusion of the Rabani paper.

I am just trying to help the authors from grossly over-stating what can be achieved with Live-seq over metabolic labelling. Having the ability with Live-seq to sample the same cell twice is unique and offers new experimental strategies but it will not have any impact on studying direct effects of perturbations on transcription – there metabolic labelling is drastically better – yet reading this manuscript gives the complete opposite

impression. Like I said in the first comments, Live-seq is better at longer timescales – which is great.

We thank the reviewer for the advice and for explicitly making this argument. The latter made us aware that there might be a misunderstanding. For studying short-term gene expression dynamics after a perturbation, we fully agree that metabolic labeling or similar technologies are more powerful than Live-seq. However, we would like to highlight that we did not develop Live-seq to study perturbation-related expression changes. Rather, we engineered Live-seq to study the expression differences among individual cells at their initial state, which may explain the magnitude and/or rate by which a perturbation affects the cell. There, Live-seq does excel as we can directly observe the full transcriptome before a perturbation, without having to infer it or make strong assumptions to do so, as acknowledged for metabolic labeling-based tools (e.g. Cao et al., Nature Biotechnology, 2020).

To avoid confusing the reader with this, we had a fresh look at our abstract, introduction and discussion, and substantially revised it:

- In the introduction, we now more explicitly mention the problem that Live-seq tries to solve, namely the profiling of an initial transcriptome before a perturbation.
- In the second and third paragraphs of the discussion, we now explicitly mention the use cases where Live-seq excels, and those where alternative approaches are potentially better, while considering the practical challenges and biological assumptions made by all technologies.

Still, we do want to respond to some of the claims made by the reviewer for the sake of scientific discussion:

- The 5% argument only holds for the “whole transcriptome”, and so includes all housekeeping genes. Many genes of interest have much lower half-lives though, including *Nfkb1a*, which, according to the scSLAM-seq paper, has a half-life of half an hour. In this case, 75% of the mRNA will be turned over every hour.
- With respect to degradation, we do not agree that this has a minimal role. In fact, Rabani et al. (Nature Biotechnology, 2011) notes that still over 500 genes have dynamic degradation, and these genes are enriched for e.g. NF-κB signaling. While this is indeed a relatively small fraction of all 20k genes, this ignores the fact that most of these 20k genes are not relevant in the cell type anyway.

Reviewer Reports on the Third Revision:

Referees' comments:

Referee #2 (Remarks to the Author):

The authors have adequately address several of the issues we raised in this revision, but a few clarifications are still needed. Overall, while we applaud the authors for demonstrating the potential to make transcriptomic measurements from live cells, we maintain that the biological insight afforded by this manuscript is small – especially considering the multitude of very interesting questions that could be answered with this approach; as such, we strongly encourage the authors to appropriately position their work (e.g.,).

Regarding Point 2. Thank you for clarifying the innovation of the technique. We urge you to make further modification to the abstract to decrease the emphasis on sequential sampling and more clearly state when elements are validatory. As written, it appears that your longitudinal sampling was used to give insight into the macrophage LPS-response. This is misleading given your statement that this is but a proof-of-concept. Relatedly, a more nuanced discussion of how sampling might impact cellular response dynamics would be a welcome addition as the positioning in the conclusion is too strong given what's shown.

Regarding Point 3. We agree with the authors that the NFKBIA evidence is a good proof-of-concept but maintain that the biological insight provided is small (also see point 2 above re: some of the claims made in the rebuttal). We look forward to seeing Live-seq's true potential in the future.

Regarding Point 4. Thank you for clarifying the methods. Please also include information on how genes were chosen to represent the system of interest. This highlights the necessity of doing this analysis on a per system basis, since different genes will be important to describe dynamics of different systems.

Regarding Point 10. It is more appropriate to report the downsampled numbers given the difference in coverage between the two techniques, which creates a bias.

Referee #3 (Remarks to the Author):

The revised manuscript has takes my last comments into consideration satisfactorily.

Laboratory of Systems Biology and Genetics
Bart Deplancke, School of Life Sciences
Building SV 3823
Station 19, CH-1015 Lausanne

<http://deplanckelab.epfl.ch>
+41216939681
bart.deplancke@epfl.ch
[@BartDeplancke](https://twitter.com/BartDeplancke)

Author Rebuttals to Third Revision:

Referees' comments:

Referee #2 (Remarks to the Author):

The authors have adequately address several of the issues we raised in this revision, but a few clarifications are still needed. Overall, while we applaud the authors for demonstrating the potential to make transcriptomic measurements from live cells, we maintain that the biological insight afforded by this manuscript is small – especially considering the multitude of very interesting questions that could be answered with this approach; as such, we strongly encourage the authors to appropriately position their work (e.g.,).

>We thank the reviewer for appreciating the novelty of our work and the many questions that can now be addressed using Live-seq. We believe that in our last revision, we have adequately highlighted the main advances that our technology offers, whilst also discussing current limitations and future opportunities.

Regarding Point 2. Thank you for clarifying the innovation of the technique. We urge you to make further modification to the abstract to decrease the emphasis on sequential sampling and more clearly state when elements are validatory. As written, it appears that your longitudinal sampling was used to give insight into the macrophage LPS-response. This is misleading given your statement that this is but a proof-of-concept. Relatedly, a more nuanced discussion of how sampling might impact cellular response dynamics would be a welcome addition as the positioning in the conclusion is too strong given what's shown.

>Regarding the abstract, we have now replaced “most importantly” with “In addition” to emphasize that the “sequential Live-seq” and “transcriptomic recorder” experiments are not directly connected. In the main text, we have emphasized the desire to further lower the cytoplasmic extraction volumes, aiming to reduce any cellular impact that this extraction may have to a minimum, which is obviously highly relevant for studying cellular dynamics, especially in smaller cells such as T cells and stem cells (Lines 485-488).

Regarding Point 3. We agree with the authors that the NFKBIA evidence is a good proof-of-concept but maintain that the biological insight provided is small (also see point 2 above re: some of the claims made in the rebuttal). We look forward to seeing Live-seq's true potential in the future.

> We agree with the reviewer that Live-seq opens up new methodological avenues to address outstanding biological questions.

Regarding Point 4. Thank you for clarifying the methods. Please also include information on how genes were chosen to represent the system of interest. This highlights the necessity of doing this analysis on a per system basis, since different genes will be important to describe dynamics of different systems.

Laboratory of Systems Biology and Genetics
Bart Deplancke, School of Life Sciences
Building SV 3823
Station 19, CH-1015 Lausanne

<http://deplanckelab.epfl.ch>
+41216939681
bart.deplancke@epfl.ch
[@BartDeplancke](https://twitter.com/BartDeplancke)

> The information about the selection of genes was already included in the Methods section (Line 1138-1140).

Regarding Point 10. It is more appropriate to report the downsampled numbers given the difference in coverage between the two techniques, which creates a bias.

> The scRNA-seq data is down-sampled per cell to have the same density distribution of the number of features as the corresponding Live-seq data, rather than an absolute number. This information was indicated in the Methods section (Line 1098-1106).

Referee #3 (Remarks to the Author):

The revised manuscript has takes my last comments into consideration satisfactorily.

>Thank you.